# Adaptive gradient descent on Riemannian manifolds and its applications to Gaussian variational inference

**Jiyoung Park**[*]
Texas A&M University
wldyddl5510@tamu.edu

**Jaewook J. Suh**[*]
Rice University
jacksuh@rice.edu

**Bofan Wang**
Nanjing University
wangbf@smail.nju.edu.cn

**Anirban Bhattacharya**
Texas A&M University
anirbanb@stat.tamu.edu

**Shiqian Ma**
Rice University
sqma@rice.edu

## Abstract

We propose RAdaGD, a novel family of adaptive gradient descent methods on general Riemannian manifolds. RAdaGD adapts the step size parameter without line search, and includes instances that achieve a non-ergodic convergence guarantee, $f(x_k) - f(x_\star) \leq \mathcal{O}(1/k)$, under local geodesic smoothness and generalized geodesic convexity. A core application of RAdaGD is Gaussian Variational Inference, where our method provides the first convergence guarantee in the absence of $L$-smoothness of the target log-density, under additional technical assumptions. We also investigate the empirical performance of RAdaGD in numerical simulations and demonstrate its competitiveness in comparison to existing algorithms.

## 1 Introduction

Riemannian optimization is an actively studied area due to its various applications relating to machine learning, computer vision, and statistics (Fillard et al., 2005; Bačák, 2014a; Hosseini & Sra, 2015; 2020; Kim et al., 2025a). In this paper, we consider the Riemannian optimization problem

$$\min_{x \in N} f(x)$$

where $N \subseteq M$ is a *geodesically convex* subset of a Riemannian manifold $M$, and $f \colon N \to \mathbb{R}$ is a continuously differentiable *geodesically convex* function. A standard approach to solve this problem is Riemannian gradient descent (RGD):

$$x_{k+1} = \exp_{x_k}(-s_k \operatorname{Grad} f(x_k)).$$

Here, $\exp(\cdot)$ denotes the exponential map at $x$, $s_k$ is the step size at iteration $k$, and $\operatorname{Grad}$ represents the Riemannian gradient. Asymptotic properties of RGD have been extensively studied in the literature; see, for example, (Absil et al., 2008; Bačák, 2014a). When $f$ is additionally a geodesically $L$-smooth function, RGD is known to achieve the convergence rate $f(x_k) - f(x_\star) \leq \mathcal{O}(1/k)$ with a constant step size $s_k = \frac{1}{L}$ (Zhang & Sra, 2016).

However, the use of RGD with a fixed step size of $\frac{1}{L}$ has limitations since:

(i) assuming an *L-smooth* function with geodesic convexity is arguably a strong assumption,

(ii) it requires *prior knowledge* of the smoothness parameter $L$.

To overcome this issue, the study of *adaptive algorithms* in Euclidean space has emerged recently (Malitsky & Mishchenko, 2020; 2024; Latafat et al., 2024a;b; Oikonomidis et al., 2024; Zhou et al., 2025; Li & Lan, 2025; Suh & Ma, 2025). However, to the best of our knowledge, the extension of these methods to the Riemannian optimization setting remains largely unexplored at this stage.

---

[*]Equal contribution, alphabetically ordered.

In this work, we propose **Riemannian Adaptive Gradient Descent (RAdaGD)**, inspired the by Euclidean adaptive method proposed in Suh & Ma (2025). RAdaGD is a family of novel line-search-free adaptive algorithms on Riemannian manifolds, which includes instances that guarantee the convergence rate $f(x_k) - f(x_\star) \leq \mathcal{O}(1/k)$, similar to RGD with constant step-size. We relax the usual $L$-smoothness assumption to *local smoothness* and show that this condition is satisfied by all $C^2$ functions.

Building on this local smoothness setup, we propose a new algorithm with provable guarantees. Our main focus is *Gaussian variational inference* (GVI), which approximates a target distribution $\pi$ on $\mathbb{R}^n$, most prominently a Bayesian posterior distribution, by a non-singular Gaussian. Interest in GVI is intimately connected to the celebrated Bernstein–von Mises theorem, which states that under suitable regularity conditions on the likelihood function and prior distribution, a Bayesian posterior asymptotically approaches a Gaussian shape (van der Vaart, 1998, Chapter 10). Yet, existing GVI algorithms provide convergence guarantees only for $L$-*smooth* target log-densities (Lambert et al., 2022; Quiroz et al., 2023; Diao et al., 2023; Kim et al., 2023). To the best of our knowledge, ours is the first method with provable guarantees in the non-$L$-smooth setting. Moreover, numerical results show it is competitive with or outperforms prior algorithms.

### 1.1 CONTRIBUTIONS

We summarize the contributions of this paper as follows.

- We provide RAdaGD, a family of novel adaptive gradient descent methods for Riemannian optimization. RAdaGD automatically tunes the step size by approximating the local smoothness parameter, without an expensive calculation such as line search. We provide instances that achieve a non-ergodic convergence rate $f(x_k) - f(x_\star) \leq \mathcal{O}(1/k)$. To the best of our knowledge, this is the first of its kind for Riemannian adaptive algorithms.

- While RAdaGD is inspired by Suh & Ma (2025), its extension to Riemannian manifolds requires highly nontrivial arguments. Our algorithms differ from Suh & Ma (2025) in the step size rule and the analyses, specifically to handle the parameter $\zeta$ related to manifold curvature. We first provide the analysis when $\zeta$ is bounded and known, and also consider cases when it is unbounded or unknown.

- Our results are provided for *locally geodesically smooth* and generalized geodesically convex functions. Although generalized geodesic convexity can be considered a somewhat strong assumption, local geodesic smoothness is a weaker assumption compared to the conventional $L$-smooth assumption, allowing a broader class of functions. In particular, we show that all $C^2$ functions on a complete Riemannian manifold are locally geodesically smooth.

- Leveraging the advanced properties of the locally smooth setup discussed above, we show that RAdaGD can be applied to Gaussian variational inference (GVI) under an additional technical assumption. To the best of our knowledge, our algorithm is the first to guarantee convergence for this application without assuming $L$-smoothness of target log-density. While having a superior theoretical convergence rate, we also show that our algorithm performs among the best in practice. We also validate the effectiveness of our algorithm through numerical experiments in possibly negatively curved spaces.

## 2 PRELIMINARIES

We provide the organization of prior works in Appendix A. A brief introduction to the Riemannian notation is given in this section, while the detailed explanation is deferred to Appendix B.1. Throughout this paper, we denote $M$ as a $n$-dimensional differentiable manifold and $\langle \cdot, \cdot \rangle_x$ as the inner product defined on each tangent space $T_x M$, where $x \in M$. We also denote $\|v\|_x := \sqrt{\langle v, v \rangle_x}$ and omit the subscript when there is no ambiguity. For $x, y \in M$, we denote the distance by $d(x, y)$. We denote geodesic curves (curves with zero acceleration) by $\gamma$. A geodesic that satisfies $\gamma(0) = x$, $\gamma(1) = y$, and $d(x, y) = \int_0^1 \|\gamma'(t)\| \, dt$ is called a minimizing geodesic. The exponential map $\exp_x \colon T_x M \to M$ is defined by $\exp_x(v) = \gamma(1)$, where $\gamma(0) = x$ and $\gamma'(0) = v$. Here, $T_x M$ is the tangent space at $x$. We call the locally well-defined inverse the logarithmic map and denote it

by $\log_x$. The parallel transport $\Gamma_x^y \colon T_x M \to T_y M$ transports tangent vectors parallelly along the geodesic curve $\gamma$ such that $\gamma(0) = x$ and $\gamma(1) = y$.

Given a differentiable function $f : M \to \mathbb{R}$, the Riemannian gradient of $f$ at $x$, denoted by $\mathrm{Grad}\, f(x) \in T_x M$ is the tangent vector satisfying $\langle \mathrm{Grad}\, f(x), v \rangle_x = df_x(v)$ fod all $v \in T_x M$. Here, $df_x$ is a differential of $f$ at $x$. We say $N \subseteq M$ is a geodesically convex subset of $M$ if for all $x, y \in N$ there exists a geodesic $\gamma$ such that $\gamma(0) = x, \gamma(1) = y$, and $\gamma(t) \in N$ for all $t \in [0, 1]$. We say a differentiable function $f : N \to \mathbb{R}$ is geodesically convex if

$$f(y) \geq f(x) + \langle \mathrm{Grad}\, f(x), \log_x y \rangle_x, \quad \forall x, y \in N. \tag{1}$$

We impose the following assumptions on the manifold and the function, which are standard in Riemannian optimization literature (Alimisis et al., 2020; 2021; Kim & Yang, 2022).

**Assumption 2.1.** *The manifold $M$ and the target function $f$ satisfy the following conditions.*

*(i) $M$ is a smooth complete manifold, and $N \subseteq M$ is a geodesically convex subset of $M$. $N$ has a sectional curvature lower bound $K_{\min} > -\infty$.*

*(ii) Exponential maps, logarithmic maps, and parallel transports are all well-defined via the minimizing geodesic on $N$. In particular, the exponential map and parallel transports are computationally tractable.*

*(iii) The function $f : N \to \mathbb{R}$ admits a minimizer on $N$.*

Based on the first assumption, we define a parameter $\zeta : [0, \infty) \to [1, \infty)$ related to $K_{\min}$ as

$$\zeta(\rho) := \begin{cases} 1 & \text{if } K_{\min} \geq 0, \\ \rho \sqrt{-K_{\min}} \coth\left(\rho \sqrt{-K_{\min}}\right) & \text{otherwise.} \end{cases} \tag{2}$$

In most cases, we will consider $\rho = d(x, x_\star)$ for some $x \in N$. Lastly, we follow the arithmetic convention that $\frac{1}{0} = \infty$ and $\frac{0}{0} = 0$.

## 3 LOCAL GEODESIC SMOOTHNESS & GENERALIZED GEODESIC CONVEXITY

In this section, we define local geodesic smoothness and generalized geodesic convexity, which are central to our analysis. We begin with local geodesic smoothness. This assumption allows for a wider range of functions compared to the conventional geodesic $L$-smoothness assumption usually considered in gradient descent. We discuss the details in the next subsection.

**Definition 3.1** (Local geodesic smoothness). *A function $f : M \to \mathbb{R}$ is called* locally geodesically smooth *if for every compact set $K \subseteq M$ there exists a constant $L_K$ such that*

$$\|\Gamma_x^y \mathrm{Grad}\, f(x) - \mathrm{Grad}\, f(y)\|_y \leq L_K d(x, y), \qquad \forall x, y \in K.$$

We next define generalized geodesic convexity (Ambrosio et al., 2008, Definition 9.2.4), also known as convexity along generalized geodesics. While the definition was initially formulated in Wasserstein space, we use the extended definition for a general Riemannian manifold introduced in Park et al. (2025, Definition C.1). As mentioned in Altschuler et al. (2021, § A.2), the added flexibility of generalized geodesics is sometimes crucial for the study of RGD.

**Definition 3.2** (Generalized geodesic convexity). *A function $f : N \to \mathbb{R}$ is called generalized geodesically convex with base $z \in M$ if for all $x, y \in N$, we have,*

$$f(y) \geq f(x) + \langle \Gamma_x^z \mathrm{Grad}\, f(x), \log_z y - \log_z x \rangle_z. \tag{3}$$

*$f$ is called* generalized geodesically convex *if (3) holds for all $z \in N$.*

We clarify that generalized geodesic convexity implies geodesic convexity (1) by taking $z = x$, and therefore constitutes a stronger assumption. Still, we note that various applications have been studied under this assumption (Salim et al., 2020; Chewi et al., 2020; Ahn & Chewi, 2021; Altschuler et al., 2021; Diao et al., 2023; Park et al., 2025).

### 3.1 IMPORTANT PROPERTIES OF THE SETUP

We now show that every $C^2(M)$ function on a complete Riemannian manifold is locally geodesically smooth. Even though this statement is somewhat trivial in Euclidean space, we were unable to find its formal extension to Riemannian manifolds. Since this property is a core ingredient that enables fruitful applications of our setup, we state it as a theorem to emphasize its importance.

**Theorem 3.3.** *Let $M$ be a smooth complete Riemannian manifold. Assume that $f \in C^2(M)$. Then, $f$ is locally geodesically smooth.*

We provide the proof in Appendix C.1. Theorem 3.3 shows that the local geodesic smoothness is a nice relaxation of the usual geodesic smoothness. For example, the squared distance function $x \mapsto d^2(x, p)$ is not globally geodesically $L$-smooth on non-Euclidean Hadamard manifold (Criscitiello & Kim, 2025), but it is locally geodesically smooth as it is a $C^2$ function.

Now we prove the local version of the co-coercivity condition on a Riemannian manifold. This extends the Euclidean version proposed in Suh & Ma (2025, Lemma 1). The proof is deferred to Appendix C.3.

**Proposition 3.4.** *Let $K$ be a compact set on $N$. Assume that $f : N \to \mathbb{R}$ is locally geodesically smooth and generalized geodesically convex. Suppose $K \subseteq \overline{B}(c, R)$ and $\overline{K} := \overline{B}(c, 3R) \subseteq N$ for some $c \in N$ and $R > 0$. Denote the local smooth parameter of $\overline{K}$ as $L_{\overline{K}}$. Then, for all $x, y \in K$,*

$$f(x) - f(y) - \left\langle \mathrm{Grad}\, f(y), \log_y x \right\rangle_y - \frac{1}{2L_{\overline{K}}} \left\| \Gamma_y^x \mathrm{Grad}\, f(y) - \mathrm{Grad}\, f(x) \right\|_x^2 \geq 0. \quad (4)$$

## 4 CONVERGENCE ANALYSIS OF RADAGD

Our ***R**iemannian **Ada**ptive **G**radient **D**escent (RAdaGD)* method is described in Algorithm 1. Our algorithm is tractable as long as the exponential maps, Riemannian gradients, and parallel transports are computationally tractable.

---

**Algorithm 1 R**iemannian **Ada**ptive **G**radient **D**escent (RAdaGD)

1: **Input:** $x_0 \in \mathbb{R}^n$, $s_0 > 0$, $\{A_k\}_{k \geq 0}$, $\{B_k\}_{k \geq 0}$, $\{\tilde{B}_k\}_{k \geq 0}$, $A_{-1} = 0$
2: **for** $k = 0, 1, \dots$ **do**
3:

$$x_{k+1} = \exp_{x_k}(-s_k \mathrm{Grad}\, f(x_k))$$

$$L_{k+1} = -\frac{1}{2} \frac{\left\| \Gamma_{x_{k+1}}^{x_k} \mathrm{Grad}\, f(x_{k+1}) - \mathrm{Grad}\, f(x_k) \right\|_{x_k}^2}{f(x_{k+1}) - f(x_k) + s_k \left\langle \Gamma_{x_{k+1}}^{x_k} \mathrm{Grad}\, f(x_{k+1}), \mathrm{Grad}\, f(x_k) \right\rangle_{x_k}} \quad (5)$$

$$r_k^s = \min\left\{ \frac{A_{k-1} + 1}{A_k}, \frac{A_k}{\tilde{B}_{k+1}} \right\}, \qquad r_k^L = \begin{cases} \frac{A_k}{\tilde{B}_{k+1}} & \text{if } s_k < \frac{1}{L_{k+1}} \\ \left( \frac{A_k}{B_k} + \frac{\tilde{B}_{k+1}}{A_k} \right)^{-1} & \text{if } s_k \geq \frac{1}{L_{k+1}} \end{cases} \quad (6)$$

$$s_{k+1} = \min\left\{ r_k^s s_k, \ r_k^L \frac{1}{L_{k+1}} \right\} \quad (7)$$

4: **end for**

---

Throughout this section, with the definition (2) in mind, we denote

$$\bar{\zeta} := \sup_{x \in N} \zeta\left( d(x, x_\star) \right), \qquad \zeta_k = \zeta\left( d(x_k, x_\star) \right).$$

The algorithm above is defined by the sequences $A_k$, $B_k$, and $\tilde{B}_k$. To start with the simplest case, we assume $\bar{\zeta}$ is upper bounded and known. Such an assumption commonly appears in the Riemannian optimization literature (Zhang & Sra, 2016; Zhang et al., 2016; Alimisis et al., 2021; Kim & Yang, 2022). We provide a representative choice of $A_k$, $B_k$, and $\tilde{B}_k$ that achieves the non-ergodic convergence rate $f(x_k) - f_\star \leq \mathcal{O}(1/k)$, which matches the non-adaptive rate. To the best of our knowledge, this rate is the first of its kind for Riemannian adaptive methods.

**Theorem 4.1.** *Assume that $\bar{\zeta} < \infty$ and known. Suppose $f : N \to \mathbb{R}$ is locally geodesically smooth and generalized geodesically convex. Suppose $\{x_k\}_{k\geq 0}$ is generated by RAdaGD with $A_k = \alpha(k+1) + 1 + \bar{\zeta}$, $B_k = \alpha(k+1)$ and $\tilde{B}_k = \alpha(k+1) + \bar{\zeta}$, where $\alpha \in (0,1]$. Define $L_0$ as*

$$L_0 = \frac{\left\| \Gamma_{\tilde{x}_0}^{x_0} \operatorname{Grad} f(\tilde{x}_0) - \operatorname{Grad} f(x_0) \right\|_{x_0}}{\left\| \log_{x_0} \tilde{x}_0 \right\|_{x_0}} \tag{8}$$

*with arbitrary $\tilde{x}_0$ and initial point $x_0$ such that $x_0 \neq \tilde{x}_0$. Let $r = \left( \frac{A_0}{B_0} + \frac{B_1 + \bar{\zeta}}{A_0} \right)^{-1}$, $s_0 = rA_0 \frac{1}{L_0}$, $R^2 = d(x_0, x_\star)^2 + (B_0 + \zeta_0)s_0^2 \left\| \operatorname{Grad} f(x_0) \right\|^2$ and $L$ be a smoothness parameter of $f$ on $\bar{B}_{3R}(x_\star) \cup \bar{B}_{3d(\tilde{x}_0, x_\star)}(x_\star)$. Then the following convergence rates hold for function value:*

$$f(x_k) - f_\star \leq \frac{L}{2r} \frac{1}{\alpha(k+1) + 1 + \bar{\zeta}} R^2 = \mathcal{O}\left( \frac{L}{k} \right) \tag{9}$$

*Also, the following convergence rates hold for gradient norm square:*

$$\min_{i \in \{0,\ldots,k\}} \left\| \operatorname{Grad} f(x_i) \right\|^2 \leq \frac{2L^2}{r^2} \frac{1}{\alpha(k+1)(k+2)} R^2 = \mathcal{O}\left( \frac{L^2}{k^2} \right).$$

## 4.1 PROOF OF GENERALIZED CHOICE OF $A_k$, $B_k$ AND $\tilde{B}_k$

To prove Theorem 4.1, we establish the convergence rate for a more general choice of $A_k$, $B_k$ and $\tilde{B}_k$. We start by establishing a Lyapunov analysis. While our analysis is inspired by Suh & Ma (2025, Proposition 12), $\tilde{B}_k$ is introduced as a new parameter to handle $\zeta_k$. We provide the proof in Appendix D.1.

**Proposition 4.2.** *Suppose $f : N \to \mathbb{R}$ is locally geodesically smooth and generalized geodesically convex. Suppose $\{x_k\}_{k\geq 0}$ is generated by RAdaGD with $\{A_k\}_{k\geq 0}$, $\{B_k\}_{k\geq 0}$, and $\{\tilde{B}_k\}_{k\geq 0}$ satisfying*

$$\frac{A_k + 1}{A_{k+1}} \geq 1, \qquad \frac{A_k}{\tilde{B}_{k+1}} \geq 1, \qquad \left( \frac{A_k}{B_k} + \frac{\tilde{B}_{k+1}}{A_k} \right)^{-1} \geq r, \qquad \forall k \geq 0, \tag{10}$$

*with some $r \in (0,1)$. Moreover, let $s_0 = r \max\left\{ A_0, \frac{\tilde{B}_1}{A_0} \right\} \frac{1}{L_0}$, where $L_0$ is defined in (8). Define $A_{-1} = 0, B_{-1} = B_0 + \zeta_0$, $x_{-1} = x_0$, $s_{-1} = s_0$, and the Lyapunov function as*

$$V_k = s_{k+1}A_k(f(x_k) - f_\star) + \frac{1}{2}s_k^2 B_k \left\| \operatorname{Grad} f(x_k) \right\|^2 + \frac{1}{2}\left\| \log_{x_{k+1}} x_\star \right\|^2, \quad \forall k \geq -1, \tag{11}$$

*Fix $k \geq -1$. Suppose that the following inequality holds:*

$$\tilde{B}_{k+1} \geq B_{k+1} + \zeta_{k+1}. \tag{12}$$

*Then the following inequality holds:*

$$\begin{aligned}
V_{k+1} - V_k &\leq (s_{k+1}A_k + s_{k+1} - s_{k+2}A_{k+1})(f_\star - f(x_{k+1})) \\
&\quad - \min\left\{ \frac{A_k}{2} \frac{s_{k+1}}{L_{k+1}}, \frac{B_k}{2} s_k^2 \right\} \left\| \operatorname{Grad} f(x_k) \right\|^2 \leq 0.
\end{aligned} \tag{13}$$

We immediately obtain the following boundedness result from the above Lyapunov analysis.

**Corollary 4.3.** *In addition to (10), suppose that (12) holds for all $k \geq k_0 - 1$, for some $k_0 \geq 0$. Then, $\{x_k\}_{k\geq 0}$ is bounded. Specifically, $x_k \in \bar{B}_R(x_\star)$ for all $k \geq k_0 + 1$, where $R^2 = 2V_{k_0-1}$.*

Next, we establish a lower bound on the step size $s_k$. We can obtain this lower bound by leveraging the boundedness of the iterate $x_k \in \bar{B}_R(x_\star)$. The proof is deferred to Appendix D.2.

**Lemma 4.4.** *Let $f$ be geodesically smooth and generalized geodesically convex. Suppose $\{s_k\}_{k\geq 0}$ is generated by RAdaGD with $\{A_k\}_{k\geq 0}$, $\{B_k\}_{k\geq 0}$, $\{\tilde{B}_k\}_{k\geq 0}$ and $s_0$ satisfying the assumptions in Corollary 4.3. Let $L$ be a smoothness parameter of $f$ on $\bar{B}_{3R}(x_\star) \cup \bar{B}_{3d(\tilde{x}_0, x_\star)}(x_\star)$, where $\tilde{x}_0$ is the vector in (8) and $R$ is defined as in Corollary 4.3. Then, we have:*

$$s_k \geq \frac{r}{L}, \qquad \forall k \geq 1. \tag{14}$$

Leveraging Proposition 4.2 and Lemma 4.4, we obtain convergence rates for a generalized choice of $A_k$, $B_k$ and $\tilde{B}_k$. The proof is provided in Appendix D.3.

**Theorem 4.5.** *Suppose $f : M \to \mathbb{R}$ is locally geodesically smooth and generalized geodesically convex. Define $L_0$ as in (8). Suppose $\{x_k\}_{k\geq 0}$ is generated by RAdaGD with $\{A_k\}_{k\geq 0}$, $\{B_k\}_{k\geq 0}$, $\{\tilde{B}_k\}_{k\geq 0}$ and $s_0$ satisfying the assumptions in Corollary 4.3. Define $R$ as in Corollary 4.3 and $L$ as in Lemma 4.4.*

*(i) The following holds, which indicates a non-ergodic convergence rate when $\lim_{k\to\infty} A_k = \infty$:*

$$f(x_k) - f_\star \leq \frac{L}{2rA_k}R^2 = \mathcal{O}\left(\frac{L}{A_k}\right).$$

*As a result, applying Proposition 3.4 with $y = x_k$ and $x = x_\star$ we have*

$$\|\mathrm{Grad}\, f(x_k)\|^2 \leq \frac{L^2}{rA_k}R^2 = \mathcal{O}\left(\frac{L^2}{A_k}\right).$$

*(ii) Also, $\{x_k\}_{k\geq 0}$ achieves minimum selection convergence rate*

$$\min_{i\in\{k_0,\ldots,k\}}(f(x_i) - f_\star) \leq \frac{1}{2\sum_{i=k_0}^k s_i}R^2 = \mathcal{O}\left(\frac{L}{k}\right),$$

*and*

$$\min_{i\in\{k_0,\ldots,k\}}\|\mathrm{Grad}\, f(x_i)\|^2 \leq \frac{L^2}{r^2\sum_{i=k_0}^k \min\{A_i, B_i\}}R^2 = \mathcal{O}\left(\frac{L^2}{\sum_{i=k_0}^k A_i}\right).$$

*(iii) Lastly, we have $\lim_{k\to\infty} x_k = \bar{x}_\star$ for some minimizer $\bar{x}_\star$, when $\liminf_k \min\{A_k, B_k\} > 0$.*

Readers may notice that the convergence results above depend on $k_0$, which was mentioned in (12). However, we can set $k_0 = 0$ by properly defining $\tilde{B}_k$, as demonstrated in the corollary below. Note that the choices of $\{A_k\}_{k\geq 0}$, $\{B_k\}_{k\geq 0}$, and $\{\tilde{B}_k\}_{k\geq 0}$ in Theorem 4.1 are based on Corollary 4.6.

**Corollary 4.6.** *Assume that $\bar{\zeta} < \infty$ and known. Define $\tilde{B}_k = B_k + \bar{\zeta}$ for $k \geq 0$. Suppose $\{A_k\}_{k\geq 0}$ and $\{B_k\}_{k\geq 0}$ are chosen to satisfy (10). Then Theorem 4.5 holds true with*

$$k_0 = 0, \qquad R^2 = d(x_0, x_\star)^2 + (B_0 + \zeta_0)s_0^2\|\mathrm{Grad}\, f(x_0)\|^2.$$

In the above corollary, we assumed that the value of $\bar{\zeta}$ is bounded and known. This is a standard assumption in previous literature (Zhang & Sra, 2016; Zhang et al., 2016; Alimisis et al., 2021; Kim & Yang, 2022) and is unproblematic in the non-negatively curved setting, where $\bar{\zeta} = 1$. However, if the manifold is negatively curved, such an assumption can be restrictive. In the next subsection, we introduce two corollaries that extend to the case where each assumption is dropped.

## 4.2 EXTENSIONS TO POSSIBLY NEGATIVELY CURVED SPACES

Assuming $\bar{\zeta} < \infty$ can be ensured when the whole manifold is bounded, but such an assumption can be thought of as too restrictive. The following corollary demonstrates that this boundedness condition can be weakened through an inductive argument. Specifically, it suffices to assume that the initial distance $d(x_0, x_\star)$ is bounded above by a known constant $\bar{d}_0$, from which we can derive an explicit upper bound $\bar{\zeta}_0$ for $\zeta_0$. The boundedness of subsequent $\zeta_k$ (for $k > 0$) then follows inductively from the algorithm's update rules, and the choices of $\{A_k\}$, $\{B_k\}$, $\{\tilde{B}_k\}$. The proof is provided in Appendix D.5.

**Corollary 4.7** (Extension for unbounded $\bar{\zeta}$). *Assume that $d(x_0, x_\star)$ is bounded above by a known constant $\bar{d}_0$. Suppose $\{A_k\}_{k\geq 0}$ and $\{B_k\}_{k\geq 0}$ are chosen to satisfy (10). Denote $\bar{\zeta}_0 = \zeta(\bar{R})$ where $\bar{R}^2 = \bar{d}_0^2 + (B_0 + \zeta(\bar{d}_0))s_0^2\|\mathrm{Grad}\, f(x_0)\|^2$. Set $\tilde{B}_k = B_k + \bar{\zeta}_0$ for all $k \geq 0$. Then*

$$\zeta_k \leq \bar{\zeta}_0, \qquad \forall k \geq 0,$$

*and Theorem 4.5 holds true with $k_0 = 0$, $R^2 = d(x_0, x_\star)^2 + (B_0 + \zeta_0)s_0^2\|\mathrm{Grad}\, f(x_0)\|^2$.*

Corollary 4.7 tells us that the boundedness of $\bar{\zeta}$ in Corollary 4.6 is not mandatory. While Corollary 4.7 offers a weaker assumption compared to Corollary 4.6, it suffers from the requirement of prior knowledge of $\bar{\zeta}_0$. The following corollary states that, as a trade-off, if $\bar{\zeta}$ is known to be bounded, we can still achieve the same asymptotic convergence rate without any prior knowledge of $\zeta$. A similar idea–replacing the known $\zeta$ condition with a boundedness assumption–was also explored in Dodd et al. (2024).

**Corollary 4.8** (Extension for unknown $\bar{\zeta}$). *Assume that* $\bar{\zeta} < \infty$. *Suppose that* $\{A_k\}_{k \geq 0}$, $\{B_k\}_{k \geq 0}$ *and* $\{\tilde{B}_k\}_{k \geq 0}$ *satisfy* (10) *and* $\lim_{k \to \infty}(\tilde{B}_k - B_k) = \infty$. *Then there exists* $k_0 \geq 0$ *that satisfies* (12)*, and the convergence results in Theorem 4.5 hold true.*

**Remark 4.9.** Note that we can find proper $A_k, B_k$, and $\tilde{B}_k$ that satisfy Corollary 4.8 without knowing $\bar{\zeta}$ explicitly. For example, we may consider alternative choices $A_k = \alpha(k + 1) + 1$, $B_k = \tilde{\alpha}(k + 1)$, and $\tilde{B}_k = \alpha(k + 1)$ for Theorem 4.1, with $\tilde{\alpha} \in (0, \alpha)$, when $\bar{\zeta}$ is unknown. We provide the details in Appendix D.6.

Although Theorem 4.5 and its corollaries establish convergence rates under the conditions of Proposition 4.2, some readers may still wonder about the respective roles and the design mechanism behind $A_k$, $B_k$, and $\tilde{B}_k$. Basically, all the sequences $A_k$, $B_k$, and $\tilde{B}_k$ are carefully designed to satisfy the inequalities in (23), thereby ensuring the validity of the Lyapunov analysis in Appendix D.1, which forms the core of the convergence analysis. We provide further discussion in Appendix D.7.

One may naturally ask how the theoretical guarantees of RAdaGD depend on the curvature of the underlying manifold. To illustrate this dependence, we provide an analysis in the simplified setting where $f$ is geodesically $L$-smooth, which makes the curvature dependence of our method explicit. Consider the parameters of Theorem 4.1 with $\alpha = 1$. Under this setup, careful calculations yield

$$f(x_k) - f_\star \leq \frac{3 + \bar{\zeta}}{2(k + 2 + \bar{\zeta})} \left(1 + \frac{(2 + \bar{\zeta})^2(1 + \bar{\zeta})}{(3 + \bar{\zeta})^2}\right) L d(x_0, x_\star)^2.$$

On the other hand, for the classical RGD with a fixed stepsize (Zhang & Sra, 2016, Theorem 13), one gets

$$f(x_k) - f_\star \leq \frac{\bar{\zeta} L}{2(k - 1 + \bar{\zeta})} d(x_0, x_\star)^2, \quad \text{for all } k \geq 1.$$

When comparing this non-adaptive RGD with our method, our convergence rate includes an extra constant factor of order $\mathcal{O}(\bar{\zeta})$. This factor is harmless when the manifold has nonnegative curvature (in which case $\bar{\zeta} = 1$, and only the 17/2 constant appears), but it may become large on manifolds with strong negative curvature. At this moment, it remains an open question whether this additional factor is an inherent cost of adaptivity or merely an artifact of the analysis.

Lastly, we conclude this section with providing a remark regarding the computational complexity of RAdaGD. In general, it is difficult to provide a universal computational complexity comparison between RAdaGD and standard Riemannian gradient descent (RGD). The additional cost of RAdaGD depends on the underlying manifold, as each iteration requires evaluating two Riemannian metrics and performing one parallel transport in the computation of $L_{k+1}$. Consequently, the overall computational complexity is governed by the cost of these geometric operations. In our main application in Section 5, these additional operations have the same time complexity as the basic steps of RGD. Therefore, the asymptotic time complexity remains unchanged.

## 5 APPLICATIONS

In this section, we provide a core application of our method, Gaussian variational inference (GVI). GVI can be formulated as an optimization problem on a non-negatively curved Riemannian manifold. Additional applications under different geometries and tasks can be found in Appendix G.

The parameter choices we focus on in the experiments are as follows, which we found useful after testing various candidates:

**Lemma 5.1.** *Suppose that the manifold has non-negative curvature. Then* $A_k = 2(k + 2)^{0.1} + 2$, $B_k = 2(k + 2)^{0.1}$, *and* $\tilde{B}_k = 2(k + 2)^{0.1} + 1$ *satisfy the conditions in Corollary 4.6.*

We note that these choices achieve a non-ergodic convergence rate $f(x_k) - f(x_\star) \leq \mathcal{O}\left(1/k^{0.1}\right)$. The proof of Lemma 5.1 can be done by simply checking that (10) holds, as we provide in Appendix E.1.1. While the parameters in Theorem 4.1 may be theoretically sound, they can suffer from conservative stepsizes, as discussed in Suh & Ma (2025, § 5.3).

## 5.1 GAUSSIAN VARIATIONAL INFERENCE

The goal of GVI is to find the Gaussian distribution closest (in a Kullback–Leibler (KL) sense) to the target distribution $\pi$, which has density with respect to the Lebesgue measure given by $\pi(x) \propto e^{-V(x)}$ for some measurable function $V : \mathbb{R}^n \to \mathbb{R}$. Denoting by $\mathrm{BW}(\mathbb{R}^n)$ the space of non-singular Gaussians on $\mathbb{R}^n$, this optimization problem can be written as

$$\min_{\mu \in \mathrm{BW}(\mathbb{R}^n)} \mathcal{F}(\mu) := \underbrace{\mathbb{E}_{X \sim \mu}[V(X)]}_{:=\mathcal{V}(\mu)} + \underbrace{\mathbb{E}_{X \sim \mu}[\log \mu(X)]}_{:=\mathcal{H}(\mu)}, \tag{15}$$

for the functional $\mathcal{F} : \mathrm{BW}(\mathbb{R}^n) \to \mathbb{R}$. Here, $V$ is called the potential function (or simply the potential) of $\pi$, $\mathcal{V}$ denotes the potential energy, and $\mathcal{H}$ denotes the negative entropy. It is well known that $\mathcal{F}(\mu)$ equals $\mathrm{D}_{\mathrm{KL}}(\mu \,\|\, \pi)$ up to an additive constant. For readers unfamiliar with GVI, one may view $\mathcal{F}$ as the objective function $f$ and $\mu$ as the variable $x$ used in Section 4.

A favorable aspect of this problem is that $\mathrm{BW}(\mathbb{R}^n)$ admits a Riemannian structure, namely the *Bures–Wasserstein (BW) geometry* (Bhatia et al., 2019; Chewi et al., 2020; Altschuler et al., 2021; Han et al., 2021; Thanwerdas & Pennec, 2022a;b; Thanwerdas, 2022)[1]. Any non-singular Gaussian measure can be parameterized by its mean and covariance, *i.e.*, $\mu \simeq (m, \Sigma) \in \mathbb{R}^n \times \mathrm{SPD}(n)$, where $\mathrm{SPD}(n)$ denotes the space of positive definite matrices in $\mathbb{R}^{n \times n}$. Under the BW metric, $\mathrm{BW}(\mathbb{R}^n)$ becomes a product Riemannian manifold $\mathbb{R}^n \times \mathrm{SPD}(n)$ with *non-negative curvature* (Takatsu, 2009, Theorem 1.1). Consequently, in this setting we have $\zeta = 1$ (Takatsu, 2009, Theorem 1.1). We throughout use $\mu$ and $(m, \Sigma)$ interchangeably to denote points on $\mathrm{BW}(\mathbb{R}^n)$. Further details on the Bures-Wasserstein geometry are provided in Appendix B.2.

This geometric structure naturally suggests applying RGD methods, in particular RAdaGD, to solve (15) with objective function $f = \mathcal{F}$. To this end, we first check whether $\mathcal{F}$ is generalized geodesically convex and locally geodesically smooth, properties determined by the target distribution $\pi$ through its potential $V$. Lemma 5.2 gives a sufficient condition for $\mathcal{F}$ to be generalized geodesically convex. Although this result is well known in Wasserstein geometry (Ambrosio et al., 2008; Salim et al., 2020; Diao et al., 2023), we include a proof in Appendix E.1.2 for completeness.

**Lemma 5.2** (Generalized geodesic convexity of $\mathcal{F}$). *$\mathcal{F}$ is generalized geodesically convex on $\mathrm{BW}(\mathbb{R}^n)$ if $V$ is convex in $\mathbb{R}^n$.*

We next establish the local geodesic smoothness of $\mathcal{F}$. To the best of our knowledge, this property has not been identified in either the Riemannian optimization or Wasserstein geometry literature. The result is nontrivial, since the *geodesic incompleteness* of $\mathrm{BW}(\mathbb{R}^n)$ prevents a direct application of Theorem 3.3 (Thanwerdas & Pennec, 2022b). Nevertheless, the same conclusion holds in $\mathrm{BW}(\mathbb{R}^n)$, as shown in Proposition 5.3. The proof is given in Appendix E.1.3.

**Proposition 5.3** (Local geodesic smoothness of $\mathcal{F}$). *The following facts hold:*

   *(i) Every $C^2(\mathrm{BW}(\mathbb{R}^n))$ function is locally geodesically smooth on $\mathrm{BW}(\mathbb{R}^n)$.*

   *(ii) $\mathcal{V} \in C^\infty(\mathrm{BW}(\mathbb{R}^n))$ if the following holds for some $C, p, a > 0$ and $\beta \in [0, 2)$:*

$$|V(X)| \leq C(1 + \|X\|^p) \exp(a \|X\|^\beta). \tag{16}$$

   *(iii) $\mathcal{H} \in C^\infty(\mathrm{BW}(\mathbb{R}^n))$.*

   *(iv) Accordingly, $\mathcal{F}$ locally geodesically smooth on $\mathrm{BW}(\mathbb{R}^n)$ if (16) holds.*

Lemma 5.2 and Proposition 5.3 together imply that $\mathcal{F}$ is locally geodesically smooth and generalized geodesically convex whenever the target potential $V$ is convex, *i.e.*, $\pi$ is log-concave, and satisfies

---

[1]There exists another natural Riemannian metric on the space of Gaussian measures, namely the Fisher-Rao metric. However, for GVI, the Bures-Wasserstein geometry is theoretically preferable. See Appendix G.1.1 for a detailed comparison.

the rather mild growth condition. Many statistical models involve such potential $V$, ranging from classical $L$-smooth potentials like Gaussian and logistic regression to more complex non-$L$-smooth potentials. In particular, Lemma E.2 highlights examples including generalized linear models such as Poisson regression and Gamma regression with log-link (McCullagh & Nelder, 1989).

With this background, we apply the RAdaGD algorithm to solve GVI (15) when $V$ is convex and satisfies (16). The extension of RAdaGD and Corollary 4.6 to GVI problem is nontrivial, due to the geodesic incompleteness of $\mathrm{BW}(\mathbb{R}^n)$. Nevertheless, with minor algorithmic adjustments and an additional technical assumption, the same convergence guarantee holds, as shown in Corollary 5.4. The modified algorithm BWAdaGVI and its proof are provided in Appendix E.1.4.

**Corollary 5.4** (Adaptive Bures-Wasserstein gradient descent). *Consider the optimization* (15) *where $V$ is convex and satisfies* (16). *Choose* $\{A_k\}_{k\geq 0}$, $\{B_k\}_{k\geq 0}$, *and* $\{\tilde{B}_k\}_{k\geq 0}$ *to satisfy the assumptions in Corollary 4.6 with $\bar{\zeta} = 1$. Let $\delta \in (0,1)$ be an arbitrarily small number. Run RAdaGD to solve* (15), *with an additional minimization step on the step size (which corresponds to BWAdaGVI):*

$$s_{k+1} \leftarrow \min\left\{s_{k+1}, \frac{1-\delta}{\max_i \left|\lambda_i(\mathbb{E}_{X\sim\mu_{k+1}}[\nabla^2 V(X)] - \Sigma_{k+1}^{-1})\right|}\right\}. \tag{17}$$

*If there exists $\epsilon > 0$ such that $\lambda_{\min}(\Sigma_k) \geq \epsilon$ for all $k \geq 0$, then Corollary 4.6 holds with $N = M = \mathrm{BW}(\mathbb{R}^n)$ and $r \leq 1 - \delta$.*

**Remark 5.5.** The additional minimization of the step size in (17) and the uniform eigenvalue lower bound in Corollary 5.4 are introduced to address geodesic incompleteness. Bounds on extreme eigenvalues commonly appear in the Wasserstein geometry literature; see discussions on Xu & Li (2025, Assumption 2). Particularly, Lambert et al. (2022, Lemma 6) enforced such condition via maximum eigenvalue clipping under strongly convex $V$, while we did not pursue this direction. We also note theoretically ideal cases where the condition holds (Appendix E.1.4). Empirically, the assumption appears benign, as eigenvalues of the iterates stay bounded away from zero (Figure 1).

The ingredients of RAdaGD for $\mathrm{BW}(\mathbb{R}^n)$ space–the exponential map (Definition B.12), BW-gradients (Definition B.13), and parallel transports (Definition B.15)–are explicitly known, ensuring tractability. Both the exponential map and parallel transport are fully computable. A technical consideration is that $\mathcal{F}$ and $\mathrm{Grad}_{\mathrm{BW}}\mathcal{F}$ involve expectations (Remark B.14), which may lack closed-form expressions. However, many practical statistical problems admit closed-form BW-gradients (Lemma E.2); when they do not, stochastic approximations can be used (Ranganath et al., 2014; Kim et al., 2023; Lambert et al., 2022; Diao et al., 2023; Luu et al., 2025), although we defer extending our theory to this generality to future work. Thus, the algorithm remains tractable for a wide range of applications.

Corollary 5.4 improves upon previous GVI algorithms by replacing the global $L$-smoothness requirement on $V$ with a weaker growth condition. This relaxation is crucial, as existing methods establish convergence only under the $L$-smoothness assumption (Domke, 2020; Lambert et al., 2022; Diao et al., 2023; Kim et al., 2023), whereas many practical problems lack global smoothness. Thus, our algorithm is, to the best of our knowledge, the first to provide provable convergence guarantees for GVI in this broader setting under mild assumptions.

Bayesian Poisson regression is a representative case where global $L$-smoothness fails. For response $Y_i \in \mathbb{N}$ and predictor $X_i \in \mathbb{R}^n$ ($i = 1, \ldots, \ell$), a Poisson regression model assumes $Y_i \mid X_i \sim \mathrm{Poisson}(\exp(X_i^T\theta))$ independently across $i$. Assuming an improper prior on $\theta$, the potential $V$ in (15) takes the form

$$V(\theta) = \sum_{i=1}^{\ell} \left(\exp(X_i^T\theta) - Y_i X_i^T\theta\right).$$

Because of its exponential dependence on $\theta$, this $V$ is not globally $L$-smooth. However, by Lemma E.2, it is convex, satisfies the growth condition (16), and both $\mathcal{V}$ and its BW-gradient admit analytic forms. Thus, RAdaGD is theoretically guaranteed and fully tractable for this problem.

We conduct numerical experiments on Poisson regression GVI, comparing our method with the state-of-the-art algorithm of Diao et al. (2023) (denoted FBGVI). Since FBGVI requires global $L$-smoothness of $V$, it lacks theoretical guarantees for this task. For our algorithm, we chose $A_k, B_k$, and $\tilde{B}_k$ as in Lemma 5.1, with $\ell = 50$ and $n = 25$, using independently generated $X_i \sim N(0, I_n)$

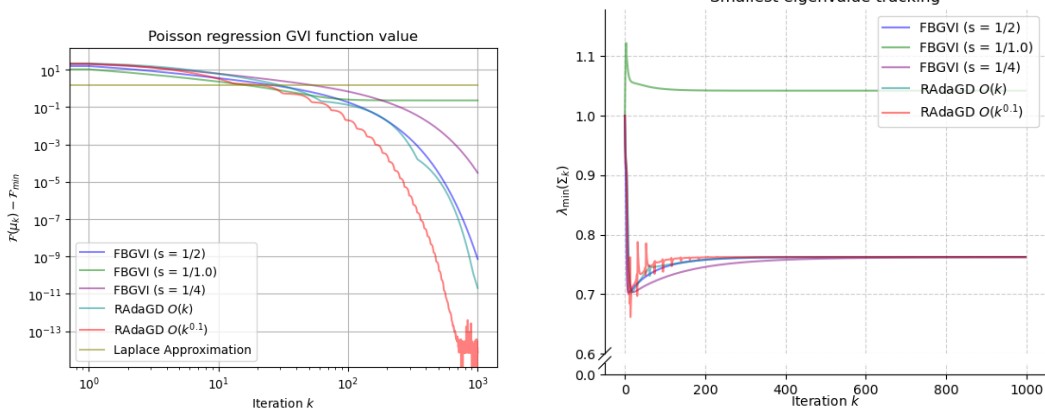

Figure 1: Comparison between FBGVI (Diao et al., 2023) and our methods on Poisson regression target $\pi$ with $\ell = 50, n = 25$. **Left**: $\mathcal{F}(\mu_k) - \mathcal{F}_{\min}$, where $\mathcal{F}_{\min}$ is the minimum value among all experiments. **Right**: Minimum eigenvalues $\lambda_{\min}(\Sigma_k)$ over the iteration.

and $Y_i \sim \text{Poisson}(\exp(\theta^T X_i))$ for $i = 1, \ldots, \ell$. Figure 1 shows that our method outperforms FBGVI across step sizes $s = 1, 1/2, 1/4$.

Our algorithm extends beyond the preceding example. Within GVI, the growth condition in Proposition 5.3 covers many practical, non-$L$-smooth potentials (see Lemma E.2), where our method remains applicable and effective. Beyond GVI, there are tasks under other geometries that either fully satisfy our assumptions (Appendix G.3.1) or violate them but still see strong empirical performance (Appendix G.2, G.3). Additional applications, including further GVI tasks and broader Riemannian optimization problems, are presented in Appendix G.

Codes for our experiments can be found at https://github.com/wldyddl5510/RAdaGD.

## 6 CONCLUSION

In this work, we propose RAdaGD, a family of line-search-free adaptive gradient descent methods on Riemannian manifolds. Instances of RAdaGD achieve a non-ergodic convergence rate $f(x_k) - f(x_\star) \leq \mathcal{O}(1/k)$ and extend to possibly negatively curved manifolds, which is the first of its kind. We establish convergence guarantees for functions that are locally geodesically smooth and generalized geodesically convex. Local geodesic smoothness broadens applicability beyond the standard $L$-smooth assumption, yielding the first algorithm with a theoretical guarantee for GVI under locally smooth targets (BWAdaGVI).

We believe this work opens several directions for future research. Extending Euclidean adaptive acceleration methods (Li & Lan, 2025; Suh & Ma, 2025) or non-adaptive accelerated Riemannian methods (Zhang & Sra, 2018; Ahn & Sra, 2020; Kim & Yang, 2022) to develop Nesterov-type acceleration on manifolds would be natural next steps. A proximal variant of our method could further broaden its applicability (Chen et al., 2020; Huang & Wei, 2022; Wang et al., 2022; Diao et al., 2023). Another direction is to clarify the fundamental cost of adaptivity. As shown in Section 4, our method incurs an additional $\mathcal{O}(\bar{\zeta})$ factor in the convergence rate compared to non-adaptive RGD. Whether this factor is intrinsic or merely an artifact of the analysis remains open. Extensions incorporating retraction maps or inexact parallel transport, which are relevant for practice (e.g., see Appendix G.1.1), are also of interest. Finally, developing stochastic variants with controlled error in estimating $L_{k+1}$ would be meaningful (Bonnabel, 2013; Ranganath et al., 2014; Zhang et al., 2016; Kim et al., 2023; Diao et al., 2023; Luu et al., 2025).

ACKNOWLEDGMENTS

Authors thank anonymous reviewers for their insightful feedback during the review process. Research of Jaewook J. Suh was supported in part by Basic Science Research Program through the National Research Foundation of Korea (NRF) funded by the Ministry of Education (RS-2024-00410486). Research of Shiqian Ma was supported in part by NSF grants CCF-2311275 and ECCS-2326591.

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

APPENDIX

# A  PRIOR WORK AND MOTIVATION

## A.1  RIEMANNIAN OPTIMIZATION

Riemannian optimization has become an essential framework for solving constrained problems where the solution space admits a smooth manifold structure. Foundational works by Udrişte (1994), Absil et al. (2008), and Bačák (2014a) established the theoretical basis for this field by introducing key geometric concepts for optimization purpose. These geometric foundations enable the generalization of classical optimization methods such as gradient descent, conjugate gradients, and Newton's method to the Riemannian setting.

Early research in this area primarily established asymptotic convergence guarantees. More recently, attention has turned toward non-asymptotic convergence analyses. A particular milestone was achieved by Zhang & Sra (2016), who derived a non-asymptotic convergence rate for Riemannian gradient descent (RGD), obtaining $f(x_k) - f(x_\star) \leq O(1/k)$ under the assumptions of geodesic $L$-smoothness and convexity, using a constant step size $s_k = 1/L$. Further studies of RGD and its variants under a generalized geodesic convex setup are presented in Chewi et al. (2020); Altschuler et al. (2021).

Subsequent efforts extended these results in several directions. Several authors developed accelerated Riemannian optimization methods for geodesically (strongly) convex functions (Zhang & Sra, 2018; Ahn & Sra, 2020; Alimisis et al., 2020; 2021; Kim & Yang, 2022). However, full acceleration in the sense of Nesterov has been shown to be impossible on Riemannian manifolds with negative curvature (Hamilton & Moitra, 2021; Criscitiello & Boumal, 2022; 2023b).

Beyond the smooth setting, nonsmooth Riemannian optimization has also been explored. Ferreira & Oliveira (2002) proved asymptotic convergence of Riemannian proximal point methods, with non-asymptotic $O(1/k)$ rates later provided by Bento et al. (2017). Chen et al. (2020) first developed proximal gradient methods specifically for optimization over the Stiefel manifold with composite objectives.

There are also some results on stochastic optimization over Riemannian manifold. In particular, asymptotic convergence of Riemannian stochastic gradient descent was first shown by Bonnabel (2013), with non-asymptotic rates for geodesically convex functions later established by Zhang & Sra (2016). Subsequent work introduced variance-reduced variants such as Riemannian SVRG and SAGA (Zhang et al., 2016; Sato et al., 2019).

A persistent challenge across these works is the reliance on strong assumptions that limit practical applicability. Most convergence analyses that base on geodesic $L$-smoothness and convexity require: 1) global $L$-smoothness on a known set; 2) prior knowledge of the smoothness constant $L$; and 3) boundedness of iterates, either assumed a priori or enforced through projections. Such restriction is crucial on Riemannian manifolds; as noted in Alimisis et al. (2020, §4), there is some sort of duality between convexity and smoothness with respect to the curvature of the manifold. For instance, on non-positively curved manifolds, even a simple function $x \mapsto d^2(x, p)$ is not geodesically $L$-smooth. This simple instance implies that the condition 1, global $L$-smoothness, may become a very restrictive condition on certain manifold, conjecturally manifolds with non-positive curvature.

Recent years have witnessed efforts to relax these restrictive assumptions. For instance, Martínez-Rubio et al. (2024) and Martínez-Rubio & Pokutta (2023) removed the iterate boundedness assumption for RGD and an accelerated variant, respectively. Nevertheless, these methods still require knowledge of the geodesic smoothness parameter $L$ over a predetermined set, limiting their applicability in settings where such global information is unavailable.

Our work represents a significant advance beyond the current state of the art by addressing these persistent limitations simultaneously. Unlike most existing methods, our approach eliminates the need for prior knowledge of the smoothness parameter $L$ while maintaining non-vanishing stepsize and achieving $O(1/k)$ non-ergodic convergence rates. This combination of features has not been attained by any prior stochastic or deterministic method in the literature. Theoretically, our method requires only local geodesic smoothness assumptions without needing to assume iterate boundedness. While we employ a stronger generalized geodesic convexity condition, this trade-off enables

substantially weaker smoothness requirements that are satisfied by $C^2$ functions and thus widely applicable to practical problems like Gaussian variational inference where global $L$-smoothness does not hold. We discuss this topic in more detail in the next subsection.

## A.2    ADAPTIVE ALGORITHMS

The textbook step size selection of gradient descent

$$x_{k+1} = x_k - s_k \nabla f(x_k)$$

is $s_k = \frac{1}{L}$ for $L$-smooth convex functions, achieving a convergence rate of $f(x_k) - f(x_\star) \le \mathcal{O}(1/k)$. There are several limitations of this step size selection, which are:

(i)   The assumption that the function class requires global smoothness $L$ for $\nabla f$ can be restrictive.

(ii)  Even when $f$ is $L$-smooth, prior knowledge of $L$ may be difficult to obtain.

(iii) $L$ is a global parameter, so $\frac{1}{L}$ can be too conservative a choice.

One classical approach that addressing this issue is line search (Goldstein, 1962; Larry Armijo, 1966), which has been widely studied in the literature, but its computational cost can be considered a limitation. For a more detailed discussion, please refer to (Malitsky & Mishchenko, 2024, § 1).

The development of an adaptive step size method that is

(i)   line-search-free,

(ii)  parameter-free; adapts the approximated local smoothness parameter to choose the step size,

(iii) guarantees theoretical convergence, and

(iv)  for locally smooth convex functions

was first proposed by Malitsky & Mishchenko (2020). To clarify the terminology, we note that from now on when we refer to 'adaptive' we will mean methods that satisfy these four properties.[2] The method AdGD, introduced in this paper, selects the step size based on an approximation of the local smoothness parameter using the last two iterates, defined as $L_k = \frac{\|\nabla f(x_k) - \nabla f(x_{k-1})\|}{\|x_k - x_{k-1}\|}$. Even though it achieves an ergodic convergence rate $f(\hat{x}_k) - f(x_\star) \le \mathcal{O}(1/k)$, where $\hat{x}_k$ is a weighted average of $x_1, \ldots, x_k$, AdGD was the first adaptive method that satisfies all four properties mentioned above for convex function minimization.

Initiated by Malitsky & Mishchenko (2020), various follow-up works studying adaptive methods have since emerged; here, we introduce a few. Oikonomidis et al. (2024) introduced an adaptive method in the prox-grad setup, named AdaPDM. They leveraged two pieces of local information, $\ell_k = \frac{\langle \nabla f(x_{k-1}) - \nabla f(x_k), x_{k-1} - x_k \rangle}{\|x_{k-1} - x_k\|}$ and $c_k = \frac{\|\nabla f(x_{k-1}) - \nabla f(x_k)\|}{\langle \nabla f(x_{k-1}) - \nabla f(x_k), x_{k-1} - x_k \rangle}$, while choosing the step size. They also extended their method to primal-dual setting and introduced AdaPGM. Latafat et al. (2024a) generalized AdaPGM to a family of algorithms AdaPG$^{q,r}$. Malitsky & Mishchenko (2024) also introduced an adaptive method for the prox-gradient setup, called AdProxGD, by sharpening and extending their previous method, AdGD. Zhou et al. (2025) introduced an adaptive method, named AdaBB, based on the Barzilai-Borwein method Barzilai & Borwein (1988). They chose the step size based on the approximation $L_k = \frac{\langle \nabla f(x^k) - \nabla f(x^{k-1}), x^k - x^{k-1} \rangle}{\|\nabla f(x^k) - \nabla f(x^{k-1})\|^2}$, and achieved a larger lower bound on the step size: $\sum_{i=1}^k s_i \ge \frac{k-2+\sqrt{2}}{L} = \Omega(k)$, closing an open problem raised by Malitsky & Mishchenko (2024). Furthermore, AdaBB also considers extensions to the prox-gradient setup and to locally strongly convex functions. Nevertheless, achieving a non-ergodic convergence rate remains a challenge for all of these adaptive gradient descent methods, as noted in Malitsky & Mishchenko (2024).

The first adaptive method with non-ergodic (or last-iterate) convergence was introduced by Li & Lan (2025) and is named AC-FGM. They choose the step size based on a novel approximation of the smoothness parameter,

$$L_k = \frac{-\frac{1}{2} \|\nabla f(x_k) - \nabla f(x_{k-1})\|^2}{f(x_k) - f(x_{k-1}) + \langle \nabla f(x_k), x_{k-1} - x_k \rangle}. \tag{18}$$

---

[2]This clarification is based on (Malitsky & Mishchenko, 2024, § 1).

Surprisingly, they achieved not only a non-ergodic rate but also an accelerated rate, $f(x_k) - f(x_\star) \leq \mathcal{O}\left(1/k^2\right)$. Leveraging the same approximation of the smoothness parameter, Suh & Ma (2025) sharpened the constant with their new method, AdaNAG, by providing a Lyapunov analysis. Inspired by the Lyapunov analysis, Suh & Ma (2025) also introduced AdaGD, which was the first adaptive gradient descent method with a non-ergodic convergence rate, $f(x_k) - f(x_\star) \leq \mathcal{O}(1/k)$, without momentum. As a side note, while momentum may not pose significant difficulties in the Euclidean setup, it can become a bottleneck when extending to Riemannian setup (Criscitiello & Boumal, 2022; 2023b).

### A.2.1 ADAPTIVE ALGORITHMS FOR RIEMANNIAN OPTIMIZATION AND LOCAL SMOOTHNESS

Similar to Euclidean optimization, RGD $x_{k+1} = \exp_{x_k}(-s_k \operatorname{Grad} f(x_k))$ with $s_k = \frac{1}{L}$ is a popular method for geodesically convex and geodesically $L$-smooth functions. While the limitations discussed at the beginning of this subsection also apply to Riemannian optimization, developing adaptive methods can also be beneficial.

However, compared to Euclidean optimization, extending to locally smooth functions can be even more critical, since the conventional geodesic $L$-smoothness assumptions of RGD exclude a number of important applications in Riemannian optimization. For instance, as noted throughout the paper, even the simple function $x \mapsto d^2(x, p)$ fails to be geodesically $L$-smooth on a Hadamard manifold (Criscitiello & Kim, 2025). Consequently, many applications have been considered without fully established theoretical guarantees, even when certain problem parameters are known.

Despite this demand, studies of adaptive methods that satisfy the four properties discussed earlier are limited,[3] particularly regarding the extension to locally smooth functions. This motivated us to develop a Riemannian adaptive gradient descent method inspired by AdaGD (Suh & Ma, 2025). Recalling that AdaGD leverages the approximation (18) to achieve a non-ergodic convergence rate, our method RAdaGD employs a similar approximation (5). Since the well-definedness of (5) heavily relies on the inequality (4), we assume generalized geodesic convexity (Definition 3.2), which is often assumed in the study of RGD, as mentioned in (Altschuler et al., 2021, § A.2). In the meantime, RAdaGD benefits from the extension to local smoothness, as $C^2$ functions satisfy this property (Theorem 3.3), enabling a wider range of interesting applications. In particular, RAdaGD enables applications to the Gaussian variance inference problem with a target log-density that lacks $L$-smoothness, for which we provide more context in the next subsection.

### A.3 VARIATIONAL INFERENCE

Variational inference (VI) (Peterson & Hartman, 1989; Jordan et al., 1999; Wainwright & Jordan, 2008; Blei et al., 2017) is a task to approximate a target distribution $\pi$ to some family of distributions $\mathcal{M}$, called variational family. Formally, one can write the variational inference task as follows:

$$\text{Find } \mu_* := \arg\min_{\mu \in \mathcal{M}} D(\mu \| \pi) \tag{19}$$

for some discrepancy metric $D$ between probability measures. The need of such approximation roots from the sampling; often, $\pi$ is a distribution whose sample is hard to obtain. Natural case is when $\pi$ is a posterior distribution from Bayesian inference. In principle, one can obtain the exact samples of $\pi$ from Markov Chain Monte Carlo (MCMC) methods. However, MCMC method is often require heavy computational costs, particularly in high-dimensional setting. VI detours this problem by allowing reduced emphasis on sample accuracy in return for computational gains. Specifically, one chooses $\mathcal{M}$ in (19) to be easy-to-sample distributions. If one obtains $\mu_*$ by solving (19), one can use samples from $\mu_*$ as *approximated* samples from $\pi$. In this regard, VI provides a fast and scalable alternative to MCMC.

---

[3]While (Cai et al., 2025; Shi et al., 2025) also consider adaptive methods for Riemannian optimization, their problem setups differ from ours (bilevel and strongly monotone settings, respectively), and they only partially satisfies the discussed properties. We also regard methods with nonincreasing step sizes, such as Dodd et al. (2024), as a different line of work. This is because the step size may not capture local smoothness information, as discussed in (Malitsky & Mishchenko, 2024, § 1), and thus violates (ii). (Yagishita & Ito, 2025) studies nonconvex adaptive methods and also introduces a Riemannian variant, but their stepsize is nonincreasing and does not incorporate curvature-aware scaling.

In Equation (19), there are two choices that the user have to make: a variational family $\mathcal{M}$ and a discrepancy metric $D$. The choices of these components determine the nature of the optimization problem (19). For the discrepancy metric $D$, the most canonical choice is KL-divergence. In this setting, if $\pi$ has a density of the form $\pi(x) \propto \exp(-V(x))$ for some measurable function $V : \mathbb{R}^n \to \mathbb{R}$, one can show that $\mathrm{D}_{\mathrm{KL}}(\mu\|\pi) = \mathcal{F}(\mu) + \mathrm{const}$ where $\mathcal{F}$ is in (15). Thus, solving the optimization problem (15) is equivalent to performing variational inference with respect to KL-divergence. For a variational family $\mathcal{M}$, two notable cases are $\mathcal{M} = \mathrm{BW}(\mathbb{R}^n)$, the family of Gaussian measures, and $\mathcal{M} = \mathcal{P}(\mathbb{R})^{\otimes n}$, the family of product measures. These correspond to Gaussian variational inference (GVI) and mean-field variational inference (MFVI), respectively.

The literature on variational inference (VI) can be broadly divided into two strands: analyzing its statistical approximation properties, and developing algorithms with provable guarantees. In this work, we focus on its algorithmic aspects. For statistical approximation, interested readers may refer to Han & Yang (2019) for MFVI and Katsevich & Rigollet (2024); Margossian & Saul (2025) for GVI.

For the algorithmic aspects, the study of algorithms for MFVI are mainly based on coordinate ascent method, called CAVI (Blei et al., 2017, §2.4). The algorithmic convergence guarantees for MFVI is an active area of research (Bhattacharya et al., 2025; Arnese & Lacker, 2024; Lavenant & Zanella, 2024; Jiang et al., 2025; Kim et al., 2025b). For GVI, one of the most direct and standard approaches to solving (15) is to view it as an optimization problem over the parameter space, treating the parameterization as a subset of $\mathbb{R}^d$ (Paisley et al., 2012; Ranganath et al., 2014; Domke, 2020; Quiroz et al., 2023; Kim et al., 2023). In contrast, recent developments emphasize respecting the natural geometric structure of the $\mathrm{BW}(\mathbb{R}^n)$ space. In particular, beginning with Lambert et al. (2022), works such as Diao et al. (2023); Luu et al. (2025) have studied solving (15) with respect to the Bures–Wasserstein geometry. Our work aligns with this direction.

## B  PRELIMINARIES

### B.1  RIEMANNIAN GEOMETRY

In this section, we review the basics of Riemannian geometry, and introduce the important functional properties. For the full detail, we refer to the classical textbooks, such as Lee (2012; 2018); Boumal (2023).

A $n$-dimensional differentiable manifold $M$ is a topological space that is locally diffeomorphic to an open set in $\mathbb{R}^n$. At a point $x \in M$, a tangent vector is a velocity vector of a smooth curve on $M$ that passes through $x$. A tangent space $T_x M$ is the vector space consisting all such tangent vectors. The space $TM := \coprod_{x \in M} T_x M$ is called the tangent bundle.

Often, this tangent space $T_p M$ is conveniently expressed in the form of the vector field, which takes a point in a manifold as an input and returns a tangent space vector at that point. A map $X : C^\infty(M) \to C^\infty(M)$ is called a smooth vector field if it is a derivation, *i.e.*, $X$ satisfies

$$X.(fg) = X.(f)g(\cdot) + f(\cdot)X.(g).$$

Here $\cdot \in M$ is the input of the function. A set of smooth vector fields on $M$ is denoted by $\mathfrak{X}(M)$. As the name derivation indicates, one can think of the vector field as a directional derivative along the direction of the vector field.

A Riemannian manifold $(M, \langle \cdot, \cdot \rangle_x)$ is a differentiable manifold equipped with a Riemannian metric $\langle \cdot, \cdot \rangle_x$, which is an inner product on each tangent space $T_x M$ and a smooth map on $x \in M$. A Riemannian manifold is a manifold equipped with an inner product for each tangent space, called a Riemannian metric. Then, the notion of norm naturally extends as $\|v\|_x := \sqrt{\langle v, v \rangle_x}$. For $x, y \in M$, the distance $d(x, y)$ is the infimum of the length of all piecewise continuously differentiable curves from $x$ to $y$.

For $(x, v) \in TM$, a smooth curve $\gamma_v : [0, 1] \to M$ with $\gamma_v(0) = x$ and $\gamma_v'(0) = v$ is called a geodesic if it is a critical point of the energy functional defined by

$$E(\gamma) = \frac{1}{2} \int_0^1 \|\gamma'(t)\|^2 \, dt.$$

For instance, a straight line $\gamma_v(t) = x + tv$ in Euclidean space is a geodesic. If a geodesic such that $\gamma(0) = x$ and $\gamma(1) = y$ satisfies $d(x, y) = \int_0^1 \|\gamma'(t)\| \, dt$, then we call this geodesic the minimizing geodesic.

Geodesics induce the notion of exponential map and logarithmic map. The exponential map $\exp_x : T_xM \to M$ is the map defined by $\exp_x(v) = \gamma_v(1)$. The exponential map transports the point $x$ in the direction of the tangent vector $v$ following the geodesic $\gamma_v$. It is well-known that the exponential map is local diffeomorphism in some neighborhood of $0 \in T_xM$, denoted by $U$. Thus, the exponential map allows the inverse map on $U$, which is called the logarithmic map $\log_x : \exp_x(U) \to T_xM$. Note that the exponential map and logarithmic map are locally well-defined, but may not be globally. The manifold $M$ is called complete if $\exp_x$ is defined on entire $T_xM$.

Geodesics also provide a way to transport vectors from one tangent space to another, called parallel transport. A parallel transport $\Gamma(\gamma)_{t_0}^{t_1} : T_{\gamma(t_0)}M \to T_{\gamma(t_1)}M$ is a way to transport a tangent vector along the curve $\gamma$ parallelly. Given a geodesic, the parallel transport need to satisfy the following property:

$$\Gamma(\gamma)_s^t \gamma'(s) = \gamma(t).$$

It is known that the linear operator $\Gamma$ with the above property is unique. Hence, the notion of geodesic uniquely determines the parallel transport. If $\gamma$ is a geodesic curve such that $\gamma(0) = x, \gamma(1) = y$, then we simply denote $\Gamma(\gamma)_0^1$ as $\Gamma_x^y$, a (geodesic) parallel transport from $T_xM$ to $T_yM$.

Geodesics and the corresponding parallel transports define an operator called Levi-Civita connection or covariant derivative, which moves a tangent vector to another tangent vector along the certain direction. Specificaly, a covariant derivative is an operator $\nabla. : \mathfrak{X}(M) \times \mathfrak{X}(M) \to \mathfrak{X}(M)$ defined by

$$\nabla_{\gamma'(t)} X(\gamma(t)) := \lim_{h \to 0} \frac{\Gamma(\gamma)_{t+h}^t X(\gamma(t+h)) - X(\gamma(t))}{h}.$$

Under Levi-Civita connection, geodesics have *zero-acceleration*. Meaning, the following equation holds if and only if $\gamma$ is a geodesic:

$$\nabla_{\gamma'(t)} \gamma'(t) = 0.$$

In this regard, often geodesics are denoted by *constant speed* geodesic.

The last geometric concept induced from the covariant derivative is curvature. The Riemannian curvature tensor $R(\cdot, \cdot). : \mathfrak{X}(M) \times \mathfrak{X}(M) \times \mathfrak{X}(M) \to \mathfrak{X}(M)$ is defined by the following formula:

$$R(X, Y)Z = \nabla_X \nabla_Y Z - \nabla_Y \nabla_X Z - \nabla_{[X,Y]} Z$$

where $[\cdot, \cdot]$ denotes a Lie bracket.

The key geometric quantity in our analysis is sectional curvature, which generalizes Gaussian curvature in a 2-dimensional surface. For $p \in M$, denote $\Sigma_p$ a set of two-dimensional subspaces in $T_pM$. For $\sigma_p \in \Sigma_p$, the sectional curvature $K : \Sigma_p \to \mathbb{R}$ is defined by the following formula:

$$K(\sigma_p) = \langle R(u, v)v, u \rangle_p$$

where $\{u, v\}$ is an orthonormal basis of $\sigma_p$.

Lastly, we introduce the notion of Riemannian gradient and geodesic convexity. Given a differentiable function $f : M \to \mathbb{R}$, the Riemannian gradient of $f$ at $x$, denoted by $\operatorname{Grad} f(x) \in T_xM$ is the tangent vector satisfying the following property:

$$\langle \operatorname{Grad} f(x), v \rangle_x = df_x(v) \quad \forall v \in T_xM.$$

Here, $df_x$ is a differential of $f$ at $x$. We say $N \subseteq M$ is a geodesically convex subset of $M$ if for all $x, y \in N$ there exists a geodesic $\gamma$ such that $\gamma(0) = x, \gamma(1) = y$, and $\gamma(t) \in N$ for all $t \in [0, 1]$. We say a differentiable function $f : N \to \mathbb{R}$ is geodesically convex if for all $x, y \in N$,

$$f(y) \geq f(x) + \langle \operatorname{Grad} f(x), \log_x y \rangle_x.$$

We note that the choice of geodesic, parallel transport, and covariant derivative are mutually equivalent: once one of these structures is specified, the other two are uniquely determined. In this work, we adopt the perspective of starting with geodesics, defined as critical points of the energy functional, which in turn uniquely determine the associated parallel transport and the Levi–Civita connection. This approach follows Spivak (1979), whereas other texts, such as Lee (2018), take the alternative route of beginning with a connection and subsequently developing the other structures.

## B.2 BURES-WASSERSTEIN GEOMETRY

In this appendix, we briefly introduce Bures-Wasserstein space $\mathrm{BW}(\mathbb{R}^n)$, a space of Gaussian measures equipped with $W_2$ metric. This space is known to be a Riemannian manifold (Modin, 2017; Bhatia et al., 2019; Han et al., 2021; Thanwerdas, 2022).

One of the most intriguing aspects of this space is that it admits two equivalent geometric formulations: the optimal transport & Otto calculus viewpoint and the Riemannian geometric viewpoint. While both lead to the same underlying geometry, they offer distinct perspectives. Depending on the problem at hand, one may choose the Otto calculus formulation or the Riemannian formulation, as each has its own advantages in different contexts.

In terms of the analyses of gradient methods (including RAdaGD or precisely BWAdaGVI), it is more common to take Otto calculus based approaches (Jordan et al., 1998; Salim et al., 2020; Lambert et al., 2022; Diao et al., 2023; Park et al., 2025). In this regard, we mainly provide the Otto calculus formulation of $\mathrm{BW}(\mathbb{R}^n)$, and briefly mention the equivalence between Riemannian formulation. Main takeaways of this appendix is to provide the detail on how one can implement ingredients in RAdaGD (exponential map, parallel transport, BW-gradients) under $\mathrm{BW}(\mathbb{R}^n)$ geometry.

We list the results mainly without providing the full derivation. For detail, we refer to Ambrosio et al. (2008); Villani (2008); Santambrogio (2015); Chewi (2024) for general Otto calculus, and Takatsu (2009); Altschuler et al. (2021); Lambert et al. (2022); Diao et al. (2023); Park et al. (2025) for Otto calculus on $\mathrm{BW}(\mathbb{R}^n)$.

### B.2.1 OTTO CALCULUS

We first begin with the general Otto calculus on the space of probability measures, and then focus on $\mathrm{BW}(\mathbb{R}^n)$, which allows the finer analyses. We denote $\mathcal{P}_p(\mathbb{R}^n)$ to be the space of probability measures on $\mathbb{R}^n$) which have the finite $p$th moment, $\mathcal{P}_{2,ac}(\mathbb{R}^n)$ to be the space of probability measures on $\mathbb{R}^n$, which are absolutely continuous with respect to Lebesgue measure and have the finite second moment, and $\mathcal{L}^2(\mu) = \left\{ f : \mathbb{R}^n \to \mathbb{R}^n \mid \int \|f(x)\|^2 \, d\mu(x) < \infty \right\}$. For $\mu, \nu \in \mathcal{P}_p(\mathbb{R}^n)$, let $\Gamma(\mu, \nu)$ be a set of couplings of $\mu$ and $\nu$. Wasserstein distance between $\mu$ and $\nu$ is defined as follows.

**Definition B.1** (Wasserstein metric). *Let $\mu, \nu \in \mathcal{P}_p(\mathbb{R}^n)$. Denote $\Gamma(\mu, \nu)$ to be a set of coupling measures of $\mu$ and $\nu$. $p$-Wasserstein distance between $\mu$ and $\nu$ is defined as follows:*

$$W_p^p(\mu, \nu) := \inf_{\gamma \in \Gamma(\mu, \nu)} \mathbb{E}_{(x,y) \sim \gamma} \left[ \|x - y\|^p \right].$$

This is known to be a well-defined metric, and the metric space $(\mathcal{P}_p(\mathbb{R}^n), W_p)$ is called $p$-Wasserstein space.

Brenier (1991); Jordan et al. (1998); Otto (2001) showed $(\mathcal{P}_{2,ac}(\mathbb{R}^n), W_2)$ endows a richer geometric properties than mere $\mathcal{P}_p(\mathbb{R}^n)$. Specifically, while $(\mathcal{P}_{2,ac}(\mathbb{R}^n), W_2)$ is not precisely a Riemannian manifold, its geometry is almost same to the non-negatively curved Riemannian manifold. The following theorem serves as the key ingredient for such observation (Brenier, 1991):

**Theorem B.2** (Brenier Theorem). *If $\mu, \nu \in \mathcal{P}_{2,ac}(\mathbb{R}^n)$, then*

$$W_2^2(\mu, \nu) = \min_{T \in \mathcal{L}^2(\mu) \, s.t. \, T_{\#\mu} = \nu} \mathbb{E}_{x \sim \mu} \left[ \|T(x) - x\|^2 \right] = \min_{T \in \mathcal{L}^2(\mu) \, s.t. \, T_{\#\mu} = \nu} \|T - id\|_{\mathcal{L}^2(\mu)}^2.$$

*Denote the minimizer as $T_{\mu,\nu}$. Then $T_{\mu,\nu}$ is unique $\mu$-a.e. and is a gradient of some convex function $\phi$ on $\mathbb{R}^n$ $\mu$-a.e. Furthermore, $T_{\mu,\nu} \circ T_{\nu,\mu} = id$. The minima $T_{\mu,\nu}$ is called the optimal transport map from $\mu$ to $\nu$.*

Theorem B.2 gives a notion of tangent direction at $\mu$ (Chewi, 2024, §1.3).

**Definition B.3** (Riemannian metric in 2-Wasserstein space). *For $\mu \in \mathcal{P}_{2,ac}(\mathbb{R}^n)$, a tangent space $T_\mu \mathcal{P}_{2,ac}(\mathbb{R}^n) \subset \mathcal{L}^2(\mu)$ is defined by*

$$T_\mu \mathcal{P}_{2,ac}(\mathbb{R}^n) = \overline{\{\nabla \psi \mid \psi \in C_c^\infty(\mathbb{R}^n)\}}^{\mathcal{L}^2(\mu)} = \overline{\{\lambda(T - id) \mid \lambda > 0, T \text{ is an optimal transport map}\}}^{\mathcal{L}^2(\mu)}.$$

*Here, $C_c^\infty(\mathbb{R}^n)$ is a set of compactly supported smooth functions on $\mathbb{R}^n$. The Riemannian metric here is $\mathcal{L}^2(\mu)$-inner product, i.e., $\langle v, w \rangle_\mu = \mathbb{E}_{x \sim \mu}[\langle v(x), w(x) \rangle]$.*

One can naturally define a geodesic curve in $(\mathcal{P}_{2,ac}(\mathbb{R}^n), W_2)$, by pushforwarding the interpolation between particles to the measure space.

**Definition B.4** (Geodesic in Wasserstein space). *A geodesic curve $\gamma : [0,1] \to \mathcal{P}_{2,ac}(\mathbb{R}^n)$ such that $\gamma(0) = \mu$ and $\gamma(1) = \nu$ can be defined as follows:*

$$\gamma(t) = ((1-t)id + tT_{\mu,\nu})_{\#\mu}.$$

The exponential map and logarithmic map are then defined accordingly.

**Definition B.5** (Exponential map and Logarithmic map in Wasserstein space). *For $\mu, \nu \in \mathcal{P}_{2,ac}(\mathbb{R}^n)$ and $v \in \mathcal{L}^2(\mu)$, exponential map and logarithmic map of $(\mathcal{P}_{2,ac}(\mathbb{R}^n), W_2)$ are defined as follows:*

$$exp_\mu(v) = (v + id)_{\#\mu},$$
$$\log_\mu(\nu) = T_{\mu,\nu} - id.$$

Ambrosio & Gigli (2008) defined a geodesic parallel transport on 2-Wasserstein space.

**Definition B.6** (Parallel transport). *For $\mu, \nu \in \mathcal{P}_{2,ac}(\mathbb{R}^n)$ and $v \in T_\mu \mathcal{P}_{2,ac}(\mathbb{R}^n)$,*

$$\Gamma_\mu^\nu v := \Pi_\nu(v \circ T_{\nu,\mu}).$$

*Here, $\Pi.$ is a projection operator $\mathcal{L}^2(\cdot) \to T.\mathcal{P}_{2,ac}(\mathbb{R}^n)$.*

While this definition of parallel transport is theoretically sound, in terms of tractability this notion of parallel transport is hard to deal with; it involves the projection operator $\Pi$ which lacks an explicit form. In this regard, one often considers dropping out $\Pi$ and using the *un-projected* parallel transport $\Gamma_\mu^\nu v = v \circ T_{\nu,\mu}$ as the parallel transport from $\mathcal{L}^2(\nu)$ to $\mathcal{L}^2(\mu)$. Such construction is widely adopted in 2-Wasserstein geometry literature; see Ambrosio et al. (2008); Salim et al. (2020); Park et al. (2025) for instances. Perhaps surprisingly, this *un-projected* parallel transport still imposes favorable properties of parallel transports.

**Proposition B.7** (*Un-projected* parallel transport). *For $\mu, \nu \in \mathcal{P}_{2,ac}(\mathbb{R}^n)$ and $v \in \mathcal{L}^2(\mu)$, define $\Gamma_\mu^\nu v := v \circ T_{\nu,\mu}$. Then,*

1. *$\Gamma_\mu^\nu$ is linear operator on $\mathcal{L}^2(\mu)$.*

2. *$\Gamma_\mu^\nu \circ \Gamma_\nu^\mu = id$.*

3. *$\langle v, w \rangle_\mu = \langle \Gamma_\mu^\nu v, \Gamma_\mu^\nu w \rangle_\nu$.*

*Proof.* The property 1 is direct: for $v, w \in \mathcal{L}^2(\mu)$ and $a, b \in \mathbb{R}$, $\Gamma_\mu^\nu(av + bw) = av \circ T_{\mu,\nu} + bw \circ T_{\mu,\nu} = a\Gamma_\mu^\nu v + b\Gamma_\mu^\nu w$.

The property 2 is from Theorem B.2.

The property 3 is a direct consequence of the change of the measure formula:

$$\langle v, w \rangle_\mu = \int \langle v(x), w(x) \rangle \, d(T_{\nu,\mu})_{\#\nu}(x) = \int \langle v \circ T_{\nu,\mu}(x), w \circ T_{\nu,\mu}(x) \rangle \, d\nu(x) = \langle \Gamma_\mu^\nu v, \Gamma_\mu^\nu w \rangle_\nu.$$

$\square$

By Proposition B.7, we can consider using the *un-projected* parallel transport $\cdot \circ T_{\nu,\mu}$ as a parallel transport.

Next, we introduce the notion of Wasserstein gradient. Wasserstein gradient is defined analogously to the Riemannian gradient formula $df_x(v) = \langle \text{Grad} f(x), v \rangle_x$.

**Definition B.8** (Wasserstein gradient). *For a functional $\mathcal{F} : \mathcal{P}_{2,ac}(\mathbb{R}^n) \to \mathbb{R} \cup \{\infty\}$, the Wasserstein gradient of $\mathcal{F}$ at $\mu$ the mapping $\text{Grad}_{W_2} \mathcal{F}(\mu) \in \mathcal{L}^2(\mu)$ satisfying the following equation:*

$$\partial_t \mathcal{F}(\mu_t)\big|_{t=0} = \langle \text{Grad}_{W_2} \mathcal{F}(\mu), v_0 \rangle_\mu.$$

*Here, $\mu_t$ is arbitrary sufficiently regular curve of measures on $\mathcal{P}_{2,ac}(\mathbb{R}^n)$ with $\mu_0 = \mu$ and $v_t \in \mathcal{L}^2(\mu_t)$ is the vector field satisfying the continuity equation $\partial_t \mu_t = -\text{div}(\mu_t v_t)$.*

*One has the following explicit formula:*

$$\mathrm{Grad}_{\mathrm{W}_2} \mathcal{F}(\mu) = \nabla \frac{\delta \mathcal{F}(\mu)}{\delta \mu}.$$

*Here, $\nabla$ is Euclidean gradient and $\frac{\delta \mathcal{F}(\mu)}{\delta \mu}$ is the first variation.*

Here, the role of $\gamma'(0)$ is changed to $v_0$. More generally, one can think of $v_t$ as a velocity at $\mu_t$, playing the similar role to $\gamma'(t)$ in Riemannian manifolds. For the derivation, we refer to Chewi (2024, Theorem 1.4.1).

Consequently, all the notions of (generalized) geodesic convexity and local geodesic smoothness can be formulated in terms of this *un-projected* parallel transport and Wasserstein gradient.

**Definition B.9** (Generalized geodesic convexity and local geodesic smoothness in Wasserstein space). *Let $\mathcal{F} : \mathcal{P}_{2,ac}(\mathbb{R}^n) \to \mathbb{R} \cup \{\infty\}$ be a differentiable functional.*

1. *$\mathcal{F}$ is called generalized geodesically convex with base $\pi \in \mathcal{P}_{2,ac}(\mathbb{R}^n)$ if for all $\mu, \nu \in \mathcal{P}_{2,ac}(\mathbb{R}^n)$*
$$\mathcal{F}(\nu) \geq \mathcal{F}(\mu) + \langle \mathrm{Grad}_{\mathrm{W}_2} \mathcal{F}(\mu) \circ T_{\pi,\mu}, T_{\pi,\nu} - T_{\pi,\mu} \rangle_\pi.$$
   *If for given $\mu$ and $\nu$ $\mathcal{F}$ is generalized geodesically convex with base $\pi = \mu$, it is called geodesically convex, and if $\mathcal{F}$ is generalized geodesically convex with base $\pi$ for all $\pi \in \mathcal{P}_{2,ac}(\mathbb{R}^n)$, then it is called generalized geodesically convex.*

2. *$\mathcal{F}$ is locally geodesically smooth if for all compact set $K \subset \mathcal{P}_{2,ac}(\mathbb{R}^n)$ there exists a constant $L_K \geq 0$ such that*
$$\|\mathrm{Grad}_{\mathrm{W}_2} \mathcal{F}(\nu) \circ T_{\mu,\nu} - \mathrm{Grad}_{\mathrm{W}_2} \mathcal{F}(\mu)\|_\mu \leq L_K W_2^2(\mu, \nu) \qquad \forall \mu, \nu \in K.$$

### B.2.2 OTTO CALCULUS ON $\mathrm{BW}(\mathbb{R}^n)$

Now, we turn our attention to specifically to $\mathrm{BW}(\mathbb{R}^n)$, where much finer analyses are doable.

The first advantage to consider $\mathrm{BW}(\mathbb{R}^n)$ space is that the optimal transport map has the explicit form, particularly as the affine map.

**Definition B.10** (Optimal transport map between Gaussian). *The optimal transport map between $\mu_0 = N(m_0, \Sigma_0)$ and $\mu_1 = N(m_1, \Sigma_1)$ is defined as follows:*

$$T_{\mu_0,\mu_1}(x) = m_1 + \Sigma_0^{-1/2}(\Sigma_0^{1/2}\Sigma_1\Sigma_0^{1/2})^{1/2}\Sigma_0^{-1/2}(x - m_0).$$

Definition B.10 provides two favorable results.

First, since affine transform of the Gaussian is also a Gaussian, from Definition B.4 every geodesic interpolation between two Gaussians is also Gaussian. This shows $\mathrm{BW}(\mathbb{R}^n)$ is a geodesically convex subset of 2-Wasserstein space. In addition it implies $\mathrm{BW}(\mathbb{R}^n)$ is totally geodesic submanifold of 2-Wasserstein space (Lee, 2018, Exercise 8.4).

Second, we can identify $\mu = N(m, \Sigma) \cong (m, \Sigma) \in \mathbb{R}^n \times \mathrm{SPD}(n)$ and $T_\mu \mathrm{BW}(\mathbb{R}^n) \cong (a, S) \in \mathbb{R}^n \times \mathrm{Sym}(n)$. Here, $\mathrm{SPD}(n)$ is the space of the symmetric positive definite $\mathbb{R}^{n \times n}$ matrices, *i.e.*, positive semi-definite matrices, and $\mathrm{Sym}(n)$ is the space of $\mathbb{R}^{n \times d}$ symmetric matrices. By writing an affine map as $T(x) = a + S(x - m)$ for fixed $m$ (which is the mean of $\mu$), any affine map starting at $\mu = N(m, \Sigma)$ can be parameterized by $(a, S)$. Under this identification, we can view $\mathrm{BW}(\mathbb{R}^n)$ space as a product Riemannian manifold of $\mathbb{R}^n \times \mathrm{SPD}(n)$. Then one can parameterize every quantity in Appendix B.2.1 by this product manifold sense. For instance, the vector corresponding to the optimal transport map $T_{\mu_0,\mu_1}$ is $(m_1, \Sigma_0^{-1/2}(\Sigma_0^{1/2}\Sigma_1\Sigma_0^{1/2})^{1/2}\Sigma_0^{-1/2})$.

We can define Riemannian metric, exponential map, logarithmic map, and Bures-Wasserstein gradient in terms of parameters as well. These are basically just parameter-wise representations of the formula in Appendix B.2.1. We want to emphasize that the formulation we will introduce may be different with some other Bures-Wasserstein geometry literature (Thanwerdas & Pennec, 2022b). This is again because we are taking into account the Otto calculus formulation, and these two formulations are equivalent.

**Definition B.11** (Riemannian metric of Bures-Wasserstein space)**.** *Let $\mu = N(m, \Sigma)$. The Riemannian metric of* $\mathrm{BW}(\mathbb{R}^n)$ *is define by*

$$\langle (a_0, S_0), (a_1, S_1) \rangle_\mu = \langle a_0, a_1 \rangle_{\mathbb{R}^n} + tr(S_0 \Sigma S_1).$$

*The 2-Wasserstein distance between two Gaussian distributions, $\mu_i = N(m_i, \Sigma_i)$ for $i = 0, 1$, has the explicit form*

$$W_2^2(\mu_0, \mu_1) = \|m_0 - m_1\|^2 + \underbrace{\mathrm{tr}(\Sigma_0 + \Sigma_1 - 2(\Sigma_0^{1/2} \Sigma_1 \Sigma_0^{1/2})^{1/2})}_{:=d_{BW}(\Sigma_0, \Sigma_1)}.$$

**Definition B.12.** *(Lambert et al., 2022, Appendix B.3) Let $\mu_i = N(m_i, \Sigma_i)$. The exponential map and a logarithm map in* $\mathrm{BW}(\mathbb{R}^n)$ *are defined by*

$$\exp_{\mu_0}((a, S)) = N\left(a + m_0, (S + I)\Sigma_0 (S + I)\right),$$
$$\log_{\mu_0}(\mu_1) = (m_1 - m_0, \Sigma_0^{-1/2}(\Sigma_0^{1/2} \Sigma_1 \Sigma_0^{1/2})^{1/2} \Sigma_0^{-1/2} - I)$$

*for $a \in \mathbb{R}^n, -I \prec S \in \mathrm{Sym}(n)$.*

**Definition B.13.** *(Lambert et al., 2022, Appendix B.3) Bures-Wasserstein metric of the functional $\mathcal{F}$ can be written as a function on $\mathbb{R}^n \times \mathrm{PSD}(n)$, the space of the mean and covariance. Then, for $m \in \mathbb{R}$ and $\Sigma \in \mathrm{PSD}(n)$,*

$$\mathrm{Grad}_{\mathrm{BW}} \mathcal{F}(m, \Sigma) = (\nabla_m \mathcal{F}(m, \Sigma), 2\nabla_\Sigma \mathcal{F}(m, \Sigma)).$$

See Lambert et al. (2022); Diao et al. (2023) for further discussion.

**Remark B.14** (BW-gradients for KL-divergence)**.** In particular for the functional $\mathcal{F} = \mathcal{V} + \mathcal{H}$ we are considering in our main paper (Section 5.1), one gets the following form of the BW-gradients (Lambert et al., 2022):

$$\mathrm{Grad}_{\mathrm{BW}} \mathcal{V}(m, \Sigma) = \left( \mathbb{E}_{X \sim N(m, \Sigma)}[\nabla V(X)], \mathbb{E}_{X \sim N(m, \Sigma)}[\nabla^2 V(X)] \right)$$
$$\mathrm{Grad}_{\mathrm{BW}} \mathcal{H}(m, \Sigma) = (0, -\Sigma^{-1})$$
$$\mathrm{Grad}_{\mathrm{BW}} \mathcal{F}(m, \Sigma) = \left( \mathbb{E}_{X \sim N(m, \Sigma)}[\nabla V(X)], \mathbb{E}_{X \sim N(m, \Sigma)}[\nabla^2 V(X)] - \Sigma^{-1} \right)$$

If $V$ is not $C^1$ or $C^2$, $\nabla V$ and $\nabla^2 V$ can be regarded as the weak derivatives.

Using the isometry between the function representation and the vector-matrix representation of $T_p \mathrm{BW}(\mathbb{R}^n)$, we can define the following operation, which can be used to construct the (un-projected) parallel transport.

**Definition B.15.** *For $(a, S) \in T_{\mu_1} \mathrm{BW}(\mathbb{R}^n)$ and $(b, R) \in T_{\mu_0} \mathrm{BW}(\mathbb{R}^n)$, we have the following operation.*

$$(a, S) \circ (b, R) = (a + Sb - Sm_1, SR).$$

*In particular,*

$$\Gamma_{(m_1, \Sigma_1)}^{(m_0, \Sigma_0)}(a, S) = (a, S\Sigma_0^{-1/2}(\Sigma_0^{1/2} \Sigma_1 \Sigma_0^{1/2})^{1/2} \Sigma_0^{-1/2}).$$

Note the above formulations are just parameterization of the general Otto calculus in Appendix B.2.1, and one can write in either way to proceed. Throughout the paper, we will interchangably use the transport map formulation and vector-matrix formulations of $\mathrm{BW}(\mathbb{R}^n)$ geometry. Precisely, we will mainly use the transport map based formulation for the proofs, and the vector-matrix formulation for the algorithmic aspects.

It is known that $\mathrm{BW}(\mathbb{R}^n)$ space is geodesically incomplete, if one restricts to non-singular Gaussian distributions (Thanwerdas & Pennec, 2022a). On the other hand, if one includes the singluar distributions, then the space become geodesically complete (Takatsu, 2009), while the space is no longer precisely a manifold, but a cone. In addition, it is also known that $\mathrm{BW}(\mathbb{R}^n)$ is non-negatively curved manifold without the curvature upper bound. See Park et al. (2025, Lemma A.33, A.42) for the detail.

## C  DEFERRED PROOFS FOR SECTION 3

### C.1  PROOF OF THEOREM 3.3

For any compact set $K$, we have $\mathrm{diam}(K) := D < \infty$. We denote $\overline{B}(K, D) := \{p \mid d(p, K) \leq D\}$.

**Claim:** The below quantity $L_K$ is well-defined and finite:

$$L_K := \sup_{p \in \overline{B}(K,D)} \|\mathrm{Hess}\, f(p)\|_{op}.$$

To prove the claim, note $\overline{B}(K, D)$ is compact from Hopf-Rinow Theorem. In addition, from $f \in C^2(M)$, the map $p \mapsto \|\mathrm{Hess}\, f(p)\|_{op}$ is continuous. As a result, the function $p \mapsto \|\mathrm{Hess}\, f(p)\|_{op}$ admits the minimum and maximum on a compact set $\overline{B}(K, D)$. This proves the claim.

Take $x, y \in K$, let $\gamma$ be the minimizing geodesic with $\gamma(0) = y$ and $\gamma(1) = x$. Define

$$G(t) = \Gamma^y_{\gamma(t)}\, \mathrm{Grad}\, f(\gamma(t)).$$

Observe

$$
\begin{aligned}
G'(t) &= \lim_{h \to 0} \frac{\Gamma^y_{\gamma(t+h)}\, \mathrm{Grad}\, f(\gamma(t+h)) - \Gamma^y_{\gamma(t)}\, \mathrm{Grad}\, f(\gamma(t))}{h} \\
&\overset{(i)}{=} \lim_{h \to 0} \frac{(\Gamma^y_{\gamma(t)} \circ \Gamma^{\gamma(t)}_{\gamma(t+h)})\, \mathrm{Grad}\, f(\gamma(t+h)) - \Gamma^y_{\gamma(t)}\, \mathrm{Grad}\, f(\gamma(t))}{h} \\
&\overset{(ii)}{=} \Gamma^y_{\gamma(t)} \lim_{h \to 0} \left( \frac{\Gamma^{\gamma(t)}_{\gamma(t+h)}\, \mathrm{Grad}\, f(\gamma(t+h)) - \mathrm{Grad}\, f(\gamma(t))}{h} \right) \\
&\overset{(iii)}{=} \Gamma^y_{\gamma(t)} \nabla_{\dot\gamma(t)}\, \mathrm{Grad}\, f(\gamma(t)).
\end{aligned}
$$

Here, (i) is from the property of the parallel transport and the fact that $\gamma(t), \gamma(t+h)$ and $y$ lie on the same geodesic $\gamma$, (ii) is from the fact that the parallel transport is the continuous linear operator, and (iii) comes from the definition of the covariant derivative and the fact that $f \in C^2(M)$.

Next, we show

$$\left\| \nabla_{\dot\gamma(t)}\, \mathrm{Grad}\, f(\gamma(t)) \right\| \leq \|\mathrm{Hess}\, f(\gamma(t))\|_{op} \|\dot\gamma(t)\|. \tag{20}$$

Note this inequality trivially holds if $\nabla_{\dot\gamma(t)}\, \mathrm{Grad}\, f(\gamma(t)) = 0$. Otherwise, observe the following calculations:

$$
\begin{aligned}
\left\| \nabla_{\dot\gamma(t)}\, \mathrm{Grad}\, f(\gamma(t)) \right\|^2 &= \left\langle \nabla_{\dot\gamma(t)}\, \mathrm{Grad}\, f(\gamma(t)), \nabla_{\dot\gamma(t)}\, \mathrm{Grad}\, f(\gamma(t)) \right\rangle \\
&\overset{(i)}{=} \left\langle \mathrm{Hess}\, f(\gamma(t))\dot\gamma(t), \nabla_{\dot\gamma(t)}\, \mathrm{Grad}\, f(\gamma(t)) \right\rangle \\
&\overset{(ii)}{\leq} \|\mathrm{Hess}\, f(\gamma(t))\dot\gamma(t)\| \left\| \nabla_{\dot\gamma(t)}\, \mathrm{Grad}\, f(\gamma(t)) \right\| \\
&\leq \|\mathrm{Hess}\, f(\gamma(t))\|_{op} \|\dot\gamma(t)\| \left\| \nabla_{\dot\gamma(t)}\, \mathrm{Grad}\, f(\gamma(t)) \right\|.
\end{aligned}
$$

Here, (i) is from the definition of Riemannian Hessian, and (ii) is from Cauchy-Schwarz inequality.

Furthermore, we have $\gamma([0, 1]) \subset \overline{B}(K, D)$, as for all $t \in [0, 1]$

$$d(\gamma(t), K) \leq d(\gamma(t), y) \leq t\, d(x, y) \leq D.$$

Together with the construction of $L_K$, we get

$$\sup_{t \in [0,1]} \|\mathrm{Hess}\, f(\gamma(t))\|_{op} \leq L_K. \tag{21}$$

With these ingredients, we apply Fundamental Theorem of Calculus to get

$$\|\Gamma_x^y \operatorname{Grad} f(x) - \operatorname{Grad} f(y)\| = \left\|\int_0^1 G'(t)dt\right\| \le \int_0^1 \|G'(t)\| \, dt$$

$$\overset{(i)}{=} \int_0^1 \left\|\nabla_{\dot\gamma(t)} \operatorname{Grad} f(\gamma(t))\right\| dt$$

$$\overset{(ii)}{\le} \int_0^1 \|\operatorname{Hess} f(\gamma(t))\|_{op} \|\dot\gamma(t)\| \, dt \le \sup_{t\in[0,1]} \|\operatorname{Hess} f(\gamma(t))\|_{op} \int_0^1 \|\dot\gamma(t)\| \, dt$$

$$\overset{(iii)}{=} L_K d(x,y).$$

Here, (i) is from the isometry of the parallel transport, (ii) is from (20), and (iii) is from (21) and the fact that $\gamma$ is the minimizing geodesic. $\qquad\square$

## C.2 An auxiliary lemma used in the proof of Proposition 3.4

In this subsection, we state and prove a lemma that is used in the proof of Proposition 3.4. This can be considered as a local version of inequality (11.11) in Boumal (2023).

**Lemma C.1.** *Let $f : M \to \mathbb{R}$ is locally geodesically smooth function. For a compact set $K \subseteq M$, there exists $L_K$ such that*

$$f(y) \le f(x) + \langle \operatorname{Grad} f(x), \log_x y \rangle_x + \frac{L_K}{2} d^2(x,y) \qquad \forall x, y \in K.$$

The proof follows (Park et al., 2025, Lemma A.18). By the definition of the Riemannian logarithmic map, we get $\gamma'(0) = \log_x y$. By Fundamental Theorem of Calculus and properties of the parallel transport,

$$f(y) = f(\gamma(1)) = f(\gamma(0)) + \int_0^1 \frac{d}{dt}(f \circ \gamma)(t)dt = f(x) + \int_0^1 \langle \operatorname{Grad} f(\gamma(t)), \gamma'(t)\rangle \, dt$$

$$= f(x) + \int_0^1 \left\langle \Gamma_{\gamma(t)}^{\gamma(0)} \operatorname{Grad} f(\gamma(t)), \gamma'(0)\right\rangle dt = f(x) + \int_0^1 \left\langle \Gamma_{\gamma(t)}^x \operatorname{Grad} f(\gamma(t)), \log_x y \right\rangle dt.$$

Then, by subtracting $f(x) + \langle \operatorname{Grad} f(x), \log_x y \rangle$ from the both hand sides,

$$f(y) - f(x) - \langle \operatorname{Grad} f(x), \log_x y \rangle = \int_0^1 \left\langle \Gamma_{\gamma(t)}^x \operatorname{Grad} f(\gamma(t)) - \operatorname{Grad} f(x), \log_x y \right\rangle$$

$$\overset{(i)}{\le} \int_0^1 \left\|\Gamma_{\gamma(t)}^x \operatorname{Grad} f(\gamma(t)) - \operatorname{Grad} f(x)\right\| \|\log_x y\| \, dt$$

$$\overset{(ii)}{\le} \int_0^1 L_K d(\gamma(t),x)d(x,y)dt \overset{(iii)}{=} L_K d^2(x,y) \int_0^1 t dt$$

$$= \frac{L_K}{2} d^2(x,y).$$

For (i) we used Cauchy-Schwartz inequality. For (ii), we used the local geodesic $L$-smoothness with $L_K$ be the local geodesic smoothness parameter for the compact set $\overline{B}(K,D)$ where $D := \operatorname{diam}(K)$. This holds as $\gamma([0,1]) \subset \overline{B}(K,D)$, which was shown in the proof of Theorem 3.3. For (iii) we used the fact that the geodesic curve satisfies $d(x,\gamma(t)) = td(x,y)$ due to the constant speed property. $\qquad\square$

## C.3 Proof of Proposition 3.4

*Proof of Proposition 3.4.* We combine the proof strategy of Park et al. (2025, Lemma B.2) and Suh & Ma (2025, Lemma 1). Let $z := \exp_y\left(-\frac{1}{L_{\overline{K}}}\left(\operatorname{Grad} f(y) - \Gamma_x^y \operatorname{Grad} f(x)\right)\right)$. Note $x, y \in$

$\overline{B}(c, R) \subset \overline{B}(c, 3R)$. We first show $z \in \overline{B}(c, 3r)$.

$$d(z, c) \leq d(c, x) + d(y, z) \leq R + \left\| \frac{1}{L_{\overline{K}}} \left( \operatorname{Grad} f(y) - \Gamma_x^y \operatorname{Grad} f(x) \right) \right\|_y$$

$$\leq R + \frac{1}{L_{\overline{K}}} \left\| \operatorname{Grad} f(y) - \Gamma_x^y \operatorname{Grad} f(x) \right\|_y \leq R + d(x, y)$$

$$\leq R + d(x, c) + d(c, y) \leq 3R.$$

Therefore, $z \in \overline{B}(c, 3R)$. Note $\overline{K}$ is compact by Hopf-Rinow theorem.

Now, since $z \in N$ by the assumption, observe

$$f(x) - f(y) = f(x) - f(z) + f(z) - f(y)$$

$$\overset{(i)}{\leq} - \left\langle \Gamma_x^y \operatorname{Grad} f(x), \log_y z - \log_y x \right\rangle_y + \left\langle \operatorname{Grad} f(y), \log_y z \right\rangle_y + \frac{L_{\overline{K}}}{2} \left\| \log_y z \right\|_y^2$$

$$\overset{(ii)}{=} \left\langle \Gamma_x^y \operatorname{Grad} f(x), \log_y x \right\rangle - \frac{1}{2L_{\overline{K}}} \left\| \operatorname{Grad} f(y) - \Gamma_x^y \operatorname{Grad} f(x) \right\|_y^2$$

$$\overset{(iii)}{=} - \left\langle \operatorname{Grad} f(x), \log_x y \right\rangle - \frac{1}{2L_{\overline{K}}} \left\| \operatorname{Grad} f(x) - \Gamma_y^x \operatorname{Grad} f(y) \right\|_x^2.$$

We used generalized geodesic convexity and Lemma C.1 for (i), the construction of $z$ for (ii), and Lemma F.2 for (iii). Rearranging the final formula and swapping $x$ and $y$ yield the desired result. □

We additionally provide a useful corollary of Proposition 3.4 in the convergence analysis proofs presented later.

**Corollary C.2.** *(Suh & Ma, 2025, Corollary 2) Let $f : N \to \mathbb{R}$ be a locally geodesically smooth and generalized geodesically convex function. Assume $x, y \in \overline{B}(c, R)$ and $\overline{K} := \overline{B}(c, 3R) \subseteq N$ for some $c \in N$ and $R > 0$. Then, the following statements hold:*

1. *If $f(x) - f(y) + \left\langle \operatorname{Grad} f(y), \log_y x \right\rangle = 0$, then*

$$\left\| \Gamma_y^x \operatorname{Grad} f(y) - \operatorname{Grad} f(x) \right\|_x = 0.$$

2. *If $f(x) - f(y) + \left\langle \operatorname{Grad} f(y), \log_y x \right\rangle \neq 0$, then*

$$0 \leq -\frac{1}{2} \frac{\left\| \Gamma_y^x \operatorname{Grad} f(y) - \operatorname{Grad} f(x) \right\|_x^2}{f(y) - f(x) + \left\langle \operatorname{Grad} f(y), \log_y x \right\rangle_y} \leq L_{\overline{K}}$$

*where $L_{\overline{K}}$ is the local smoothness parameter of $\overline{K}$.*

*Proof.* For the first argument, the result directly follows from Proposition 3.4.

For the second argument, geodesic convexity and non-zeroness yield

$$f(x) - f(y) + \left\langle \operatorname{Grad} f(y), \log_y x \right\rangle > 0.$$

Since $\|\cdot\|^2 \geq 0$, this proves the non-negativeness. For the upper bound, since $f$ is generalized geodesically convex and locally geodesically smooth, Proposition 3.4 holds. Dividing the both hands sides of (4) by $\frac{L_{\overline{K}}}{f(y) - f(x) - \langle \operatorname{Grad} f(x), \log_x y \rangle}$ leads to the desired claim. □

## D   DEFERRED PROOFS FOR SECTION 4

### D.1   PROOF OF PROPOSITION 4.2

We show the following lemma to show Proposition 4.2.

**Lemma D.1.** *The following equality holds for any $k \geq -1$:*

$$
V_{k+1} - V_k
$$

$$
= \frac{1}{2} \left( \left\| \log_{x_{k+2}} x_\star \right\|^2 - \left\| \log_{x_{k+1}} x_\star \right\|^2 - \zeta_{k+1} \left\| \log_{x_{k+1}} x_{k+2} \right\|^2 + 2 \left\langle \log_{x_{k+1}} x_{k+2}, \log_{x_{k+1}} x_\star \right\rangle \right)
$$

$$
+ (s_{k+1} A_k + s_{k+1} - s_{k+2} A_{k+1})(f_\star - f(x_{k+1}))
$$

$$
+ s_{k+1} \left( f(x_{k+1}) - f_\star + \left\langle \operatorname{Grad} f(x_{k+1}), \log_{x_{k+1}} x_\star \right\rangle \right)
$$

$$
+ s_{k+1} A_k \bigg( f(x_{k+1}) - f(x_k) + \left\langle \operatorname{Grad} f(x_{k+1}), \log_{x_{k+1}} x_k \right\rangle_{x_{k+1}}
$$

$$
+ \frac{1}{2 L_{k+1}} \left\| \Gamma^{x_k}_{x_{k+1}} \operatorname{Grad} f(x_{k+1}) - \operatorname{Grad} f(x_k) \right\|^2 \bigg)
$$

$$
+ Q_k,
\tag{22}
$$

*where if $k \geq 0$*

$$
Q_k = -\frac{s_{k+1} A_k}{2 L_{k+1}} \bigg[ \left( 1 - \frac{B_{k+1} + \zeta_{k+1}}{A_k} s_{k+1} L_{k+1} \right) \left\| \operatorname{Grad} f(x_{k+1}) \right\|^2
$$

$$
- 2 \left( 1 - s_k L_{k+1} \right) \left\langle \Gamma^{x_k}_{x_{k+1}} \operatorname{Grad} f(x_{k+1}), \operatorname{Grad} f(x_k) \right\rangle + \left( 1 + \frac{B_k}{A_k} \frac{s_k^2}{s_{k+1}} L_{k+1} \right) \left\| \operatorname{Grad} f(x_k) \right\|^2 \bigg],
$$

*and $Q_{-1} = 0$.*

*Proof.* Reorganizing difference of $\frac{1}{2} \left\| \log_{x_{k+1}} x_\star \right\|^2$ term:

$$
\frac{1}{2} \left\| \log_{x_{k+2}} x_\star \right\|^2 - \frac{1}{2} \left\| \log_{x_{k+1}} x_\star \right\|^2
$$

$$
= \frac{1}{2} \left( \left\| \log_{x_{k+2}} x_\star \right\|^2 - \left\| \log_{x_{k+1}} x_\star \right\|^2 - \zeta_{k+1} s_{k+1}^2 \left\| \operatorname{Grad} f(x_{k+1}) \right\|^2 - 2 s_{k+1} \left\langle \operatorname{Grad} f(x_{k+1}), \log_{x_{k+1}} x_\star \right\rangle \right)
$$

$$
+ \frac{1}{2} \zeta_{k+1} s_{k+1}^2 \left\| \operatorname{Grad} f(x_{k+1}) \right\|^2 + s_{k+1} \left\langle \operatorname{Grad} f(x_{k+1}), \log_{x_{k+1}} x_\star \right\rangle
$$

$$
= \frac{1}{2} \left( \left\| \log_{x_{k+2}} x_\star \right\|^2 - \left\| \log_{x_{k+1}} x_\star \right\|^2 - \zeta_{k+1} \left\| \log_{x_{k+1}} x_{k+2} \right\|^2 + 2 \left\langle \log_{x_{k+1}} x_{k+2}, \log_{x_{k+1}} x_\star \right\rangle \right)
$$

$$
+ \frac{1}{2} \zeta_{k+1} s_{k+1}^2 \left\| \operatorname{Grad} f(x_{k+1}) \right\|^2 + s_{k+1} \left\langle \operatorname{Grad} f(x_{k+1}), \log_{x_{k+1}} x_\star \right\rangle
$$

Reorganizing difference of $s_{k+1} A_k (f(x_k) - f_\star)$ term:

$$
s_{k+2} A_{k+1}(f(x_{k+1}) - f_\star) - s_{k+1} A_k(f(x_k) - f_\star)
$$

$$
= (s_{k+1} A_k + s_{k+1} - s_{k+2} A_{k+1})(f_\star - f(x_{k+1})) + s_{k+1}(f(x_{k+1}) - f_\star)
$$

$$
+ s_{k+1} A_k \bigg( f(x_{k+1}) - f(x_k) + \left\langle \operatorname{Grad} f(x_{k+1}), \log_{x_{k+1}} x_k \right\rangle_{x_{k+1}}
$$

$$
+ \frac{1}{2 L_{k+1}} \left\| \Gamma^{x_k}_{x_{k+1}} \operatorname{Grad} f(x_{k+1}) - \operatorname{Grad} f(x_k) \right\|^2 \bigg)
$$

$$
- s_{k+1} A_k \left\langle \operatorname{Grad} f(x_{k+1}), \log_{x_{k+1}} x_k \right\rangle_{x_{k+1}} - \frac{s_{k+1} A_k}{2 L_{k+1}} \left\| \Gamma^{x_k}_{x_{k+1}} \operatorname{Grad} f(x_{k+1}) - \operatorname{Grad} f(x_k) \right\|^2.
$$

Combining two, we have

$$\left(\frac{1}{2}\left\|\log_{x_{k+2}} x_\star\right\|^2 + s_{k+2}A_{k+1}(f(x_{k+1}) - f_\star)\right) - \left(\frac{1}{2}\left\|\log_{x_{k+1}} x_\star\right\|^2 + s_{k+1}A_k(f(x_k) - f_\star)\right)$$

$$= \frac{1}{2}\left(\left\|\log_{x_{k+2}} x_\star\right\|^2 - \left\|\log_{x_{k+1}} x_\star\right\|^2 - \zeta_{k+1}\left\|\log_{x_{k+1}} x_{k+2}\right\|^2 + 2\left\langle\log_{x_{k+1}} x_{k+2}, \log_{x_{k+1}} x_\star\right\rangle\right)$$

$$+ (s_{k+1}A_k + s_{k+1} - s_{k+2}A_{k+1})(f_\star - f(x_{k+1}))$$

$$+ s_{k+1}A_k\left(f(x_{k+1}) - f(x_k) + \left\langle\operatorname{Grad} f(x_{k+1}), \log_{x_{k+1}} x_k\right\rangle_{x_{k+1}}\right.$$

$$\left. + \frac{1}{2L_{k+1}}\left\|\Gamma_{x_{k+1}}^{x_k} \operatorname{Grad} f(x_{k+1}) - \operatorname{Grad} f(x_k)\right\|^2\right)$$

$$+ \tilde{Q}_k,$$

where

$$\tilde{Q}_k = \frac{1}{2}\zeta_{k+1}s_{k+1}^2\left\|\operatorname{Grad} f(x_{k+1})\right\|^2 - s_{k+1}A_k\left\langle\operatorname{Grad} f(x_{k+1}), \log_{x_{k+1}} x_k\right\rangle_{x_k}$$

$$- \frac{s_{k+1}A_k}{2L_{k+1}}\left\|\Gamma_{x_{k+1}}^{x_k} \operatorname{Grad} f(x_{k+1}) - \operatorname{Grad} f(x_k)\right\|^2.$$

First, consider the boundary case $k = -1$. By the definition that $A_{-1} = 0$, $B_{-1} = B_0 + \zeta_0$, and $x_{-1} = x_0$, we obtain

$$\tilde{Q}_{-1} + \frac{1}{2}s_0^2 B_0\left\|\operatorname{Grad} f(x_0)\right\|^2 - \frac{1}{2}s_{-1}^2 B_{-1}\left\|\operatorname{Grad} f(x_{-1})\right\|^2 = 0,$$

which proves the result.

For $k \geq 0$, we conclude the desired equation by verifying

$$\tilde{Q}_k + \frac{1}{2}s_{k+1}^2 B_{k+1}\left\|\operatorname{Grad} f(x_{k+1})\right\|^2 - \frac{1}{2}s_k^2 B_k\left\|\operatorname{Grad} f(x_k)\right\|^2$$

$$= -\frac{s_{k+1}A_k}{2L_{k+1}}\left[\left(1 - \frac{B_{k+1} + \zeta_{k+1}}{A_k}s_{k+1}L_{k+1}\right)\left\|\operatorname{Grad} f(x_{k+1})\right\|^2\right.$$

$$\left. - 2\left(1 - s_k L_{k+1}\right)\left\langle\Gamma_{x_{k+1}}^{x_k}\operatorname{Grad} f(x_{k+1}), \operatorname{Grad} f(x_k)\right\rangle + \left(1 + \frac{B_k}{A_k}\frac{s_k^2}{s_{k+1}}L_{k+1}\right)\left\|\operatorname{Grad} f(x_k)\right\|^2\right],$$

where right hand side is the definition of $Q_k$. This can be verified by using Lemma F.2 and comparing the coefficients carefully. $\square$

Leveraging Lemma D.1, we can show Proposition 4.2.

*Proof of Proposition 4.2.* First, for the fixed $k$ (as stated in the Proposition, which satisfies $k \geq -1$), we can verify the following inequality by the stepsize rule (7):

$$s_{k+2} \leq \frac{A_k + 1}{A_{k+1}}s_{k+1}. \tag{23a}$$

Furthermore, if this $k$ satisfies $k \geq 0$, *i.e.*, $k \neq -1$, then the following inequalities also hold based on stepsize rule (7) and the condition (12):

$$s_{k+1} \leq \frac{A_k}{B_{k+1} + \zeta_{k+1}}s_k, \tag{23b}$$

$$s_{k+1} \leq \frac{A_k}{B_{k+1} + \zeta_{k+1}}\frac{1}{L_{k+1}} \qquad \text{if } s_k L_{k+1} \leq 1, \tag{23c}$$

$$s_{k+1} \leq \left(\frac{A_k}{B_k} + \frac{B_{k+1} + \zeta_{k+1}}{A_k}\right)^{-1}\frac{1}{L_{k+1}} \qquad \text{if } s_k L_{k+1} \geq 1. \tag{23d}$$

The desired inequality (13) then follows from applying Lemma D.1 with $\zeta = \zeta_{k+1}$. We now show how each term is upper bounded in Lemma D.1 with $\zeta = \zeta_{k+1}$.

From the definition of $\zeta_{k+1}$, we obtain $\zeta(d(x_{k+1}, x_\star)) = \zeta_{k+1}$. Applying Lemma F.1 with $x = x_{k+1}, y = x_{k+2}, p = x_\star$ yields:

$$\left\|\log_{x_{k+2}} x_\star\right\|^2 \leq \left\|\log_{x_{k+1}} x_\star\right\|^2 + \zeta_{k+1} \left\|\log_{x_{k+1}} x_{k+2}\right\|^2 - 2\left\langle \log_{x_{k+1}} x_{k+2}, \log_{x_{k+1}} x_\star\right\rangle$$

which implies the first term in Lemma D.1 is nonpositive.

The second term appears exactly in the upper bound of (13), and is nonpositive by (23a) and $f_\star \leq f(x_{k+1})$.

For the third term, geodesic convexity (which is guaranteed by the generalized geodesic convexity) between $x_{k+1}$ and $x_*$ leads to

$$f(x_{k+1}) - f(x_*) + \left\langle \operatorname{Grad} f(x_{k+1}), \log_{x_{k+1}} x_*\right\rangle_{x_{k+1}} \leq 0.$$

The fourth term vanishes, by the construction of $L_{k+1}$.

It only remains to show the bound on $Q_k$, *i.e.*, it remains to show:

$$Q_k \leq -\min\left\{\frac{A_k}{2}\frac{s_{k+1}}{L_{k+1}}, \ \frac{B_k}{2}s_k^2\right\}\|\operatorname{Grad} f(x_k)\|^2.$$

For $k = -1$, both sides equal zero by definition, verifying the inequality.

For $k \geq 0$, it suffices to show:

$$Q_k + \frac{B_k}{2}s_k^2\|\operatorname{Grad} f(x_k)\|^2 \leq 0 \qquad \text{if } s_k L_{k+1} \leq 1$$

$$Q_k + \frac{A_k}{2}\frac{s_{k+1}}{L_{k+1}}\|\operatorname{Grad} f(x_k)\|^2 \leq 0 \qquad \text{if } s_k L_{k+1} \geq 1.$$

First, we show the coefficient of the $\|\operatorname{Grad} f(x_{k+1})\|^2$ term is nonnegative. If $s_k L_{k+1} \leq 1$, from (23d), we have

$$1 - \frac{B_{k+1} + \zeta_{k+1}}{A_k}s_{k+1}L_{k+1} \geq 1 - \left(\frac{A_k}{B_k} + \frac{B_{k+1} + \zeta_{k+1}}{A_k}\right)s_{k+1}L_{k+1} \geq 0.$$

If $s_k L_{k+1} \geq 1$, inequality (23c) directly ensures the nonnegativity of $1 - \frac{B_{k+1} + \zeta_{k+1}}{A_k}s_{k+1}L_{k+1}$.

Then we use Lemma F.3 with $v = \operatorname{Grad} f(x_k)$ and $w = \operatorname{Grad} f(x_{k+1})$. Considering the discriminant of the quadratic form, our goal reduces to showing:

$$(1 - s_k L_{k+1})^2 - \left(1 - \frac{B_{k+1} + \zeta_{k+1}}{A_k}s_{k+1}L_{k+1}\right) \leq 0 \qquad \text{if } s_k L_{k+1} \leq 1,$$

$$(1 - s_k L_{k+1})^2 - \left(1 - \frac{B_{k+1} + \zeta_{k+1}}{A_k}s_{k+1}L_{k+1}\right)\frac{B_k}{A_k}\frac{s_k^2}{s_{k+1}}L_{k+1} \leq 0 \qquad \text{if } s_k L_{k+1} \geq 1.$$

Each case can be proved as follows:

- $s_k L_{k+1} \leq 1$. From (23b) we have

$$\frac{B_{k+1} + \zeta_{k+1}}{A_k}s_{k+1}L_{k+1} \leq s_k L_{k+1}.$$

  Note that For arbitrary $\delta_1, \delta_2 \in [0, 1]$, if $\delta_2 \leq \delta_1$ we know

$$(1 - \delta_1)^2 - (1 - \delta_2) \leq (1 - \delta_1) - (1 - \delta_2) = \delta_2 - \delta_1 \leq 0.$$

  We obtain the desired result by substituting $\delta_1 = s_k L_{k+1}$ and $\delta_2 = \frac{B_{k+1} + \zeta_{k+1}}{A_k}s_{k+1}L_{k+1}$.

- $s_k L_{k+1} \geq 1$. From (23d) we know

$$\frac{1}{s_{k+1} L_{k+1}} \geq \frac{A_k}{B_k} + \frac{B_{k+1} + \zeta_{k+1}}{A_k}$$

holds. Therefore, recalling that $1 - 2 s_k L_{k+1} \leq 0$, we obtain:

$$\left( 1 - \frac{B_{k+1} + \zeta_{k+1}}{A_k} s_{k+1} L_{k+1} \right) \frac{B_k}{A_k} \frac{s_k^2}{s_{k+1}} L_{k+1}$$

$$= \frac{B_k}{A_k} \left( \frac{1}{s_{k+1} L_{k+1}} - \frac{B_{k+1} + \zeta_{k+1}}{A_k} \right) s_k^2 L_{k+1}^2 \geq s_k^2 L_{k+1}^2 \geq (1 - s_k L_{k+1})^2 .$$

Now we move on to the case $L_{k+1} = 0$. Note that this implies $s_k L_{k+1} = 0 \leq 1$. From Corollary C.2 we have $\mathrm{Grad}\, f(x_k) = \Gamma^{x_k}_{x_{k+1}} \mathrm{Grad}\, f(x_{k+1})$. Then from the same calculation done to (22) but without $\frac{1}{L_{k+1}} \left\| \mathrm{Grad}\, f(x_k) - \Gamma^{x_k}_{x_{k+1}} \mathrm{Grad}\, f(x_{k+1}) \right\|^2$, and leveraging Lemma F.1 as discussed before for the first term we have:

$$\begin{aligned}
V_{k+1} - V_k &\leq (s_{k+1} A_k + s_{k+1} - s_{k+2} A_{k+1})(f_\star - f(x_{k+1})) \\
&\quad + s_{k+1}(f(x_{k+1}) - f_\star - \langle \mathrm{Grad}\, f(x_{k+1}), x_{k+1} - x_\star \rangle) \\
&\quad + s_{k+1} A_k \left( f(x_{k+1}) - f(x_k) - \langle \mathrm{Grad}\, f(x_{k+1}), x_{k+1} - x_k \rangle \right) \\
&\quad + Q_k \\
&\leq (s_{k+1} A_k + s_{k+1} - s_{k+2} A_{k+1})(f_\star - f(x_{k+1})) + Q_k,
\end{aligned} \tag{24}$$

where

$$\begin{aligned}
Q_k &= -s_k s_{k+1} A_k \left\langle \Gamma^{x_k}_{x_{k+1}} \mathrm{Grad}\, f(x_{k+1}), \mathrm{Grad}\, f(x_k) \right\rangle \\
&\quad + \frac{1}{2} s_{k+1}^2 (B_{k+1} + \zeta_{k+1}) \| \mathrm{Grad}\, f(x_{k+1}) \|^2 - \frac{1}{2} s_k^2 B_k \| \mathrm{Grad}\, f(x_k) \|^2 .
\end{aligned}$$

Note that following inequality follows from (23b):

$$s_{k+1} (B_{k+1} + \zeta_{k+1}) - s_k A_k \leq 0.$$

Therefore, by leveraging the fact $\mathrm{Grad}\, f(x_k) = \Gamma^{x_k}_{x_{k+1}} \mathrm{Grad}\, f(x_{k+1})$,

$$Q_k = \frac{1}{2} s_{k+1} (s_{k+1} (B_{k+1} + \zeta_{k+1}) - s_k A_k) \| \mathrm{Grad}\, f(x_k) \|^2 - \left( \frac{1}{2} s_k s_{k+1} A_k + \frac{1}{2} s_k^2 B_k \right) \| \mathrm{Grad}\, f(x_k) \|^2$$

$$\leq -\frac{1}{2} s_k^2 B_k \| \mathrm{Grad}\, f(x_k) \|^2 .$$

Substituting into (24), we obtain the desired result (13). $\qquad \square$

*Proof of Corollary 4.3.* From (13), we know that $V_{k+1} \leq V_k$ holds for all $k \geq k_0 - 1$. Consequently, we obtain

$$\left\| \log_{x_k} x_\star \right\|^2 \leq 2 V_{k-1} \leq 2 V_{k_0 - 1} = R^2, \quad \forall k > k_0$$

which implies our desired statement. This concludes the proof. $\qquad \square$

### D.2 Proof of Lemma 4.4

Define $S_k = \min \left\{ \frac{1}{L_0}, \frac{1}{L_1}, \ldots, \frac{1}{L_k} \right\}$. We prove the following by induction:

$$s_k \geq r S_k, \qquad \forall k \geq 1. \tag{25}$$

From (7) and the lower bound $r_k^L \geq \left( \frac{A_k}{B_k} + \frac{\tilde{B}_{k+1}}{A_k} \right)^{-1}$ from (6), we have

$$s_1 \geq \min \left\{ \max \left\{ A_0, \frac{\tilde{B}_1}{A_0} \right\} \min \left\{ \frac{1}{A_0}, \frac{A_0}{\tilde{B}_1} \right\} r \frac{1}{L_0}, r \frac{1}{L_1} \right\} = \min \left\{ r \frac{1}{L_0}, r \frac{1}{L_1} \right\} = r S_1.$$

Thus (25) holds for $k = 1$. Assume (25) holds for $k$, now we show that it also holds for $k + 1$. Applying the induction hypothesis $s_k \geq rS_k$ and (10), we obtain the following inequality, which completes the proof of (25):

$$s_{k+1} \geq \min \left\{ \min \left\{ \frac{A_{k-1} + 1}{A_k}, \frac{A_k}{\tilde{B}_{k+1}} \right\} rS_k, \left( \frac{A_k}{B_k} + \frac{\tilde{B}_{k+1}}{A_k} \right)^{-1} \frac{1}{L_{k+1}} \right\}$$

$$\geq \min \left\{ rS_k, r\frac{1}{L_{k+1}} \right\} = rS_{k+1}.$$

To prove (14), recall that we have $x_k \in \bar{B}_R(x_\star)$ from Proposition 4.2 and $\tilde{x}_0 \in \bar{B}_{d(\tilde{x}_0, x_\star)}(x_\star)$. Therefore, from Corollary C.2, we know that $L_k \leq L$ for $k \geq 0$, where $L$ is a smoothness parameter of $f$ on $\bar{B}_{3R}(x_\star) \cup \bar{B}_{3d(\tilde{x}_0, x_\star)}(x_\star)$. Again, the compactness of this set is guaranteed by Hopf-Rinow theorem. Therefore, we have $S_k \geq \frac{1}{L}$. From (25), we conclude (14). □

### D.3 PROOF OF THEOREM 4.5

The statements in Theorem 4.5 follow from Proposition 4.2 and Lemma 4.4.

For notation simplicity, denote $C_k = \min \left\{ \frac{A_k}{2} \frac{s_{k+1}}{L_{k+1}}, \frac{B_k}{2} s_k^2 \right\}$ First, summing up (13) from $k_0 - 1$ to $k - 1$ we have

$$V_k + \sum_{i=k_0-1}^{k-1} \left( (s_{i+1}A_i - s_{i+2}A_{i+1} + s_{i+1})(f(x_{i+1}) - f_\star) + C_i \left\| \text{Grad} f(x_i) \right\|^2 \right) \leq V_{k_0-1} = \frac{1}{2}R^2.$$

Combining with $s_{k+1}A_k(f(x_k) - f_\star) \leq V_k$, we have

$$s_{k+1}A_k(f(x_k) - f_\star) + \sum_{i=k_0}^{k} (s_i A_{i-1} - s_{i+1}A_i + s_i)(f(x_i) - f_\star) + \sum_{i=k_0-1}^{k-1} C_i \left\| \text{Grad} f(x_i) \right\|^2 \leq \frac{1}{2}R^2.$$

(26)

- *Function value, last iterate.*
  Recalling (23a), we know that the summations in (26) are nonnegative. Therefore, from (26), we have $s_{k+1}A_k(f(x_k) - f_\star) \leq \frac{1}{2}R$. Dividing both sides by $s_{k+1}A_k$ and applying Lemma 4.4, we obtain:

$$f(x_k) - f_\star \leq \frac{L}{2rA_k}R^2 = \mathcal{O}\left( \frac{L}{A_k} \right).$$

- *Gradient norm square, last iterate.*
  As mentioned in Theorem 4.5, by applying Proposition 3.4 with $y = x_k$ and $x = x_\star$ to the above inequality, we have:

$$\left\| \text{Grad} f(x_k) \right\|^2 \leq \frac{L^2}{rA_k}R^2 = \mathcal{O}\left( \frac{L^2}{A_k} \right).$$

- *Function value, minimum norm selection.*
  Next, considering only the function value terms on the left-hand side of (26), we obtain

$$s_{k+1}A_k(f(x_k) - f_\star) + \sum_{i=k_0}^{k} (s_i A_{i-1} - s_{i+1}A_i + s_i)(f(x_i) - f_\star) \leq \frac{1}{2}R^2.$$

(27)

Observing the sum of the coefficients of the $f(x_i) - f_\star$ terms, we see that

$$s_{k+1}A_k + \sum_{i=k_0}^{k} (s_i A_{i-1} - s_{i+1}A_i + s_i) = s_1 A_{k_0-1} + \sum_{i=k_0}^{k} s_i \geq \sum_{i=k_0}^{k} s_i \geq \frac{r}{L}k,$$

where the second inequality is from Lemma 4.4. From (23a), we know $s_i A_{i-1} - s_{i+1}A_i + s_i \geq 0$ for all $i \geq k_0$. Gathering these observations, from (27) we conclude

$$\min_{i \in \{k_0, \dots, k\}} (f(x_i) - f_\star) \leq \frac{L}{2rk}R^2 = \mathcal{O}\left( \frac{L}{k} \right).$$

- *Gradient norm square, minimum norm selection.*

  From Lemma 4.4, we know $s_k, s_{k+1} \geq \frac{r}{L}$, and from Corollary C.2, we know $\frac{1}{L_{k+1}} \geq \frac{1}{L}$. Focusing on the squared gradient norm terms in (26), enlarging the upper bound of the summation to $k$, and recalling the definition $C_k = \min\left\{\frac{A_k}{2}\frac{s_{k+1}}{L_{k+1}}, \frac{B_k}{2}s_k^2\right\}$, we obtain:

  $$\frac{r^2}{2L^2}\sum_{i=k_0-1}^{k}\min\{A_i, B_i\}\|\mathrm{Grad}\,f(x_i)\|^2 \leq \sum_{i=k_0}^{k}C_i\|\mathrm{Grad}\,f(x_i)\|^2 \leq \frac{1}{2}R^2. \quad (28)$$

  Note that, for simplicity, we have changed the lower bound of the summation to $k_0$. Therefore, we conclude

  $$\min_{i\in\{k_0,\ldots,k\}}\|\mathrm{Grad}\,f(x_i)\|^2 \leq \frac{L^2}{r^2\sum_{i=k_0}^{k}\min\{A_i, B_i\}}R^2.$$

  Lastly, from the second inequality of (10), it follows that $\limsup_{k\to\infty}\frac{A_k}{B_k} < \infty$. Therefore, we obtain $A_k = \mathcal{O}(\min\{A_k, B_k\})$, we conclude $\frac{1}{\sum_{i=k_0}^{k}\min\{A_i,B_i\}} = \mathcal{O}\left(\frac{1}{\sum_{i=k_0}^{k}A_i}\right)$.

- *Point convergence to a minimizer.*

  We follow the strategy of Suh & Ma (2025, Theorem 5). Take $\mathcal{X} = \{x \mid x \text{ is a minimizer of } f\}$. From the proof of Proposition 4.2, we have $x_k \in \bar{B}_R(x_*)$, which is (relatively) compact by Hopf-Rinow theorem. In addition, from (28), we have $\lim_{k\to\infty}\|\mathrm{Grad}\,f(x_k)\|^2 = 0$. Hence, cluster points of $\{x_k\}$ belong to $\mathcal{X}$. Then, setting $a_k = 2s_{k+1}A_k(f(x_k) - f_*) + s_k^2 B_k\|\mathrm{Grad}\,f(x_k)\|_{x_k}^2$ and using Lemma F.4 leads to the desired result.

### D.4  PROOF OF THEOREM 4.1

Note that Theorem 4.1 is a special case of Corollary 4.6. The condition $\tilde{B}_k = B_k + \bar{\zeta}$ is clear from the definition, it remains to check that the parameter choice of Theorem 4.1 satisfies the conditions in Theorem 4.5.

It is sufficient to check the conditions in (10). Since $A_{k+1} - A_k = \alpha \leq 1$, the first inequality of (10) holds. Also, $A_k - \tilde{B}_{k+1} = (1-\alpha) + \bar{\zeta} \geq 0$, the second inequality if (10) holds. Now, observe that

$$\frac{A_k}{B_k} + \frac{\tilde{B}_{k+1}}{A_k} = \frac{B_k + 1 + \bar{\zeta}}{B_k} + \frac{B_k + \alpha + \bar{\zeta}}{B_k + 1 + \bar{\zeta}} = 2 + \frac{(\alpha + \bar{\zeta})B_k + (1 + \bar{\zeta})^2}{B_k(B_k + 1 + \bar{\zeta})} = 2 + \frac{\alpha + \bar{\zeta} + \frac{(1+\bar{\zeta})^2}{B_k}}{B_k + 1 + \bar{\zeta}}$$

is decreasing for $k \geq 0$, since $B_k$ is increasing. Hence, the third inequality of (10) holds with $r = \left(\frac{A_0}{B_0} + \frac{B_1 + \bar{\zeta}}{A_0}\right)^{-1}$. Finally, we obtain:

$$\sum_{i=0}^{k}\min\{A_i, B_i\} = \sum_{i=0}^{k}\alpha(i+1) = \frac{1}{2}\alpha(k+1)(k+2).$$

We obtain the desired result by Corollary 4.6. $\qquad\square$

### D.5  PROOF OF COROLLARY 4.7

*Proof of Corollary 4.7.* We first establish that $V_0 \leq \frac{1}{2}R^2$. Observe that $\frac{1}{2}R^2 = V_{-1}$ as $B_{-1}$ is defined by $B_0 + \zeta_0$, and by the definition, $\zeta_0 = \zeta(d(x_0, x_*))$. Applying Proposition 4.2 with $k = -1$ yields the result $V_0 \leq V_{-1} = \frac{1}{2}R^2$, as required.

Next, we prove by induction that for all $k \geq 0$, $x_{k+1} \in \bar{B}_R(x_\star)$, $\zeta_{k+1} \leq \bar{\zeta}_0$, and the inequalities in (13) hold.

First, consider the base case when $k = 0$. Since $\zeta(r)$, defined in (2), is nondecreasing in $r$, we have $\zeta(d(x_0, x_\star)) \leq \zeta(\bar{d}_0)$. Moreover, we have $d(x_1, x_\star)^2 \leq 2V_0 \leq R^2 \leq \bar{d}_0^2 + (B_0 + \zeta(\bar{d}_0))s_0^2\|\mathrm{Grad}\,f(x_0)\|^2$. Hence, $x_1 \in \bar{B}_R(x_\star)$, and, by the monotonicity of $\zeta(\cdot)$, $\zeta_1 =$

$$\zeta(d(x_1, x_\star)) \leq \zeta\left(\sqrt{\overline{d}_0^2 + (B_0 + \zeta(\overline{d}_0))s_0^2 \|\text{Grad } f(x_0)\|^2}\right) = \overline{\zeta}_0. \text{ With } \tilde{B}_1 = B_1 + \overline{\zeta}_0 \geq B_1 + \zeta_1,$$

the conditions of Proposition 4.2 are satisfied with $k = 0$. Applying Proposition 4.2 with $k = 0$ implies (13) holds with $k = 0$.

We now proceed to the inductive step. Assume that for some $\tilde{k} \geq 0$ that $x_{k+1} \in \bar{B}_R(x_\star), \zeta_{k+1} \leq \overline{\zeta}_0$ and (13) hold for all $0 \leq k \leq \tilde{k}$. Then we have

$$d(x_{\tilde{k}+2}, x_\star)^2 \leq 2V_{\tilde{k}+1} \leq 2V_{\tilde{k}} \leq \cdots \leq 2V_0 \leq R^2 \leq \overline{d}_0^2 + (B_0 + \zeta(\overline{d}_0))s_0^2 \|\text{Grad } f(x_0)\|^2,$$

which implies $x_{\tilde{k}+2} \in \bar{B}_R(x_\star)$, and, again by monotonicity of $\zeta(\cdot)$, $\zeta_{\tilde{k}+2} \leq \overline{\zeta}_0$. Noticing $\tilde{B}_{\tilde{k}+2} = B_{\tilde{k}+2} + \overline{\zeta}_0 \geq B_{\tilde{k}+2} + \zeta_{\tilde{k}+2}$, and applying Proposition 4.2 with $k = \tilde{k} + 1$ shows that (13) holds with $k = \tilde{k} + 1$. This completes the induction.

Finally, following the same progression as in the proof of Theorem 4.5, the desired result follows directly. $\square$

### D.6 Details related to Remark 4.9

*Proof of Corollary 4.8.* Since $\lim_{k \to \infty}(\tilde{B}_k - B_k) = \infty$, there exists $k_0$ such that $\tilde{B}_k - B_k \geq \overline{\zeta}$ holds for all $k \geq k_0$. The statement follows immediately from Theorem 4.5. $\square$

**Details related to Remark 4.9.** Recall that the choices under consideration are $A_k = \alpha(k+1)+1$, $B_k = \tilde{\alpha}(k+1)$, and $\tilde{B}_k = \alpha(k+1)$, with $\alpha \in (0, 1]$ and $\tilde{\alpha} \in (0, \alpha)$. We first check that

$$\lim_{k \to \infty}(\tilde{B}_k - B_k) = (\alpha - \tilde{\alpha})(k+1) = \infty$$

holds since $\tilde{\alpha} \in (0, \alpha)$. Therefore, (12) holds by Corollary 4.8.

The rest of the argument can be carried out in a similar manner to Appendix D.4, and we provide it here for completeness. First, from $\alpha \in (0, 1]$, it follows that:

$$A_{k+1} - A_k = \alpha \leq 1, \qquad A_k - \tilde{B}_{k+1} = 1 - \alpha \geq 0, \qquad \forall k \geq 0.$$

Next, since

$$\frac{A_k}{B_k} + \frac{\tilde{B}_{k+1}}{A_k} = \frac{\alpha}{\tilde{\alpha}} + \frac{1}{B_k} + 1 - \frac{1-\alpha}{A_k} = \frac{\alpha}{\tilde{\alpha}} + 1 + \frac{A_k - (1-\alpha)B_k}{A_k B_k} = \frac{\alpha}{\tilde{\alpha}} + 1 + \frac{1}{A_k B_k} + \frac{(\alpha - \tilde{\alpha}) + \alpha\tilde{\alpha}}{\tilde{\alpha} A_k}$$

is decreasing for $k \geq 0$, since both $A_k$ and $B_k$ is increasing. Therefore, (10) holds. Lastly, we know

$$\sum_{i=k_0}^{k} \min\{A_i, B_i\} = \sum_{i=k_0}^{k} \alpha(i+1) = \frac{1}{2}\alpha\left((k+1)(k+2) - (k_0+1)(k_0+2)\right).$$

Therefore, by Corollary 4.8, we obtain

$$f(x_k) - f_\star \leq \frac{L}{2r}\frac{1}{\alpha(k+1)+1}R^2 = \mathcal{O}\left(\frac{L}{k}\right)$$

and

$$\min_{i \in \{k_0, \ldots, k\}} \|\text{Grad } f(x_i)\|^2 \leq \frac{2L^2}{\alpha r^2}\frac{1}{(k+1)(k+2) - (k_0+1)(k_0+2)}R^2 = \mathcal{O}\left(\frac{L^2}{k^2}\right).$$

### D.7 Additional Discussion on $A_k$, $B_k$, and $\tilde{B}_k$

In this subsection, we discuss the respective roles and the design mechanism behind $A_k$, $B_k$, and $\tilde{B}_k$. As a reminder, the sequences $A_k$, $B_k$, and $\tilde{B}_k$ are fundamentally designed to satisfy the inequalities in (23), thereby ensuring the validity of the Lyapunov analysis.

The convergence rates in Theorem 4.5 are written in terms of $A_k$, and a larger $A_k$ guarantees faster theoretical convergence rate. However, the first inequality of (10) implies $A_{k+1} \leq A_k + 1 \leq$

$A_0 + k + 1$, thus $A_k$ cannot grow arbitrarily large, and its largest possible order is $\mathcal{O}(k)$. While the tightest choice $A_{k+1} = A_k + 1$ yields the theoretically fastest guarantee, the resulting step size is nonincreasing, since $s_{k+2} \leq \frac{A_k+1}{A_{k+1}} s_k$ by (6) and (7). We therefore sacrifice part of the theoretical guarantee and consider a practically faster variant in the experiments in Section 5.

$\tilde{B}_k$ is introduced to account for the curvature of the manifold. It can be viewed as a variant of $B_k$ that upper-bounds $B_k$ while incorporating an additional curvature term $\zeta_k$, as seen in (12). The quantity $\tilde{B}_k$ appears in (6) and ensures that (23b), (23c), and (23d) hold, which are the core inequalities governing $s_{k+1}$ and validating the Lyapunov analysis. A larger $\tilde{B}_k$ leads to a smaller step size, which makes these inequalities easier to satisfy, but a smaller step size often results in slower convergence in practice.

As a natural follow-up, one might wonder whether $B_k$ can simply be taken arbitrarily small so as to obtain a smaller $\tilde{B}_k$, and hence a larger step size. However, if $\liminf_{k \to \infty} \frac{A_k}{B_k} = \infty$, then $r_k^L$ may converge to zero due to the term $\left( \frac{A_k}{B_k} + \frac{\tilde{B}_{k+1}}{A_k} \right)^{-1}$, which implies that the step size vanishes. Therefore, $B_k$ must have the same asymptotic order as $A_k$, and $\tilde{B}_k$ cannot be made arbitrarily small.

## E    DEFERRED PROOFS FOR SECTION 5

### E.1    DEFERRED PROOFS FOR SECTION 5.1

#### E.1.1    PROOF OF LEMMA 5.1

It is sufficient to check $A_k = 2(k+2)^{0.1} + 2$, $B_k = 2(k+2)^{0.1}$, and $\tilde{B}_k = 2(k+2)^{0.1} + 1$ satisfy the conditions in (10). First, by mean value theorem, for some $\tilde{k} \in [k+2, k+3]$ we have

$$A_{k+1} - A_k = 2 \left( (k+3)^{0.1} - (k+2)^{0.1} \right) = 2 \times 0.1 \times \tilde{k}^{-0.9}.$$

Since $\tilde{k} \mapsto \tilde{k}^{-0.9}$ is nonincreasing function and $\tilde{k} \geq 2$, we have

$$A_{k+1} - A_k \leq 2 \times 0.1 \times 2^{-0.9} \leq 1.$$

Therefore, the first inequality of (10) holds. Also,

$$A_k - \tilde{B}_{k+1} = 2(k+2)^{0.1} - 2(k+3)^{0.1} + 1 = -(A_{k+1} - A_k) + 1 \geq 0,$$

the second inequality if (10) holds. Finally, the existence of $r \in (0, 1)$ in the third inequality of (10) follows from the facts

$$\left( \frac{A_k}{B_k} + \frac{\tilde{B}_{k+1}}{A_k} \right)^{-1} > 0, \ \forall k \geq 0, \qquad \lim_{k \to \infty} \left( \frac{A_k}{B_k} + \frac{\tilde{B}_{k+1}}{A_k} \right)^{-1} = (1+1)^{-1} = \frac{1}{2} > 0.$$

$\square$

#### E.1.2    PROOF OF LEMMA 5.2

We show the generalized geodesic convexity of $\mathcal{V}$ and $\mathcal{H}$, and then how combining these two leads to the generalized geodesic convexity of $\mathcal{F}$. For the definition of generalized geodesic convexity in $\mathrm{BW}(\mathbb{R}^n)$ space, see Definition B.9.

First, we show if $V$ is convex then $\mathcal{V}$ is generalized geodesically convex. To this end, fix any base $\pi \in \mathrm{BW}(\mathbb{R}^n)$. Then, for all $\mu, \nu \in \mathrm{BW}(\mathbb{R}^n)$, there exist optimal transport maps $T_{\pi,\mu}$ and $T_{\pi,\nu}$.

Let $Z \sim \pi$. From the convexity of $V$, one gets

$$V(T_{\pi,\nu}(Z)) \geq \langle \nabla V(T_{\pi,\mu}(Z)), T_{\pi,\nu}(Z) - T_{\pi,\mu}(Z) \rangle.$$

Take the expectation over $Z \sim \pi$. Then, using the fact that Wasserstein gradient of $\mathcal{V} = \nabla V$ (from Definition B.8 and Santambrogio (2015, Section 8.2)), one gets the generalized geodesic convexity of $\mathcal{V}$ with base $\pi$. Since the above argument holds for any $\pi \in \mathrm{BW}(\mathbb{R}^n)$, we get the generalized geodesic convexity. For non-differentiable $V$, one can consider $\nabla V$ as the weak gradient.

The generalized geodesic convexity of $\mathcal{H}$ is established in Diao et al. (2023, Lemma 3.2).

Above two facts show that if $V$ is convex then $\mathcal{V}$ and $\mathcal{H}$ are generalized geodesically convex. Then, for any $\pi, \mu, \nu \in \mathrm{BW}(\mathbb{R}^n)$,

$$\mathcal{V}(\nu) \geq \mathcal{V}(\mu) + \langle \mathrm{Grad}_{\mathrm{BW}} \, \mathcal{V}(\mu) \circ T_{\pi,\mu}, T_{\pi,\nu} - T_{\pi,\mu} \rangle,$$
$$\mathcal{H}(\nu) \geq \mathcal{H}(\mu) + \langle \mathrm{Grad}_{\mathrm{BW}} \, \mathcal{H}(\mu) \circ T_{\pi,\mu}, T_{\pi,\nu} - T_{\pi,\mu} \rangle.$$

Summing these two inequalities lead to the generalized geodesic convexity of $\mathcal{F}$. □

### E.1.3 Proof of Proposition 5.3

**Proof of (i).** We first show any $C^2(\mathrm{BW}(\mathbb{R}^n))$ function is always locally geodesically smooth, as stated in Theorem 3.3. However, since $\mathrm{BW}(\mathbb{R}^n)$ is not a geodesically complete manifold, Hopf-Rinow theorem is not direct, and as a result one cannot directly apply Theorem 3.3. That said, note the only part the completeness of the manifold is used is when we show the compactness of $\overline{K} := \overline{B}(K, D)$, which is the set that contains all the minimizing geodesic segment. For $\mathrm{BW}(\mathbb{R}^n)$, we use an alternative set for $\overline{B}(K, D)$ which still guarantees the same properties (compactness, containing geodesics). Precisely, we show the lemma below:

**Lemma E.1.** *For given $\mu_0, \mu_1 \in \mathrm{BW}(\mathbb{R}^n)$, denote the minimizing geodesic from $\mu_1$ to $\mu_2$ as $\gamma_{\mu_0,\mu_1}$. Them, the set*

$$\overline{K} := \bigcup_{\mu_0,\mu_1 \in K} \gamma_{\mu_0,\mu_1}([0,1])$$

*is a compact set in $\mathrm{BW}(\mathbb{R}^n)$.*

*Proof of Lemma E.1.* Define $\mathcal{G} \colon \mathrm{BW}(\mathbb{R}^n) \times \mathrm{BW}(\mathbb{R}^n) \times [0,1] \to \mathrm{BW}(\mathbb{R}^n)$ as:

$$\mathcal{G}(\mu_0, \mu_1, t) = \gamma_{\mu_0,\mu_1}(t).$$

On the other hand, from Chen et al. (2019, Equation (9)), we know that any Bures–Wasserstein geodesic between $\mu_0 := (m_0, \Sigma_0)$ and $\mu_1 := (m_1, \Sigma_1)$ is (uniquely) defined by

$$\mathcal{G}((m_0, \Sigma_0), (m_1, \Sigma_1), t) = \left( (1-t)m_0 + tm_1, \ \Sigma_0^{-1/2} \left( (1-t)\Sigma_0 + t(\Sigma_0^{1/2}\Sigma_1\Sigma_0^{1/2})^{1/2} \right)^2 \Sigma_0^{-1/2} \right).$$

Leveraging the fact that the mappings $\Sigma \mapsto \Sigma^{-1}$ and $\Sigma \mapsto \Sigma^{1/2}$ are continuous on $\mathrm{SPD}(n)$, and that the binary operations used above (addition, scalar multiplication, matrix multiplication, etc.) are continuous, we know that $\mathcal{G}$ is a continuous function. Since $K \times K \times [0,1]$ is a compact set,

$$\overline{K} = \mathcal{G}\left( K \times K \times [0,1] \right).$$

is an image of a compact set under a continuous function $\mathcal{G}$, and therefore is a compact set. This concludes the proof. □

Now, by taking $L_K := \sup_{p \in \overline{K}} \|\mathrm{Hess}\, f(p)\|_{\mathrm{op}}$, one can proceed as in Theorem 3.3 and conclude that any $C^2$ function on $\mathrm{BW}(\mathbb{R}^n)$ is locally geodesically smooth.

**Proof of (ii).** Since we showed every $C^2$ function on $\mathrm{BW}(\mathbb{R}^n)$ is locally geodesically smooth, it is sufficient to show that potential energy functional and entropy functional are $C^2$. Note the differentiable structure of $\mathrm{BW}(\mathbb{R}^n)$ is determined by the differentiable structure of $\mathbb{R}^n \times \mathrm{SPD}(n)$. Hence, by Theorem 3.3 it is sufficient to show these functions are $C^2(\mathbb{R}^n \times \mathrm{SPD}(n))$ (and in fact they are $C^\infty(\mathbb{R}^n \times \mathrm{SPD}(n))$).

First, for $\mathcal{V}$, write the Gaussian density of $N(m, \Sigma)$ as $q_{m,\Sigma}$, *i.e.*, ,

$$q_{m,\Sigma}(z) = \frac{1}{(2\pi)^{n/2}|\Sigma|^{1/2}} \exp\left( -\frac{1}{2}(z-m)^\top \Sigma^{-1}(z-m) \right).$$

Since smoothness is local property, consider $(m, \Sigma)$, and some compact neighborhood $K$. We show $\mathcal{V}$ is smooth on $U = \mathrm{int}(K)$.

As $\mathcal{V}$ is written by the expectation, we invoke the dominated convergence theorem, which constitutes the most technical part of the proof. First, since $K$ is compact, there exist fixed constants $B > 0$ and $0 < m_K \leq M_K$ such that

$$\|m\| \leq B, \quad m_K \leq \lambda_{\min}(\Sigma) \leq \lambda_{\max}(\Sigma) \leq M_K \qquad \forall (m, \Sigma) \in K.$$

Now, for any multi-index $\alpha = (\alpha_m, \alpha_\Sigma)$ and any $(m, \Sigma) \in U$,

$$V(z) D^\alpha_{m,\Sigma} q_{m,\Sigma}(z) = V(z) \times P_\alpha(\Sigma^{-1}, z - m) q_{m,\Sigma}(z) \tag{29}$$

with some $P_\alpha$ which is a polynomial of the degree $p_\alpha \leq |\alpha_m| + 2|\alpha_\Sigma|$.

Now, recall for any $(m, \Sigma) \in K$, $\|\Sigma^{-1}\|_{op} \leq 1/m_K$, $(\det \Sigma)^{-1/2} \leq \lambda_{\min}(\Sigma)^{-n/2} \leq (m_K)^{-n/2}$, and $\lambda_{\max}(-\Sigma^{-1}) = -\frac{1}{\lambda_{\max}(\Sigma)} \leq -\frac{1}{M_K}$. Writing $y = z - m$, for any $(m, \Sigma) \in U \subset K$,

$$\left| V(z) D^\alpha_{m,\Sigma} q_{m,\Sigma}(z) \right| \leq C |V(z)| (1 + \|y\|^{p_\alpha}) |q_{m,\Sigma}(z)| \leq C |V(z)| (1 + \|y\|^{p_\alpha}) e^{-\frac{1}{2M_K} \|y\|^2}$$

for some constant $C > 0$ that depends on $m_K, M_K$, and $\alpha$, but not on $m$ and $\Sigma$.

Next, we plug-in (16). This leads to

$$\left| V(z) D^\alpha_{m,\Sigma} q_{m,\Sigma}(z) \right| \leq C(1 + \|z\|^p)(1 + \|y\|^{p_\alpha}) e^{a\|z\|^\beta} e^{-\frac{1}{2M_K} \|y\|^2}.$$

Then, from $\|m\| \leq B$, we have $(1 + \|z\|^p) \leq C(1 + \|y\|^p)$ for some $C > 0$. In addition, from $\beta \in [0, 2)$, there exists another constant $C > 0$ such that

$$e^{a\|z\|^\beta} \leq C e^{\frac{1}{4M_K} \|y\|^2}.$$

Therefore, we get

$$\left| V(z) D^\alpha_{m,\Sigma} q_{m\Sigma}(z) \right| \leq C(1 + \|y\|^{p + p_\alpha}) e^{-\frac{1}{4M_K} \|y\|^2} =: G(y)$$

where $C$ is the constant that does not depend on $(m, \Sigma)$.

Clearly, $G(y)$ is integrable over $\mathcal{L}^1(\mathbb{R}^n)$ from the fact that $\int G(y) dy$ is the Gaussian moment. Hence, $G(y)$ is the integrable majorant of $V(z) D^\alpha_{m,\Sigma} q_{m,\Sigma}(z)$ for any $(m, \Sigma) \in U$.

Therefore, by dominated convergence theorem, for any $|\alpha| \geq 0$ and any $(m, \Sigma) \in U$

$$D^\alpha_{m,\Sigma} \mathcal{V}(m, \Sigma) = \int_{\mathbb{R}^n} V(z) D^\alpha_{m,\Sigma} q_{m,\Sigma}(z) dz. \tag{30}$$

Once dominated convergence theorem is established, the rest is straightforward. Since $q_{m,\Sigma}(z)$ is smooth on $m$ and $\Sigma$, $D^\alpha_{m,\Sigma} q_{m,\Sigma}(z)$ is well-defined. In addition, from (29) and the fact that all moments of the Gaussian are well-defined, the integral at RHS of (30) is well-defined. Hence, $D^\alpha_{m,\Sigma} \mathcal{V}(m, \Sigma)$ is well-defined for any multi-index $\alpha$. This proves $\mathcal{V} \in C^\infty(\mathbb{R}^n \times \mathrm{SPD}(n))$.

**Proof of (iii).** For the entropy, one observes the Gaussian entropy has the form of

$$\mathcal{H}(m, \Sigma) = -\frac{n}{2}(1 + \log 2\pi) - \frac{1}{2} \log \det \Sigma. \tag{31}$$

This function does not involve $m$ and is clearly smooth on $\Sigma \in \mathrm{SPD}(n)$. As a result, $\mathcal{H} \in C^\infty(\mathbb{R}^n \times \mathrm{SPD}(n))$.

**Proof of (iv).** Lastly, $\mathcal{V}, \mathcal{H}$ being locally geodesically smooth trivially implies the local smoothness of $\mathcal{F}$. For $\mathcal{F} = \mathcal{V} + \mathcal{H}$ and for any compact set $K$, let the local geodesic smoothness parameters of $\mathcal{V}$ and $\mathcal{H}$ are $L_K^{\mathcal{V}}, L_K^{\mathcal{H}}$ respectively. Then, for all $x, y \in K$,

$$\|\Gamma_\nu^\mu \mathrm{Grad}\, \mathcal{F}(\mu) - \mathrm{Grad}\, \mathcal{F}(\nu)\| \leq \|\Gamma_\nu^\mu \mathrm{Grad}\, \mathcal{V}(\mu) - \mathrm{Grad}\, \mathcal{V}(\nu)\| + \|\Gamma_\nu^\mu \mathrm{Grad}\, \mathcal{H}(\mu) - \mathrm{Grad}\, \mathcal{H}(\nu)\|$$
$$\leq (L_K^{\mathcal{V}} + L_K^{\mathcal{H}}) W_2(\mu, \nu).$$

One concludes by taking $L_K^{\mathcal{F}} = L_K^{\mathcal{V}} + L_K^{\mathcal{H}}$. $\qquad\square$

### E.1.4 PROOF OF COROLLARY 5.4

---

**Algorithm 2** **B**ures-**W**asserstein **Ada**ptive **G**radient **V**ariational **I**nference (BWAdaGVI)

---

1: **Input:** Log-potential $V(X)$, $m_0 \in \mathbb{R}^n$, $\Sigma_0 \succeq \epsilon I$, $s_0 > 0$, $\{A_k\}_{k \geq 0}$, $\{B_k\}_{k \geq 0}$, $A_{-1} = 0$, $\delta \in (0, 1)$.
2: **for** $k = 0, 1, \dots$ **do**
3:

$$
m_{k+1} = m_k - s_k \mathbb{E}_{X \sim \mu_k}[\nabla V(X)], \qquad \mu_k = N(m_k, \Sigma_k)
$$
$$
S_k = \mathbb{E}_{X \sim \mu_k}[\nabla^2 V(X)] - \Sigma_k^{-1}
$$
$$
\Sigma_{k+1} = (I - s_k S_k)\Sigma_k(I - s_k S_k)
$$
$$
a_{k+1} = \mathbb{E}_{X \sim \mu_{k+1}}[\nabla V(X)], \qquad \mu_{k+1} = N(m_{k+1}, \Sigma_{k+1})
$$
$$
S_{k+1} = \mathbb{E}_{X \sim \mu_{k+1}}[\nabla^2 V(X)] - \Sigma_{k+1}^{-1}
$$
$$
\widehat{L}_{k+1} = -\frac{1}{2} \frac{\left\| \Gamma_{k+1}^k(a_{k+1}, S_{k+1}) - (a_k, S_k) \right\|_{\mu_k}^2}{\mathcal{F}(\mu_{k+1}) - \mathcal{F}(\mu_k) + s_k \left\langle \Gamma_{k+1}^k(a_{k+1}, S_{k+1}), (a_k, S_k) \right\rangle_{\mu_k}}
$$
$$
s_{k+1} = \min \left\{ \min \left\{ \frac{A_{k-1}+1}{A_k}, \frac{A_k}{\tilde{B}_{k+1}} \right\} s_k, \left( \frac{A_k}{B_k} + \frac{\tilde{B}_{k+1}}{A_k} \right)^{-1} \frac{1}{L_{k+1}}, \frac{1-\delta}{\max_{i=1,\dots,n} |\lambda_i(S_{k+1})|} \right\}
$$
$$
\tag{32}
$$

4: **end for**

---

We follow the proofs of Section 4, while incorporating the details needed to handle the geodesic incompleteness and specific geometry. We in fact prove more general statement here: any $\mathcal{F}$ that is $C^2(\text{BW}(\mathbb{R}^n))$ and generalized geodesically convex satisfies the claimed convergence rate. Over the proof, we will write $\mu = (m, \Sigma)$.

First, from the transport map representation of Bures-Wasserstein space and Definition B.13, one can write $\text{Grad}_{\text{BW}} \mathcal{F}(\mu)(x) = \nabla_m \mathcal{F}(m, \Sigma) + 2\nabla_\Sigma \mathcal{F}(m, \Sigma)(x - m)$. Hence, $\nabla \text{Grad}_{\text{BW}} \mathcal{F}(\mu) = 2\nabla_\Sigma \mathcal{F}(m, \Sigma)$, and $\|\nabla \text{Grad} \mathcal{F}(\mu)\|_{op} = 2 \max_{i=1,\dots,n} |\lambda_i(\nabla_\Sigma \mathcal{F}(m, \Sigma))|$.

Now, we check the new choices of stepsizes ensure the well-defined gradient iterates. First, observe

$$
s_k \leq \frac{1-\delta}{\max_{i=1,\dots,n} |\lambda_i(\nabla \text{Grad}_{\text{BW}} \mathcal{F}(\mu_k))|} < \frac{1}{\|\nabla \text{Grad}_{\text{BW}} \mathcal{F}(\mu_k)\|_{op}}.
$$

Hence, $I - s_k \nabla \text{Grad}_{\text{BW}} \mathcal{F}(\mu_k) \succ 0$, meaning $T_{\mu_k, \mu_{k+1}} = (id - s_k \text{Grad}_{\text{BW}} \mathcal{F}(\mu_k))$ is convex. By the Brenier theorem, this means the gradient iterate is the minimizing geodesic, and as a result the logarithmic map is properly defined as $T_{\mu_k, \mu_{k+1}} - id$. In addition, $I - s_k \nabla \text{Grad}_{\text{BW}} \mathcal{F}(\mu_k) \succ 0$ implies it is invertible, so that the exponential map is well-defined.

Now, we set $k_0 = 0$ and $\tilde{B}_k = B_k + 1$ for all $k$ in the proof of Proposition 4.2. In addition, one sets Lyapunov functional by

$$
V_k = s_{k+1} A_k (\mathcal{F}(\mu_k) - \mathcal{F}(\mu_*)) + \frac{1}{2} s_k^2 B_k \|\text{Grad}_{\text{BW}} \mathcal{F}(\mu_k)\|^2 + \frac{1}{2} \|T_{\mu_{k+1}, \mu_*} - id\|_{\mu_{k+1}}^2.
$$

This exactly coincides with (11) under $\text{BW}(\mathbb{R}^n)$ geometry. Then, Lemma D.1 with $\zeta = 1$ is still valid, as it is just the rearrangement of the terms. Next, one checks whether Proposition 4.2 is still valid under the changed stepsizes. As in Proposition 4.2, we check the non-positivity of the terms appearing in Lemma D.1.

- First term: One gets the non-positivity of the first term by the use of Lemma F.5.
- Second term: As in the proof of Proposition 4.2, the stepsize rule (7) and $\mathcal{F}(\mu_*) \leq \mathcal{F}(\mu_{k+1})$ gives the non-positivity.
- The third term is guaranteed to be non-positive from the geodesic convexity, which is guaranteed from the generalized geodesic convexity of $\mathcal{F}$.

- To check the fourth term, notice

$$L_{k+1} \geq -\frac{1}{2} \frac{\left\| \mathrm{Grad}_{\mathrm{BW}}\, \mathcal{F}(\mu_{k+1}) \circ T_{\mu_{k+1},\mu_k} - \mathrm{Grad}_{\mathrm{BW}}\, \mathcal{F}(\mu_k) \right\|^2}{\mathcal{F}(\mu_{k+1}) - \mathcal{F}(\mu_k) + s_k \left\langle \mathrm{Grad}_{\mathrm{BW}}\, \mathcal{F}(\mu_{k+1}) \circ T_{\mu_k,\mu_{k+1}}, \mathrm{Grad}_{\mathrm{BW}}\, \mathcal{F}(\mu_k) \right\rangle}.$$

The denominator is non-positive, due to the generalized geodesic convexity. Rearranging the above inequality, the fourth term becomes non-positive.

- One remains to check $Q_k \leq 0$. For $L_{k+1} \neq 0$, the sign discrimination test does not involve the explicit value of $L_{k+1}$, so we have $Q_k \leq 0$ for $L_{k+1} \neq 0$. For $L_{k+1} = 0$, since $L_{k+1} \geq 0$ it implies $\mathrm{Grad}_{\mathrm{BW}}\, \mathcal{F}(\mu_k) = \mathrm{Grad}_{\mathrm{BW}}\, \mathcal{F}(\mu_{k+1}) \circ T_{\mu_k,\mu_{k+1}}$, by the use of Corollary F.11. Hence, one can proceed the same as in Proposition 4.2.

Above calculations prove that the conclusion of Proposition 4.2, *i.e.*, non-increasing Lyapunov functional, is still valid for this problem.

Then, one is left with providing the lower bound of the stepsize, *i.e.*, analogous result to Lemma 4.4. To this end, we first observe that non-increasing Lyapunov functional gives

$$W_2^2(\mu_k, \mu_*) \leq 2V_{-1} = s_0^2(B_0 + 1) \left\| \mathrm{Grad}_{\mathrm{BW}}\, \mathcal{F}(\mu_0) \right\|^2 + W_2^2(\mu_0, \mu_*) =: R^2.$$

Then, from Definition B.11,

$$\|m_k - m_*\|^2 \leq R^2 \quad \text{and} \quad d_{BW}^2(\Sigma_k, \Sigma_*) = \mathrm{tr}(\Sigma_k + \Sigma_* - 2(\Sigma_k^{1/2}\Sigma_*\Sigma_k^{1/2})^{1/2}) \leq R^2. \quad (33)$$

Thus, for all $m_0, \ldots, m_k$ are contained in $K_m := \overline{B}(m_*, R)$ where the ball is with respect to Euclidean metric.

Next, we claim the condition on $d_{BW}^2(\Sigma_k, \Sigma_*)$ gives the uniform control over the largest eigenvalue of $\Sigma_k$. To check this, from Bhatia et al. (2019, Theorem 1),

$$\mathrm{tr}(\Sigma_k + \Sigma_* - 2(\Sigma_k^{1/2}\Sigma_*\Sigma_k^{1/2})^{1/2}) = \min_{U \in O(n)} \left\| \Sigma_k^{1/2} - \Sigma_*^{1/2}U \right\|_F^2 \leq R^2. \quad (34)$$

By triangular inequality, for any $U \in O(n)$,

$$\lambda_{\max}(\Sigma_k) = \|\Sigma_k\|_{op} \leq \left\| \Sigma_k^{1/2} - \Sigma_*^{1/2}U \right\|_{op} + \left\| \Sigma_*^{1/2}U \right\|_{op} \leq \left\| \Sigma_k^{1/2} - \Sigma_*^{1/2}U \right\|_F + \left\| \Sigma_*^{1/2} \right\|_{op}$$

$$= \left\| \Sigma_k^{1/2} - \Sigma_*^{1/2}U \right\|_F + \sqrt{\lambda_{\max}(\Sigma_*)}.$$

Since the above upper bound holds for any choice of $U \in O(n)$, by taking the minimum over $U \in O(n)$, one gets

$$\lambda_{\max}(\Sigma_k) \leq R + \sqrt{\lambda_{\max}(\Sigma_*)}. \quad (35)$$

In addition, since we assumed the uniform lower bound of the eigenvalues of $\Sigma_k$, we know for all $i = 0, \ldots, k$

$$\Sigma_i \in K_\Sigma := \left\{ \Sigma : | \epsilon I \preceq \Sigma \preceq \left( R + \sqrt{\lambda_{\max}(\Sigma_*)} \right) I \right\}.$$

Note $K_\Sigma$ is compact (e.g., one can use Heine-Borel theorem with respect to Frobenius norm).

Now, we construct the compact set $K \subset \mathrm{BW}(\mathbb{R}^n)$:

$$K = (K_m \times K_\Sigma) \cup \left\{ (\widetilde{m}_0, \widetilde{\Sigma}_0) \right\}.$$

Then, $K$ is the compact set that contains all $\mu_i$ as well as $\widetilde{\mu}_0$. Then, one gets $\overline{K} \supset K$ from Lemma F.8 and $\widetilde{L}_K$ from Proposition F.9.

Then, one repeats Lemma 4.4, with substituting $L$ by

$$\widehat{L}_K := \max \left\{ \widetilde{L}_K, \frac{\sup_{\mu \in K} \|\nabla \mathrm{Grad}_{\mathrm{BW}}\, \mathcal{F}(\mu)\|_{op}}{1 - \delta} \right\}$$

and

$$S_k = \min_{i=0,\dots,k} \left\{ \frac{1}{L_i}, \frac{1-\delta}{\|\nabla \operatorname{Grad}_{\mathrm{BW}} \mathcal{F}(\mu_i)\|_{op}} \right\}.$$

Note $\widehat{L}_K < \infty$ from the compactness of $K$. The induction part is direct from the condition $1 - \delta \geq r$. The uniform lower bound of $S_k$ can be established by $L_k \leq \widetilde{L}_K \leq \widehat{L}_K$ for $k \in \mathbb{N}$, which is from Corollary F.11, and $\sup_{k \in \mathbb{N}} \|\nabla \operatorname{Grad}_{\mathrm{BW}} \mathcal{F}(\mu_k)\|_{op} \leq \sup_{\mu \in K} \|\nabla \operatorname{Grad}_{\mathrm{BW}} \mathcal{F}(\mu)\|_{op} \leq (1-\delta)\widehat{L}_K$.

Since we established the non-increasing Lyapunov functional as well as the lower bound of the stepsizes, one can follow directly the proofs in Theorem 4.5 (i) and (ii) to obtain the desired result.

For the point convergence result (Theorem 4.5 (iii)), one can take the above $K$ as the (relatively) compact set containing $\{\mu_k\}_{k \in \mathbb{N}}$. Since Lemma F.4 holds for general metric space, the argument is valid regardless of geodesic incompleteness.

$\square$

**Comments on Remark 5.5**   We provide three theoretically ideal cases when one can ensure the uniform eigenvalue lower bound assumption.

**Case 1.** Uniform upper bound on the function value gives the uniform eigenvalue lower bound:

We show if $\sup_{i=1,\dots,k} \mathcal{F}(\mu_i) \leq M$ for some $M > 0$, then $\inf_{i=1,\dots,k} \lambda_{\min}(\Sigma_i) \geq \widetilde{\lambda}$ for some $\lambda_0 > 0$.

First, recall $m_k \in \overline{B}(m_\star, R)$ from (33). Since $V$ is continuous, one has

$$V_{\min} := \min_{m \in \overline{B}(m_*, R)} V(m) \leq V(m_k)$$

uniformly for all $k \in \mathbb{N}$.

Since $V$ is convex, from the Jensen's inequality and the explicit form of $\mathcal{H}$ in (31), one has

$$M \geq \mathcal{F}(\mu_k) = \mathbb{E}_{X \sim N(m_k, \Sigma_k)}[V(X)] - \frac{n}{2}(1 + \log 2\pi) - \frac{1}{2}\log \det \Sigma_k$$

$$\geq V(m_k) - \frac{n}{2}(1 + \log 2\pi) - \frac{1}{2}\log \det \Sigma_k$$

$$\geq V_{\min} - \frac{n}{2}(1 + \log 2\pi) - \frac{1}{2}\log \det \Sigma_k.$$

Then, one has

$$\lambda_{\max}(\Sigma_k)^{n-1}\lambda_{\min}(\Sigma_k) \geq \det \Sigma_k \geq \exp\left(2V_{\min} - n(1 + \log 2\pi) - 2M\right).$$

Using (35), one gets

$$\lambda_{\min}(\Sigma_k) \geq \frac{\exp\left(2V_{\min} - n(1 + \log 2\pi) - 2M\right)}{(R + \sqrt{\lambda_{\max}(\Sigma_\star)})^{n-1}} =: \lambda_0$$

which is the uniform bound independent of $k$.

$\square$

**Case 2.** A good initialization gives the uniform eigenvalue lower bound.

Next, we show a good initialization gives also gives the uniform lower bound of the smallest eigenvalues. Precisely, we show if $\mu_0$ is initialized to satisfy

$$R^2 = s_0^2(B_0 + 1)\|\operatorname{Grad}_{\mathrm{BW}} \mathcal{F}(\mu_0)\|^2 + W_2^2(\mu_0, \mu_*) < \lambda_{\min}(\Sigma_\star), \tag{36}$$

then there exists $\lambda_0 > 0$ which is the uniform eigenvalue lower bound.

From (34) and the standard inequality between the operator norm and Frobenius norm, there exists $U \in O(n)$ such that

$$\left\| \Sigma_k^{1/2} - \Sigma_\star^{1/2} U \right\|_{op} \leq \left\| \Sigma_k^{1/2} - \Sigma_\star^{1/2} U \right\|_F \leq R.$$

Then, from Weyl's inequality on singular value,

$$\left| \sigma_{\min}(\Sigma_k^{1/2}) - \sigma_{\min}(\Sigma_\star^{1/2}U) \right| \le \left\| \Sigma_k^{1/2} - \Sigma_\star^{1/2}U \right\|_{op} \le R.$$

Since orthogonal matrix preserves the singular values, we have

$$\sqrt{\lambda_{\min}(\Sigma_\star)} - R \le \sqrt{\lambda_{\min}(\Sigma_k)}.$$

Therefore, if $R < \sqrt{\lambda_{\min}(\Sigma_\star)}$, we have the uniform eigenvalue lower bound by $\lambda_0 = \left( \sqrt{\lambda_{\min}(\Sigma_\star)} - R \right)^2.$

$\square$

**Case 3.** A priori upper bounds on $\lambda_{\max}(\Sigma_*)$ and $W_2^2(\mu_0, \mu_*)$ give the uniform eigenvalue lower bound:

Lastly, we show if one has the information about $\lambda_{\max}(\Sigma_\star)$ and $W_2^2(\mu_0, \mu_*)$, then one can ensure the eigenvalue lower bound, though requiring much smaller stepsize. To this end, write $\overline{R} \ge R + \sqrt{\lambda_{\max}(\Sigma_\star)}$ in (35). Note this quantity requires the a priori upper bounds of $\lambda_{\max}(\Sigma_\star)$ and $W_2^2(\mu_0, \mu_*)$. We precisely show if $\Sigma_k \succeq \frac{1}{9\overline{R}}I$, then $\Sigma_{k+1}$ also has the same bound.

To this end, we claim that by choosing $\delta$ to be

$$\delta \in \left[ 1 - \frac{2}{117\overline{R}^2}, 1 \right) \tag{37}$$

ensures the uniform eigenvalue lower bound. Under this choice of $\delta$, observe the stepsize rule satisfies

$$s_{k+1} \le \frac{2}{117\overline{R}^2 \left\| \nabla \operatorname{Grad}_{\mathrm{BW}} \mathcal{F}(\mu_{k+1}) \right\|_{op}}. \tag{38}$$

Unlike Case 1 and 2, Case 3 directly modifies the stepsize rule, so we need to check whether the proof of Corollary 5.4 is still valid. Fortunately, this is direct; since the proof of Corollary 5.4 works regardless of the choice of $\delta$ (for the condition $1 - \delta \ge r$, since $r$ is just the lower bound (10), one can choose smaller $r$ to make the condition valid for given $\delta$, with the cost of constant factor in the convergence rate), all the proof arguments will be valid. The only thing we are left is to show the above choice of $\delta$ ensures the uniform lower bound on the eigenvalues of the iterates.

We will follow the proof of Lambert et al. (2022, Lemma 6). One can consider their $h = s_k / \left\| \mathbb{E}_{\mu_k}[\nabla^2 V(X)] \right\|_{op}$, and their $\nabla^2 V(\widehat{X}_k)$ as the normalized $\mathbb{E}_k[\nabla^2 V(X)] / \left\| \mathbb{E}_k[\nabla^2 V(X)] \right\|_{op}$ in our setting. Then, by the same arguments, one can check $\Sigma_{k+1}$ is the generalized Bures-Wasserstein barycenter of

$$P = \left( 1 - 2\frac{s_k}{\left\| \mathbb{E}_{\mu_k}[\nabla^2 V(X)] \right\|_{op}} \right) \delta_{\Sigma_k} + 2\frac{s_k}{\left\| \mathbb{E}_{\mu_k}[\nabla^2 V(X)] \right\|_{op}} \left( \delta_{\Sigma_k^{-1}} + \delta_{\widetilde{\Sigma}} \right)$$

where

$$\widetilde{\Sigma} = \left( 2I - \frac{\mathbb{E}_k[\nabla^2 V(X)]}{\left\| \mathbb{E}_k[\nabla^2 V(X)] \right\|_{op}} \right) \Sigma_k \left( 2I - \frac{\mathbb{E}_k[\nabla^2 V(X)]}{\left\| \mathbb{E}_k[\nabla^2 V(X)] \right\|_{op}} \right).$$

Next, suppose $\epsilon I \preceq \Sigma_k$. Then, as (35) is still valid (note this bound holds regardless of the existence of eigenvalue lower bound), we have

$$\epsilon \le \lambda(\Sigma_k) \le \overline{R}, \quad \frac{1}{\overline{R}} \le \lambda(\Sigma_k^{-1}) \le \frac{1}{\epsilon}, \quad \epsilon \le \lambda(\widetilde{\Sigma}) \le 4\overline{R}$$

where the last upper bound is from the same argument as in Lambert et al. (2022, Lemma 6). Next, as in Lambert et al. (2022, Lemma 6), we apply Altschuler et al. (2021, Theorem 1). Using the above eigenvalue bounds, one can check their $\alpha, \beta$ in Altschuler et al. (2021, Theorem 1) for our case becomes

$$\alpha = \left( \frac{1}{2\sqrt{\overline{R}}} + \frac{\sqrt{\epsilon}}{2} \right)^2, \quad \beta = \frac{1}{2\epsilon} + 2\overline{R}.$$

Then, by Altschuler et al. (2021, Theorem 1) if $2s_k / \left\| \mathbb{E}_{\mu_k}[\nabla^2 V(X)] \right\|_{op} \leq \alpha/(2\beta)$, we get $\lambda_{\min}(\Sigma_{k+1}) \geq \alpha/4$. Since we would like to guarantee $\lambda_{\min}(\Sigma_{k+1}) \geq \epsilon$, set $\alpha/4 = \epsilon$. Then solving $\epsilon, \alpha$, and $\beta$ with respect to $\overline{R}$ gives

$$\epsilon = \frac{1}{9\overline{R}}, \quad \alpha = \frac{4}{9\overline{R}}, \quad \beta = \frac{13}{2}\overline{R}.$$

Summarizing the above arguments, if $\lambda_{\min}(\Sigma_k) \geq (9\overline{R})^{-1}$ and $s_k$ satisfies

$$s_k \leq \frac{2}{117\overline{R}^2 \left\| \mathbb{E}_{\mu_k}[\nabla^2 V(X)] \right\|_{op}}, \tag{39}$$

then $\lambda_{\min}(\Sigma_{k+1}) \geq (9\overline{R})^{-1}$. Since $(9\overline{R})^{-1}$ does not depend on $k$, we have the uniform lower bound. Then, since

$$\| \nabla \operatorname{Grad}_{\mathrm{BW}} \mathcal{F}(\mu_k) \|_{op} = \left\| \mathbb{E}_{\mu_k}[\nabla^2 V(X)] + \Sigma_k^{-1} \right\|_{op} \geq \left\| \mathbb{E}_{\mu_k}[\nabla^2 V(X)] \right\|_{op},$$

the stepsize (38) satisfies (39). This means if we initialize $\Sigma_0 \succeq (9\overline{R})^{-1}I$, the new stepsize rule (38) ensures the eigenvalue lower bound $\Sigma_k \succeq (9\overline{R})^{-1}I$ for all $k \geq 0$.

In conclusion, all the proofs of Corollary 5.4 are valid under the choice of $\delta$ in (37), and the uniform eigenvalue lower bound condition is guaranteed without any additional assumption. Hence, with $\delta$ being (37), Corollary 5.4 is valid without any additional assumption. $\qquad \square$

**Special case when one can drop the uniform eigenvalue lower bound assumption** Particularly, Case 3 gives the fully explicit algorithm which does not require any eigenvalue assumptions, if one considers *fixed mean Gaussian variational families* with *strongly convex potential* $V$. This is because from Lambert et al. (2022, Equation (7)), if $V$ is $\alpha$-strongly convex, then one has

$$\lambda_{\max}(\Sigma_*) \leq 1/\alpha, \quad W_2^2(\mu_0, \mu_*) \leq \operatorname{tr}(\Sigma_0) + \operatorname{tr}(\Sigma_*) \leq \operatorname{tr}(\Sigma_0) + d/\alpha.$$

Hence,

$$\overline{R}^2 = \left( \sqrt{s_0^2(B_0+1)^2 \| \operatorname{Grad}_{\mathrm{BW}} \mathcal{F}(\mu_0) \|^2 + \operatorname{tr}(\Sigma_0) + d/\alpha} + \sqrt{1/\alpha} \right)^2$$

which is fully explicit and known for any fixed $\mu_0$. One can plug-in this value to (37), which guarantees the convergence result of Corollary 5.4 without any additional assumption.

### E.1.5 EXPLICIT FORM OF POTENTIAL AND GRADIENT

**Lemma E.2** (Closed form of potential and BW-gradient). *For any potential $V : \mathbb{R}^n \to \mathbb{R}$ of the form*

$$V(\theta) = \frac{1}{2}\theta^T Q \theta + q^T \theta + \sum_{i=1}^{\ell} c_i \exp(a_i^T \theta) + C$$

*where $Q \succeq 0, q, a_i \in \mathbb{R}^n, c_i \geq 0$, and constant $C$, $V$ is convex and satisfies (16). Moreover, the corresponding objective $\mathcal{F}(m, \Sigma)$ and its Bures-Wasserstein gradient $\operatorname{Grad}_{\mathrm{BW}} \mathcal{F}(m, \Sigma)$ admit explicit closed-form expressions given by:*

$$\mathcal{F}(m, \Sigma) = \frac{1}{2}(\operatorname{tr}(Q\Sigma) + m^T Q m) + q^T m + \sum_{i=1}^{\ell} c_i \exp\left( X_i^T m + \frac{1}{2} X_i^T \Sigma X_i \right) - \frac{1}{2}\log\det\Sigma + C,$$

$\operatorname{Grad}_{\mathrm{BW}} \mathcal{F}(m, \Sigma) =$

$$\left( Qm + q + \sum_{i=1}^{\ell} c_i X_i \exp\left( X_i^T m + \frac{1}{2} X_i^T \Sigma X_i \right), Q + \sum_{i=1}^{\ell} c_i X_i X_i^T \exp\left( X_i^T m + \frac{1}{2} X_i^T \Sigma X_i \right) - \Sigma^{-1} \right).$$

*Proof.* Convexity of $V$ follows directly from

$$\nabla^2 V(\theta) = Q + \sum_{i=1}^{\ell} c_i X_i X_i^T \exp(X_i^T \theta) \succeq 0.$$

In addition, $V$ trivially satisfies (16).

For the closed form expression, the formula of the moment generating function of multivariate normal distribution gives for any $X_i \in \mathbb{R}^n$,

$$\mathbb{E}_{\theta \sim N(m,\Sigma)}\left[\exp(X_i^T \theta)\right] = \exp\left(X_i^T m + \frac{1}{2} X_i^T \Sigma X_i\right). \tag{40}$$

In addition, the expectation of quadratic form for any $\theta$ with mean $m$ and the covariance $\Sigma$ is explicitly known:

$$\mathbb{E}_{\theta}[\theta^T Q \theta] = \text{tr}(Q\Sigma) + m^T Q m. \tag{41}$$

Using (40), (41), linearity of the expectation, and the closed form of $\mathcal{H}$ (31) yield the explicit form of $\mathcal{F}$.

For BW-gradient of $\mathcal{F}$, we have

$$\mathbb{E}_{\theta \sim N(m,\Sigma)}[\nabla V(\theta)] = Q\mathbb{E}_{\theta}[\theta] + q + \sum_{i=1}^{\ell} c_i X_i \mathbb{E}_{\theta}[\exp(X_i^T \theta)],$$

$$\mathbb{E}_{\theta \sim N(m,\Sigma)}[\nabla^2 V(\theta)] = Q + \sum_{i=1}^{\ell} c_i X_i X_i^T \mathbb{E}_{\theta}[\exp(X_i^T \theta)].$$

Plug-in (40) to the above formula. Then, using Remark B.14 yields the claimed result. $\qquad\square$

## F  AUXILIARY LEMMAS

This section includes the auxiliary lemmas needed for Appendix C, D, and E.

### F.1  AUXILIARY LEMMAS FOR COMPLETE RIEMANNIAN MANIFOLDS

We first introduce an existing inequality concerning metric distortion on manifolds, whose proof can be found in (Zhang & Sra, 2016, Lemma 5).

**Lemma F.1.** *Let $\zeta : [0,\infty) \to [1,\infty)$ be the function defined by (2). Then, for the vertices $x, y, p \in M$ of a uniquely geodesic triangle, we have*

$$\left\|\log_y p\right\|^2 \leq \left\|\log_x p\right\|^2 + \zeta(d(p,x)) \left\|\log_x y\right\|^2 - 2\left\langle \log_x y, \log_x p\right\rangle. \tag{42}$$

We also introduce some basic property of the Riemannian logarithmic map for the intermediate calculation. For the proof, see Park et al. (2025, Lemma C.7).

**Lemma F.2.** *For all $x, y \in M$, let $\Gamma_x^y$ be a parallel transport from $x$ to $y$ induced from the minimizing geodesic connecting $x$ and $y$. Then,*

$$\Gamma_x^y \log_x y = -\log_y x.$$

Next, we show the sign determination of the quadratic form by discriminant test is still valid in Riemannian manifold.

**Lemma F.3.** *Let $M$ be a Riemannian manifold, and $p, q \in M$. Then, if $a \geq 0$, $c \geq 0$, and $b^2 - 4ac \leq 0$, then*

$$a \left\|v\right\|_p^2 + b \left\langle \Gamma_p^q v, w\right\rangle_q + c \left\|w\right\|_q^2 \geq 0, \quad \forall v \in T_p M, w \in T_q M.$$

*Proof.* If one writes $x := \left\|v\right\|_p, y = \left\|w\right\|_q$, then the standard discriminant test on real numbers gives

$$ax^2 - |b| xy + cy^2 \geq 0.$$

From the Cauchy-Schwarz inequality and the isometry of the parallel transport, for all $b \in \mathbb{R}$

$$b \left\langle \Gamma_p^q v, w\right\rangle_q \geq -|b| \left\|\Gamma_p^q v\right\|_q \left\|w\right\|_q = -|b| xy.$$

Therefore,

$$a \left\|v\right\|_p^2 + b \left\langle \Gamma_p^q v, w\right\rangle_q + c \left\|w\right\|_q^2 \geq ax^2 - |b| xy + cy^2 \geq 0.$$

$\qquad\square$

Next, we provide the generalization of the result of Malitsky & Mishchenko (2024, Lemma 2) to metric spaces, which will be used to prove the asymptotic convergence to a minimizer for our algorithm.

**Lemma F.4.** *Let $(M, d)$ be a metric space. Let $A \subseteq M$ be the relatively compact set, and $\mathcal{X} \subseteq M$ be some fixed set. Suppose $\{x_k\}_{k \in \mathbb{N}} \subset A$ be a sequence whose cluster points belong to $\mathcal{X}$ and $\{a_k\}_{k \in \mathbb{N}}$ be a sequence in $\mathbb{R}_+$. If $\{x_k\}_{k \in \mathbb{N}}$ and $\{a_k\}_{k \in \mathbb{N}}$ satisfy*

$$d^2(x_{k+1}, x) + a_{k+1} \leq d^2(x_k, x) + a_k \quad \forall x \in \mathcal{X},$$

*then $\{x_k\}_{k \in \mathbb{N}}$ converges to some element in $\mathcal{X}$.*

*Proof.* Let $\overline{x}^1, \overline{x}^2$ be any cluster points of $\{x_i\}_{i \in \mathbb{N}}$. Then, since $A$ is relatively compact, there exist $x_{k_i}$ and $x_{k_j}$, two subsequences of $\{x_k\}_{k \in \mathbb{N}}$, such that $x_{k_i} \to \overline{x}^1$ and $x_{k_j} \to \overline{x}^2$. For fixed $x \in \mathcal{X}$, the sequence $d^2(x_k, x) + a_k$ is non-negative and non-increasing, hence by the monotone convergence theorem the limit $\lim_{k \to \infty} (d^2(x_k, x) + a_k)$ exists. Then,

$$\begin{aligned}
\lim_{k \to \infty} (d^2(x_k, \overline{x}^1) + a_k) &= \lim_{i \to \infty} (d^2(x_{k_i}, \overline{x}^1) + a_{k_i}) = \lim_{i \to \infty} a_{k_i} \\
&= \lim_{j \to \infty} (d^2(x_{k_j}, \overline{x}^1) + a_{k_j}) = d^2(\overline{x}^2, \overline{x}^1) + \lim_{j \to \infty} a_{k_j}.
\end{aligned}$$

Hence, one gets $\lim_{i \to \infty} a_{k_i} = d^2(\overline{x}^2, \overline{x}^1) + \lim_{j \to \infty} a_{k_j}$. By repeating the above argument symmetrically using $\lim_{k \to \infty} (d^2(x_k, \overline{x}^2) + a_k) = \lim_{j \to \infty} (d^2(x_{k_j}, \overline{x}^2) + a_{k_j})$, one gets $\lim_{j \to \infty} a_{k_j} = d^2(\overline{x}^2, \overline{x}^1) + \lim_{i \to \infty} a_{k_i}$. Thus, one gets $d^2(\overline{x}^1, \overline{x}^2) = 0$ and as a result $\overline{x}^1 = \overline{x}^2$. $\square$

## F.2 Auxiliary lemmas for $\mathrm{BW}(\mathbb{R}^n)$ space

To prove Corollary 5.4, we need the similar results we obtained in Appendix F.1 on $\mathrm{BW}(\mathbb{R}^n)$ space. Despite the absence of the completeness, it turns out that Bures-Wasserstein space have the similar properties. We provide the analogous statements for 2-Wasserstein space and $\mathrm{BW}(\mathbb{R}^n)$ we discussed in Section 3 and Appendix F.1 that do not require the completeness.

**Lemma F.5.** *For any $\mu, \nu, \pi \in \mathcal{P}_{2,ac}(\mathbb{R}^n)$ and $T_{a,b}$ being the optimal transport map from $a$ to $b$, the following inequality holds:*

$$\|T_{\nu,\pi} - id\|_\nu^2 \leq \|T_{\mu,\pi} - id\|_\mu^2 + \|T_{\mu,\nu} - id\|_\mu^2 - 2 \langle T_{\mu,\nu} - id, T_{\mu,\pi} - id \rangle_\mu.$$

*The result trivially extends to $\mathrm{BW}(\mathbb{R}^n)$ space.*

*Proof.* Note $T_{\mu,\pi} \circ T_{\nu,\mu}$ is also a (possibly non-optimal) transport map from $\nu$ to $\pi$. By the optimality of the optimal transport map $T_{\nu,\pi}$,

$$\begin{aligned}
\|T_{\nu,\pi} - id\|_\nu^2 &\leq \|T_{\mu,\pi} \circ T_{\nu,\mu} - id\|_\nu^2 \overset{(i)}{=} \|T_{\mu,\pi} - T_{\mu,\nu}\|_\mu^2 \\
&= \|T_{\mu,\pi} - id\|_\mu^2 + \|T_{\mu,\nu} - id\|_\mu^2 - 2 \langle T_{\mu,\nu} - id, T_{\mu,\pi} - id \rangle_\mu.
\end{aligned}$$

For (i), we used Proposition B.7 and Theorem B.2. $\square$

Note while we never used the curvature information to prove this, this result is indicating $\zeta = 1$ for $\mathcal{P}_{2,ac}(\mathbb{R}^n)$ space. In fact, $\mathcal{P}_{2,ac}(\mathbb{R}^n)$ space is non-negatively curved space, so this is a natural result.

**Lemma F.6.** *For all $\mu, \nu \in \mathcal{P}_{2,ac}(\mathbb{R}^n)$, we have*

$$(T_{\mu,\nu} - id) \circ T_{\nu,\mu} = -(T_{\nu,\mu} - id).$$

*Again, the result trivially extends to $\mathrm{BW}(\mathbb{R}^n)$ space.*

*Proof.* This comes directly from Theorem B.2 and Proposition B.7. $\square$

Next, we show the sign determination of the quadratic form by discriminant test is still valid in 2-Wasserstein space.

**Lemma F.7.** *For all $\mu, \nu \in \mathcal{P}_{2,ac}(\mathbb{R}^n)$, if $a \geq 0$, $c \geq 0$, and $b^2 - 4ac \leq 0$, then*

$$a \|v\|_{\mu}^2 + b \langle v \circ T_{\nu,\mu}, w \rangle_{\nu} + c \|w\|_{\nu}^2 \geq 0, \quad \forall v \in T_{\mu} \mathcal{P}_{2,ac}(\mathbb{R}^n), w \in T_{\nu} \mathcal{P}_{2,ac}(\mathbb{R}^n).$$

*Proof.* The same proof of Lemma F.3 works, once one simply substitutes the isometry of the parallel transport by Proposition B.7. $\qquad\square$

We now extend Proposition 3.4 and Corollary C.2 to $\mathrm{BW}(\mathbb{R}^n)$ space. Due to the fact that $\mathrm{BW}(\mathbb{R}^n)$ space is not geodesically complete, one cannot directly apply Proposition 3.4 to this setting. However, it turns out that one can still obtain the same conclusion under this specific geometry. Specifically, observe that the geodesic completeness is used when verifying the compactness of the ball $\overline{K} = \overline{B}(c, 3R)$, which is chosen to contain the point $z$ in the proof of Proposition 3.4 for all $x, y \in K$. We show one can still construct the similar compact set $\overline{K}$ for this problem, by using the explicit formula of tangent space, exponential map, and parallel transport.

**Lemma F.8.** *Let $K \subset \mathrm{BW}(\mathbb{R}^n)$ be a compact set, and $\mathcal{F} : \mathrm{BW}(\mathbb{R}^n) \to \mathbb{R}$ be a continuously differentiable function. Then, there exists a $L_0 > 0$ that only depends on $K$ and $\mathcal{F}$, and $\overline{K} \supset K$ that only depends on $L_0, \mathcal{F}$, and $K$ such that for all $L \geq L_0$ and all $(m_0, \Sigma_0), (m_1, \Sigma_1) \in K$,*

$$(\widetilde{m}, \widetilde{\Sigma}) := \exp_{(m_1, \Sigma_1)} \left( -\frac{1}{L} \left( \mathrm{Grad}_{\mathrm{BW}} \mathcal{F}(m_1, \Sigma_1) - \Gamma_{(m_0, \Sigma_0)}^{(m_1, \Sigma_1)} \mathrm{Grad}_{\mathrm{BW}} \mathcal{F}(m_0, \Sigma_0) \right) \right) \in \overline{K}.$$

*Proof.* We first consider the space of 0-mean Gaussians, so that we can restrict the analyses to $\mathrm{SPD}(n)$.

Since $K$ is compact in $\mathrm{SPD}(n)$ and the eigenvalue map is continuous, there exists $M_K \geq m_K > 0$ such that for all $\Sigma \in K$

$$0 < m_K \leq \lambda_{\min}(\Sigma) \leq \lambda_{\max}(\Sigma) \leq M_K < \infty.$$

Denote $H := \mathrm{Grad}_{\mathrm{BW}} \mathcal{F}(\Sigma_1) - \Gamma_{\Sigma_0}^{\Sigma_1} \mathrm{Grad}_{\mathrm{BW}} \mathcal{F}(\Sigma_0)$. Since $\mathrm{Grad}_{\mathrm{BW}} \mathcal{F}$ is continuous, there exists $G_K \in [0, \infty)$ such that

$$\sup_{\Sigma \in K} \|\mathrm{Grad}_{\mathrm{BW}} \mathcal{F}(\Sigma)\|_{\Sigma} = G_K.$$

Then, by triangular inequality and the isometry of the parallel transport (Proposition B.7)

$$\|H\|_{\Sigma_1} \leq \|\mathrm{Grad}_{\mathrm{BW}} \mathcal{F}(\Sigma_0)\| + \|\mathrm{Grad}_{\mathrm{BW}} \mathcal{F}(\Sigma_1)\| \leq 2G_K.$$

Since $\|H\|_{\Sigma}^2 = \mathrm{tr}(H\Sigma H)$, one has

$$\|H\|_{op} \leq \|H\|_F \leq \frac{\|H\|_{\Sigma}}{\sqrt{m_K}} \leq \frac{2G_K}{\sqrt{m_K}} =: C_K.$$

Now, we claim any $L_0 > C_K$ is the desired $L_0$. Define $\epsilon := 1 - C_K/L_0$, which lies on $(0, 1]$ from the condition $L_0 > C_K \geq 0$. We then set

$$\overline{K} := \left\{ \Sigma \mid \epsilon^2 m_K I \preceq \Sigma \preceq 4M_K I \right\}.$$

We show $\overline{K}$ is the claimed compact set. This set is compact from Heine-Borel theorem on $\mathbb{R}^{n \times n}$ with a matrix norm (e.g. Frobenius). Then, from $\epsilon \leq 1$, $K \subseteq \overline{K}$. Thus, we are left to show $\widetilde{\Sigma} \in \overline{K}$. This is sufficient to show the eigenvalue bounds of $\widetilde{\Sigma}$.

Now, for any $L \geq L_0$, since the tangent space is $\mathrm{Sym}(n)$, $A_L := I - H/L$ is symmetric. Using the explicit form of the exponential map (Definition B.12),

$$\widetilde{\Sigma} = A_L \Sigma_1 A_L = \left( \Sigma_1^{1/2} A_L \right)^T \left( \Sigma_1^{1/2} A_L \right).$$

Thus,

$$\lambda_{\min}(\widetilde{\Sigma}) = \left( \sigma_{\min}(\Sigma_1^{1/2} A_L) \right)^2, \quad \lambda_{\max}(\widetilde{\Sigma}) = \left( \sigma_{\max}(\Sigma_1^{1/2} A_L) \right)^2$$

where $\sigma$ is the singular value. Now, using the inequalities between the singular values,

$$\sigma_{\min}(\Sigma_1^{1/2}A_L) \geq \sigma_{\min}(\Sigma_1^{1/2})\sigma_{\min}(A_L) = \sqrt{\lambda_{\min}(\Sigma_1)}\sigma_{\min}(A_L) \geq \sqrt{m_K}\sigma_{\min}(A_L),$$
$$\sigma_{\max}(\Sigma_1^{1/2}A_L) \leq \sigma_{\max}(\Sigma_1^{1/2})\sigma_{\max}(A_L) = \sqrt{\lambda_{\max}(\Sigma_1)}\sigma_{\max}(A_L) \leq \sqrt{M_K}\sigma_{\max}(A_L).$$

Now, we bound $\sigma_{\min}(A_L)$ and $\sigma_{\max}(A_L)$. Since $A_L$ is symmetric, the singular value $\sigma_i(A_L) = |\lambda_i(A_L)| = |1 - \lambda_i/L|$. Hence,

$$\sigma_{\min}(A_L) \geq \min_i \left|1 - \frac{\lambda_i}{L}\right| \geq 1 - \frac{\max_i |\lambda_i|}{L} \geq 1 - \frac{\|H\|_{op}}{L} = 1 - (1-\epsilon)\frac{L_0}{L} \geq \epsilon,$$
$$\sigma_{\max}(A_L) \leq \max_i \left|1 - \frac{\lambda_i}{L}\right| \leq 1 + \frac{\max_i |\lambda_i|}{L} \leq 1 + \frac{\|H\|}{L} < 1 + \frac{L_0}{L} \leq 2.$$

Aggregating the results, one gets

$$\epsilon^2 m_K \leq \lambda_{\min}(\widetilde{\Sigma}) \leq \lambda_{\max}(\widetilde{\Sigma}) \leq 4M_K.$$

This means $\widetilde{\Sigma} \in \overline{K}$, which completes the proof for the 0-mean case.

The extension to arbitrary $\mathrm{BW}(\mathbb{R}^n)$ is direct, since product of compact sets are compact. We identified the compact set for matrix component, and one can simply take the same $\overline{B}(c, 3R)$ as we did in Proposition 3.4 for Euclidean mean part. $\square$

**Proposition F.9.** *Suppose $\mathcal{F} : \mathrm{BW}(\mathbb{R}^n) \to \mathbb{R}$ is generalized geodesically convex and locally geodesically smooth. For any compact set $K$, there exists $\widetilde{L}_K > 0$ such that for all $\mu, \nu \in K$*

$$\mathcal{F}(\nu) - \mathcal{F}(\mu) - \langle \mathrm{Grad}_{\mathrm{BW}}\,\mathcal{F}(\mu), T_{\mu,\nu} - id \rangle_\mu - \frac{1}{2\widetilde{L}_K}\|\mathrm{Grad}_{\mathrm{BW}}\,\mathcal{F}(\nu) \circ T_{\mu,\nu} - \mathrm{Grad}_{\mathrm{BW}}\,\mathcal{F}(\mu)\|_\mu^2 \geq 0.$$

*Proof.* First, we take $L_0$ the same as in the proof of Lemma F.8. Then, by Lemma F.8, there exists a compact set $\overline{K}$ such that it contains all $(\widetilde{m}, \widetilde{\Sigma})$ (which is defined in Lemma F.8) for all $L \geq L_0$. Now, define

$$\widetilde{L}_K := \max\{L_0, L_{\overline{K}}\}$$

where $L_{\overline{K}}$ is the local smoothness parameter of the compact set $\overline{K}$. Then, $\widetilde{L}_K$ is also a local smoothness parameter of $\overline{K}$, which is the compact set containing both $\nu$ as well as $\pi := \exp_\nu\left(-\frac{1}{\widetilde{L}_K}(\mathrm{Grad}_{\mathrm{BW}}\,\mathcal{F}(\nu) - \Gamma_\mu^\nu\,\mathrm{Grad}_{\mathrm{BW}}\,\mathcal{F}(\mu))\right)$ by Lemma F.8.

Now, one can proceed with the exactly same proof as in Proposition 3.4, with the following substitutions:

- Use $\mu, \nu, \pi$ instead of $x, y, z$.

- Use $\widetilde{L}_K$ instead of $L_{\overline{K}}$.

- Use Definition B.9 for the definition of generalized geodesic convexity and local geodesic smoothness.

- Use Lemma F.6 instead of Lemma F.2.

- Use Definition B.5, Proposition B.7, and Definition B.8 for the exponential map, logarithmic map, parallel transport, and Riemannian gradient.

$\square$

**Remark F.10** (From $\mathcal{P}_{2,ac}(\mathbb{R}^n)$ to $\mathrm{BW}(\mathbb{R}^n)$)**.** Note unlike the previous lemmas dealt with the general $\mathcal{P}_{2,ac}(\mathbb{R}^n)$, from this proposition we narrowed down our focus to $\mathrm{BW}(\mathbb{R}^n)$ specifically. This is because the compactness argument in Lemma F.8 required the explicit form of the $\mathrm{BW}(\mathbb{R}^n)$ geometry.

**Corollary F.11.** *Let $\mathcal{F} : \mathrm{BW}(\mathbb{R}^n) \to \mathbb{R}$ be a generalized geodesically convex and locally geodesically smooth function. Then, the following statements hold:*

1. *For any $\mu, \nu \in \mathrm{BW}(\mathbb{R}^n)$ with $\mathcal{F}(\nu) - \mathcal{F}(\mu) + \langle \mathrm{Grad}_{\mathrm{BW}} \mathcal{F}(\nu), T_{\nu,\mu} - id \rangle = 0$,*

$$\|\mathrm{Grad}_{\mathrm{BW}} \mathcal{F}(\nu) \circ T_{\mu,\nu} - \mathrm{Grad}_{\mathrm{BW}} \mathcal{F}(\mu)\|_\mu = 0.$$

2. *Let $K \subset \mathrm{BW}(\mathbb{R}^n)$ be a compact set. Then there exists $\widetilde{L}_K > 0$ such that for all $\mu, \nu \in K$ with $\mathcal{F}(\nu) - \mathcal{F}(\mu) + \langle \mathrm{Grad}_{\mathrm{BW}} \mathcal{F}(\nu), T_{\nu,\mu} - id \rangle \neq 0$,*

$$0 \leq -\frac{1}{2} \frac{\|\mathrm{Grad}_{\mathrm{BW}} \mathcal{F}(\nu) \circ T_{\mu,\nu} - \mathrm{Grad}_{\mathrm{BW}} \mathcal{F}(\mu)\|_\mu^2}{\mathcal{F}(\nu) - \mathcal{F}(\mu) + \langle \mathrm{Grad}_{\mathrm{BW}} \mathcal{F}(\nu), T_{\nu,\mu} - id \rangle_\mu} \leq \widetilde{L}_K.$$

*Proof.* One can proceed the same as in the proof of Corollary C.2, but based on Proposition F.9. Note we dropped the condition on $B(c, 3R)$ and $N$, since we set $N = M = \mathrm{BW}(\mathbb{R}^n)$. $\qquad\square$

# G  ADDITIONAL EXPERIMENTS

We provide additional experiments that will also validate the strength of RAdaGD over various tasks. All experiments were conducted either in free version of Google Colab environment or Apple Macbook Air M4, and no experiment took more than 5 minutes to run.

## G.1  ADDITIONAL EXPERIMENTS ON GAUSSIAN VARIATIONAL INFERENCE

We conduct GVI experiments on two additional problems. Since both fall within the scope of Lemma E.2, BWAdaGVI is accordingly endowed with the theoretical guarantees provided by Corollary 5.4.

1. $\pi$ is a Gaussian target, corresponding to a quadratic potential $V$. In this case, $V$ is convex and $L$-smooth, so both BWAdaGVI and FBGVI (Diao et al., 2023) enjoy provable guarantees.

2. $\pi$ is the distribution of the response variable in log-link Gamma regression, a generalized linear model. Although the canonical link for Gamma regression is the inverse function, the log-link is often preferred due to the unfavorable behavior of the inverse (Czado, 2004, Page 21). In this setting, if

$$Y_i \mid X_i \overset{ind}{\sim} \Gamma\left(\text{shape} = \alpha, \ \text{scale} = \frac{\exp(X_i^\top \theta)}{\alpha}\right),$$

then the corresponding potential is

$$V(\theta) = \alpha \sum_{i=1}^{\ell} \left(Y_i \exp(-X_i^\top \theta) + X_i^\top \theta\right).$$

Typically, the shape parameter $\alpha$ is either assumed to be known (e.g., $\alpha = 1$ corresponds to exponential regression) or replaced by an estimate. For this experiment, we assume it is known. Notably, due to the exponential dependence on $\theta$, this potential is not $L$-smooth. Consequently, FBGVI (Diao et al., 2023) lacks a theoretical guarantee in this setting, whereas BWAdaGVI continues to provide one, as condition (16) is satisfied.

We chose the same $A_k$, $B_k$, and $\tilde{B}_k$ as in Section 5.1. For the Gaussian target task, we used $V(x) = \frac{1}{2}(x - \mu_{\text{true}})^T \Sigma_{\text{true}}^{-1}(x - \mu_{\text{true}})$ for randomly generated $\mu_{\text{true}}$ and $10^{-7} I \preceq \Sigma_{\text{true}} \preceq I$ on $\mathbb{R}^n \times \mathrm{PSD}(n)$ with $n = 20$. This problem has 1-smooth target log density. The experiment results are aggregated in Figure 2. Note our method outperforms FBGVI even when the task is geodesically $L$-smooth. This is due to the fact that the adaptiveness allows to choose the stepsizes at each step which may not be valid globally but valid locally.

For Gamma regression task, we used $X_i \overset{i.i.d}{\sim} N(0, I_n)$ and $Y_i \mid X_i \overset{ind}{\sim} \Gamma(5, e^{X_i^T \theta}/5)$ for $i = 1, \ldots, 50$ and $n = 25$. Again, Figure 3 shows that our method outperforms FBGVI with various

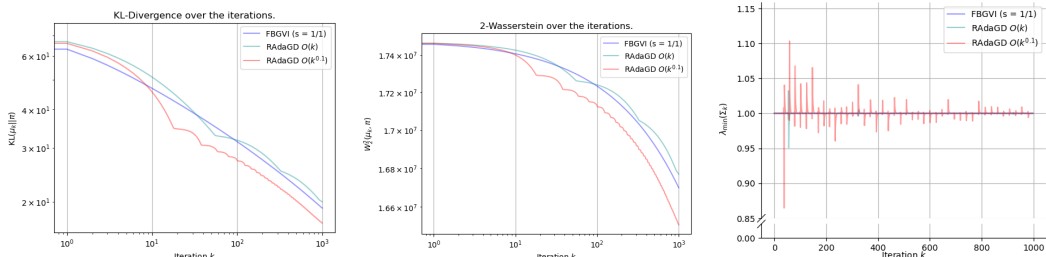

Figure 2: Comparison between Diao et al. (2023) and our methods with different choice of $A_k, B_k$ on Gaussian target $\pi \in \mathrm{BW}(\mathbb{R}^{20})$. **Left**: KL-divergence $\mathrm{D}_{\mathrm{KL}}(\mu_k \,\|\pi)$. **Middle**: Squared 2-Wasserstein distance $W_2^2(\mu_k, \pi)$. **Right**: Minimum eigenvalues $\lambda_{\min}(\Sigma_k)$ over the iterations.

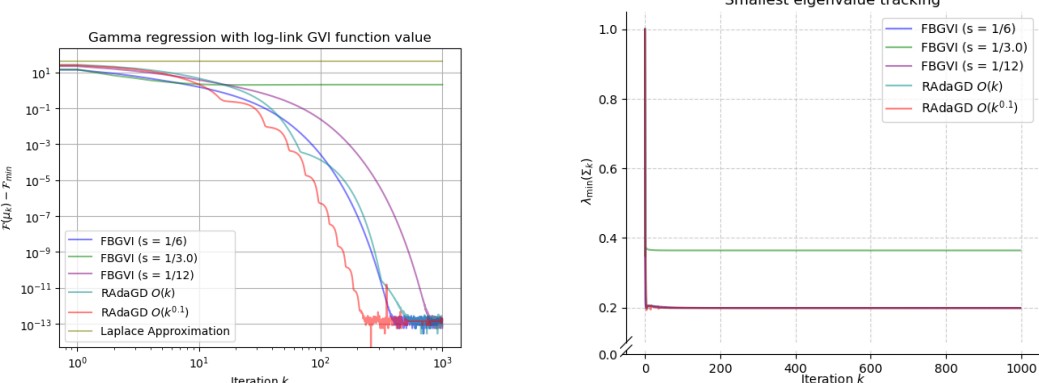

Figure 3: Comparison between FBGVI (Diao et al., 2023) and our methods on Gamma regression target $\pi$ with $\ell = 50, n = 25$. **Left**: $\mathcal{F}(\mu_k) - \mathcal{F}_{\min}$, where $\mathcal{F}_{\min}$ is the minimum value among all experiments. **Right**: Minimum eigenvalues $\lambda_{\min}(\Sigma_k)$ over the iteration.

step size selections $s = 1/3, 1/6, 1/12$. In particular, $1/6$ was in fact the numerically optimal stepsize for FBGVI among our experiments. In addition, we observe that, in both experiments, the smallest eigenvalues remain uniformly bounded away from zero.

### G.1.1 GAUSSIAN VARIATIONAL INFERENCE WITH FISHER-RAO GEOMETRY

On the manifold of non-singular multivariate Gaussian distributions, there exists another natural Riemannian metric besides the Bures-Wasserstein metric: the Fisher-Rao metric. This metric arises from information geometry (Amari, 2016), and its geometric properties have been extensively studied; see, for example, Andai (2007); Nielsen (2024).

Accordingly, one may consider RGD and RAdaGD equipped with the Fisher-Rao metric as alternative algorithms for solving (15). At present, however, only RGD is practically implementable. The obstruction lies in the parallel transport operator under the Fisher-Rao geometry for multivariate Gaussians: it is characterized by a differential equation whose closed-form solution is not tractable (Han & Park, 2014; Lawson et al., 2023). As a result, implementing RAdaGD, which relies on explicit parallel transport, is currently infeasible in this setting.

For the Gaussian variational inference (GVI) problem, the Bures-Wasserstein geometry is theoretically preferable to the Fisher-Rao geometry. The distinction is structural. The objective functional $\mathcal{F}(\mu) := \mathrm{D}_{\mathrm{KL}}(\mu\|\pi)$ exhibits more favorable properties under the Bures–Wasserstein metric—most notably, generalized geodesic convexity for any log-concave target density $\pi$, as established in Lemma 5.2. In contrast, under the Fisher-Rao metric, a strong negative result holds: for any smooth target distribution $\pi$, the functional $\mathcal{F}(\mu) = \mathrm{D}_{\mathrm{KL}}(\mu \,\|\pi)$ is never geodesically convex when considered on the Fisher-Rao manifold of all absolutely continuous probability measures on $\mathbb{R}^d$ with

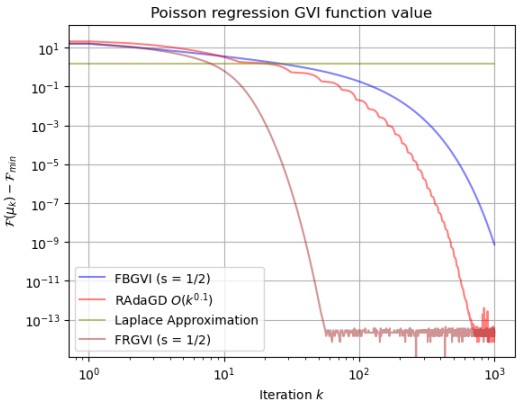

Figure 4: Comparison between Bures-Wasserstein gradient methods and the Fisher-Rao gradient method on Poisson regression target.

smooth densities (Carrillo et al., 2024, Theorem 3.3). Admittedly, this result does not exclude the possibility that geodesic convexity might hold after restricting the domain of $\mathcal{F}$ to Gaussian measures. Nevertheless, it highlights the intrinsic difficulty of establishing geodesic convexity of the KL-divergence under the Fisher-Rao metric. In fact, as can be seen from the case in their proof where both $\mu$ and $\pi$ are univariate Gaussian distributions, geodesic convexity already fails even in this highly structured setting.

Despite the absence of convexity guarantees, our numerical experiments demonstrate strong empirical performance when employing the Fisher-Rao metric for GVI.[4] In this appendix, we report results on the same Poisson regression task studied in Section 5.1, comparing the performance under the Bures-Wasserstein and Fisher-Rao geometries. The results are presented in Figure 4.

The strong empirical performance of the Fisher-Rao metric suggests that developing adaptive methods under this geometry—potentially via approximating the parallel transport—may yield practically effective algorithms. A systematic study of such methods, however, lies beyond the scope of this work. We therefore limit ourselves to reporting this empirical observation and defer a further investigation to future research.

We reiterate, however, that the Bures-Wasserstein metric provides substantially stronger theoretical guarantees, most notably the (generalized) geodesic convexity of the KL-divergence, which is unlikely to hold in most cases under the Fisher-Rao metric. In particular, among the methods compared in Figure 4, BWAdaGVI is the only algorithm that is supported by a corresponding theoretical guarantee at this moment.

### G.2 ADDITIONAL EXPERIMENTS ON THE SPHERE

We additionally demonstrate the numerical strength of our algorithm on Rayleigh quotient maximization problem, a standard benchmark task on non-negatively curved manifold. While this problem is not geodesically convex, RGD methods have shown strong empirical performance (Alimisis et al., 2021; Kim & Yang, 2022). In this regard, we provide the numerical experiments on this problem. Given $H \in \mathrm{Sym}(n)$, Rayleigh quotient maximization problem is written as follows:

$$\min_{x \in \mathbb{S}^{n-1}} f(x) = -\frac{1}{2}x^T H x.$$

The above function $f$ is geodesically $\lambda_{\max}(H) - \lambda_{\min}(H)$-smooth (Kim & Yang, 2022, Proposition 7.1), while not geodesically convex. Hence, we note neither RGD nor RAdaGD provides the theoretical guarantee. For numerical experiment, We considered $n = 1000$ and $H = \frac{1}{2}(B + B^T)$

---

[4]This behavior is not entirely unexpected. Under a favorable initialization, the Fisher-Rao gradient flow of the KL-divergence exhibits some favorable properties than the corresponding Wasserstein gradient flow; see for example Lu et al. (2019; 2023); Domingo-Enrich & Pooladian (2023).

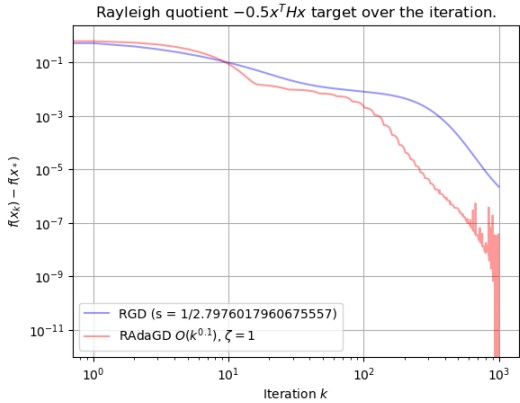

Figure 5: Rayleigh quotient maximization. The stepsize was chosen by $1/L$ with $L = \lambda_{\max}(H) - \lambda_{\min}(H)$.

where the entries of $B \in \mathbb{R}^{n \times n}$ were randomly generated by the Gaussian distribution $N(0, 1/n)$. This setup is the same setup as in Kim & Yang (2022, §7). For RAdaGD, we considered the same instance as in Lemma 5.1. The result is displayed in Figure 5. We observed numerical strength of our method.

### G.3 Additional experiments on Hadamard manifolds

While our primary motivation is the GVI problem, we also demonstrate the applicability of our method on Hadamard manifolds *i.e.*, non-positively curved manifolds. Our algorithm is particularly suitable here: as discussed in Section 3, geodesic smoothness is highly restrictive on Hadamard manifolds, whereas the $C^2$ condition remains general.

For the experiments on Hadamard manifolds, we consider two parameter choices: the benchmark parameters from Lemma 5.1, and an alternative set derived from Corollary 4.8 which can cope with unknown $\zeta$.

**Lemma G.1.** *Assume that $\bar{\zeta} < \infty$. Then $A_k = 2(k+2)^{0.1} + 2$, $\tilde{B}_k = 2(k+2)^{0.1}$, and $B_k = \alpha\tilde{B}_k$ with $\alpha = 0.999$ satisfy the conditions in Corollary 4.8.*

*Proof.* The verification of (10) can be done in almost the same way as in Lemma 5.1, whose proof we provide in Appendix E.1.1. The only difference is that $\lim_{k \to \infty} \left( \frac{A_k}{B_k} + \frac{\tilde{B}_{k+1}}{A_k} \right)^{-1} = \left( \frac{1}{\alpha} + 1 \right)^{-1}$, instead of $\frac{1}{2}$.

What remains is to show that $\lim_{k \to \infty} (\tilde{B}_k - B_k) = \infty$, which is clear since

$$\tilde{B}_k - B_k = (1 - \alpha)\tilde{B}_k = 0.001 \times 2(k+2)^{0.1} \xrightarrow{k \to \infty} \infty.$$

$\square$

#### G.3.1 Optimization on log-determinant function

In this section, we focus on the optimization problem on $\mathrm{SPD}(n)$. While Bures-Wasserstein (BW) metric provides one specific geometry on $\mathrm{SPD}(n)$, many algorithms lose its theoretical guarantee under BW metric due to the geodesic incompleteness of the space (Han et al., 2021; Thanwerdas & Pennec, 2022a; Thanwerdas, 2022). On the other hand, there is another geometry on $\mathrm{SPD}(n)$ called *Affine invariant metric* (Fillard et al., 2005; Pennec et al., 2005). This metric is defined by $d_{AI}(A, B) := \left\| \log A^{-1/2} B A^{-1/2} \right\|_F$, and $\langle S, R \rangle_A = \mathrm{tr}(A^{-1}SA^{-1}R)$ for $S, R \in \mathrm{Sym}(n)$. This metric induces non-positively curved geometry on $\mathrm{SPD}(n)$ with geodesic completeness and the curvature bound $[-1/2, 0]$ (Criscitiello & Boumal, 2023a). There is a family of functions under this geometry where the assumptions of our methods are met.

**Proposition G.2.** *For any $C^2$ convex function $\phi : \mathbb{R} \to \mathbb{R}$, the functional $f_\phi(X) = \phi(\log \det X)$ is locally geodesically smooth and generalized geodesically convex on $(\mathrm{SPD}(n), d_{AI})$.*

*Proof.* First, we consider $g(X) = \log \det X$. We first show for any arbitrary base $B$ and $X, Y \in \mathrm{SPD}(n)$,

$$g(Y) - g(X) = \left\langle \Gamma_X^B \, \mathrm{Grad}\, g(X), \log_B Y - \log_B X \right\rangle. \tag{43}$$

To verify this, first observe that $\mathrm{Grad}\, g(X) = X$. This can be verified by the definition of Riemannian gradient. Riemannain gradient is defined by the operator satisfying for all $H \in \mathrm{Sym}(n)$

$$\mathrm{tr}(X^{-1} \, \mathrm{Grad}\, g(X) X^{-1} H) = \langle \mathrm{Grad}\, g(X), H \rangle_X = dg_X(H) = \mathrm{tr}(X^{-1} H).$$

For the last inequality we used the well-known formula for the derivative of log-determinant function. Since $X \in \mathrm{SPD}(n)$, this implies $\mathrm{Grad}\, g(X) = X$.

Next, we show $\Gamma_X^B X = B$. To check this, from the definition of the parallel transport in $(\mathrm{SPD}(n), d_{AI})$ (Nguyen, 2022, Supplement 1.1),

$$\Gamma_X^B X = (BX^{-1})^{1/2} X ((BX^{-1})^{1/2})^T = B.$$

Hence,

$$\left\langle \Gamma_X^B X, \log_B Y - \log_B X \right\rangle = \langle B, \log_B Y \rangle - \langle B, \log_B X \rangle.$$

Now, for any matrix $M$, from the definition of the Riemannian metric on $d_{AI}$ and logarithmic map (Nguyen, 2022, Supplement 1.1),

$$\langle B, \log_B M \rangle = \mathrm{tr}(B^{-1} \log_B M) = \mathrm{tr}(B^{-1/2} \log(B^{-1/2} M B^{-1/2}) B^{1/2}) = \mathrm{tr}(\log(B^{-1/2} M B^{-1/2}))$$
$$= \log \det(B^{-1/2} M B^{-1/2}) = -\log \det B + \log \det M.$$

Therefore,

$$\left\langle \Gamma_X^B X, \log_B Y - \log_B X \right\rangle = \log \det Y - \log \det X = g(Y) - g(X)$$

which shows (43).

Now we are left with the $\phi$ part. By the convexity of $\phi$,

$$\phi \circ g(A) - \phi \circ g(B) \geq \phi'(g(B))(g(A) - g(B)) = \phi'(g(B)) \left\langle \Gamma_B^C \, \mathrm{Grad}\, g(B), \log_C A - \log_C B \right\rangle$$
$$= \left\langle \Gamma_B^C \, \mathrm{Grad}(\phi \circ g)(B), \log_C A - \log_C B \right\rangle$$

where the last equality is from the chain rule and linearity of the parallel transport and gradient. This completes the proof.

For the local geodesic smoothness, since $(\mathrm{SPD}(n), d_{AI})$ is the smooth and geodesically complete Riemannian manifold (Thanwerdas & Pennec, 2022b), Theorem 3.3 is applicable. As discussed in the proof of Proposition 5.3, the differential structure of the manifold is invariant with the Riemannian metric, and one gets the desired result by the smoothness of $\log \det(\cdot)$. Since $\phi$ is $C^2$ and $\log$ and $\det$ are smooth on $\mathrm{SPD}(n)$, $f_\phi$ is $C^2$. $\quad\square$

$\log \det X$ coincides (up to an affine transformation) with the Gaussian entropy $\mathcal{H}$ discussed in Section 5.1. Hence $f_\phi(X)$ corresponds to the entropy matching or regularization. Such functions frequently appear in composite objectives as the penalty term. While the full composite may not satisfy our theoretical assumptions, the $f_\phi(X)$ part does, and our method has shown strong empirical performance in these cases.

Accordingly, we report three experiments: two directly optimizing $f(X) = |\log \det X - c|^2$ and $f(X) = |\log \det X - c|^4$ for some $c \in \mathbb{R}$, which fully meet our assumptions in Theorem 4.5, and the other optimizing the composite function $f(X) = \mathrm{tr}(XC) - \log \det X$ for some $C \in \mathrm{SPD}(n)$ which, despite weaker theoretical guarantees, has important practical implications as a generalization of linear semidefinite programming as well as the log-likelihood of Wishart distribution (Boyd & Vandenberghe, 2004; Wang et al., 2010; Han et al., 2021). Note $|\log \det X - c|^2$ is geodesically $2n$-smooth, while the other two functions are not geodesically $L$-smooth. The experiments are summarized in Figure 6. Our method achieves either comparable or superior performances and shows robustness to step-size choices.

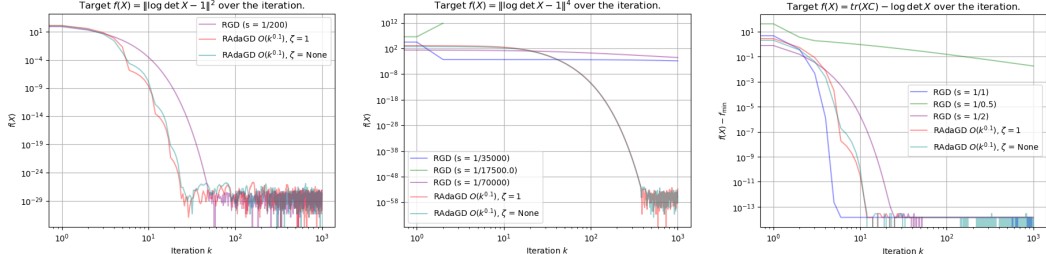

Figure 6: Comparison between RAdaGD and various stepsizes RGD on $\mathrm{SPD}(n)$ for different tasks. We set $n = 50$. We chose the RGD stepsize for non $L$-smooth tasks by the quantities that make the RGD iterates stable. **Left**: $f(X) = |\log \det X - 1|^2$. **Middle**: $f(X) = |\log \det X - 1|^4$. **Right**: $f(X) = \operatorname{tr}(XC) - \log \det X$.

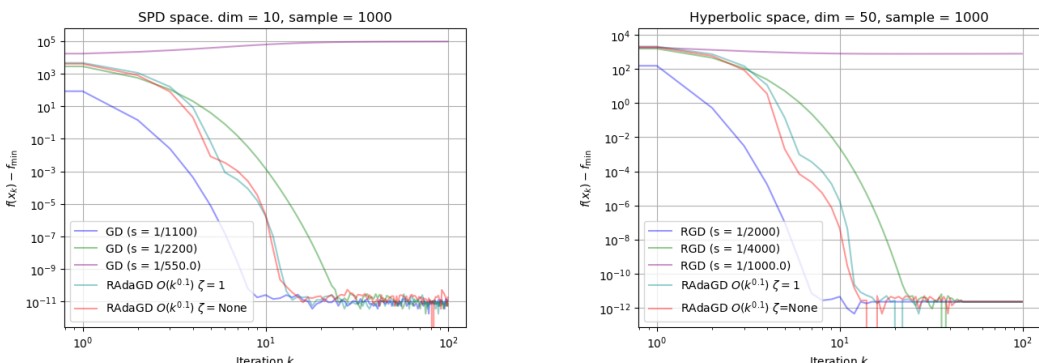

Figure 7: Comparison between RGD with different stepsizes and our method on Fréchet mean estimation problem under different geometries. We chose $L$ that makes the RGD iterates to be well-defined. **Left**: $\mathrm{SPD}(10)$ with 10000 points. **Right**: $\mathbb{H}^{50}$ with 1000 points.

### G.3.2 FRECHET MEAN ESTIMATION

We show the validity of our method on Fréchet mean estimation (or barycenter) problem, which is a standard benchmark in Riemannian optimization literature (Alimisis et al., 2021; Kim & Yang, 2022). Fréchet mean estimation problem is widely studied problem in non-Euclidean statistics. In particular, Fréchet mean imposes favorable statistical properties on non-positively curved spaces (Sturm, 2000; Bačák, 2014b; Yun & Park, 2023; Brunel & Serres, 2024; Kim et al., 2025a). For a hadamard manifold $M$ and for given $\{p_1, \ldots, p_n\} \subset M$, the empirical Fréchet mean over $p_i$ is

$$p_* := \operatorname*{arg\,min}_{x \in M} f(x) := \frac{1}{2} \sum_{i=1}^{n} d^2(x, p_i).$$

This problem is known to be geodesically $n$-strongly convex (Kim & Yang, 2022, Proposition 7.2), while not generalized geodesically convex. On the other hand, this function is $C^2$ and thus locally geodesically smooth, while not geodesically $L$-smooth. Hence, neither RGD nor RAdaGD guarantees the convergence for this problem.

In this regard, we conduct experiments on Fréchet mean estimation problem on Hadamard manifolds. We analyze two experiments: one is $\mathrm{SPD}(n)$, which was what we used in Section 5, and the other is the hyperbolic space. Hyperbolic space is widely used in machine learning for hierarchical representation learning (Nickel & Kiela, 2017), and as a result Fréchet mean on this space is also one of the standard benchmarks (Alimisis et al., 2021; Kim & Yang, 2022). The results are presented in Figure 7. While not outperforming, our method is still comparable to the standard RGD on both tasks.

## H LLM USAGE

This paper made use of Large Language Models (LLMs) primarily for editing grammar, spelling, word choice, and sentence clarity. In addition, LLMs were used occasionally to help with minor coding tasks during the experiments.

