# OpenReview forum: "Adaptive gradient descent on Riemannian manifolds and its applications to Gaussian variational inference"
_ICLR.cc/2026/Conference — ICLR 2026 Poster_

### Official Review · Reviewer_eAPu · 2025-10-29

**Soundness:** 4
**Presentation:** 3
**Contribution:** 3
**Rating:** 8
**Confidence:** 4

**Summary:**

The paper proposes RAdaGD, a family of adaptive gradient descent algorithms on Riemannian manifolds. The method generalizes Euclidean adaptive step-size rules to the manifold setting, avoiding the need for a known smoothness constant or line search. The authors prove non-ergodic O(1/k) convergence under local geodesic smoothness and generalized geodesic convexity, relaxing the assumption of global smoothness. They apply RAdaGD to Gaussian Variational Inference (GVI) and provide convergence guarantees even when the target log-density is not globally smooth.

**Strengths:**

The paper makes a novel theoretical contribution by defining local geodesic smoothness and proving that it holds for all twice-differentiable functions on complete manifolds, thereby relaxing the need for global smoothness assumptions. It introduces generalized geodesic convexity as a principled framework for analyzing adaptive Riemannian optimization methods and provides the first adaptive Riemannian gradient algorithm with non-ergodic O(1/k) convergence rates. The application to Gaussian Variational Inference is well motivated and theoretically sound, extending results from the Euclidean to the non-Euclidean setting. Overall, the theoretical analysis is rigorous and carefully adapts adaptive step-size techniques to curved spaces.

**Weaknesses:**

The paper’s main weaknesses lie in its presentation and empirical scope. The exposition is mathematically dense and at times difficult to follow, with heavy notation and definitions that could be streamlined for clarity. Empirical validation is limited to a single Poisson regression experiment, which does not fully demonstrate the algorithm’s practical benefits or robustness across different manifolds. Comparisons with other Riemannian adaptive or stochastic methods, such as Riemannian Adam, are missing, leaving open questions about relative performance. Some theoretical assumptions, including bounded curvature and known constants, may be hard to verify or implement in practice, and the computational cost of adaptive step-size evaluations is not thoroughly discussed. Overall, while the theory is strong, the work would benefit from clearer exposition and more extensive empirical evidence.

**Questions:**

1. Can the authors clarify how restrictive the local geodesic smoothness assumption is in practice? For example, are there realistic manifolds or objective functions where it might fail to hold, and how would RAdaGD behave in such cases?

2. As I've understood it, the experiments are limited to Poisson regression on the Bures–Wasserstein manifold. Could the authors provide results or discussion on how RAdaGD performs on other manifolds (e.g., SPD or hyperbolic spaces) or compare it directly to Riemannian Adam or stochastic methods?

3. How sensitive is RAdaGD to the choice of initial step size or curvature estimates in practice, and what is the computational overhead of computing the adaptive step-size parameters compared to standard Riemannian gradient descent?

---

> ### Author Response · Authors · 2025-11-21
>
> We thank the reviewer for their high evaluation of our paper. We are glad that they found our theoretical contribution based on local geodesic smoothness to be meaningful. We also thank the reviewer for their valuable feedback and will carefully consider it when preparing the revised version.
>
> > The exposition is mathematically dense and at times difficult to follow, with heavy notation and definitions that could be streamlined for clarity.
>
> We will take more careful look and edit the draft to improve the clarity.
>
> > Empirical validation is limited to a single Poisson regression experiment, which does not fully demonstrate the algorithm’s practical benefits or robustness across different manifolds.
>
> We have included additional experiments in Appendix G, which includes different GVI targets as well as different manifolds.
>
> > Comparisons with other Riemannian adaptive or stochastic methods (e.g., Riemannian Adam) are missing, leaving open questions about relative performance.
>
> Since the experiments in Section 5 focus on the Bures-Wasserstein Gaussian Variational Inference (BWGVI) problem, we compared our method with algorithms that are specifically designed for BWGVI. As discussed in Section 5, obtaining theoretical guarantees for BWGVI is more subtle than for plain Riemannian GD algorithms because the underlying space is not geodesically complete. As a result, BWGVI implementations are typically not identical to standard RGD. For example, our algorithm requires an additional stepsize modification as described in Equation (17) to cope with this geodesic incompleteness. Given these considerations, we think that comparisons with BWGVI algorithms provide a more appropriate and fair experimental setting.
>
> Regarding the Riemannian Adam method mentioned by the reviewer, our work follows the perspective of [1], which explicitly distinguishes their approach from Adam-type optimizers (i.e., methods with nonincreasing step sizes). We discuss this in Appendix A.2 (Lines 829–833) and footnote 3 (Line 913). Since several works in this line of research similarly do not compare against Adam-type methods [1,2,3,4], we followed the same practice.
>
> For the stochastic gradient setting, we are indeed considering a stochastic extension as future work. Since the stochastic gradient problem has a different nature, we focused on experiments with deterministic gradients in this paper.
>
> > Some theoretical assumptions, including bounded curvature and known constants, may be hard to verify or implement in practice.
>
> This is true, and this is in fact what many previous Riemannian optimization papers were suffering as well [5, 6]. We had the same concern with the reviewer's one when we work on this paper, and that is why we introduced Corollary 4.7 and 4.8; Corollary 4.7 reduces the bound on the entire diameter to the bound on a single iterate. Corollary 4.8 partially get over the restriction the reviewer mentioned, by demonstrating at least the asymptotic convergence.
>
> > The computational cost of adaptive step-size evaluations is not thoroughly discussed.
>
> The reason we did not discussed the computational cost of RAdaGD is mainly because it is hard to provide the general analyses for it. The computational costs of RAdaGD depends on the underlying geometry, as the added computations are computing the Riemannian metric and parallel transport (parts computing $L_{k+1}$).
>
> That said, at least the Bures-Wasserstein GVI application in Section 5 (BWAdaGVI) has the same computational complexity as the plain Bures-Wasserstein GVI. The only extra steps of BWAdaGVI, compared with plain Bures-Wasserstein GVI, are the computations of $\hat L_{k+1}$ and $s_{k+1}$ in Lines 1933 and 1937. These added procedures require only matrix multiplications, trace evaluations, and computing the maximum singular value. All of these operations have computational complexity no worse than that of the Bures-Wasserstein gradient computation itself (Line 1927), which already involves a matrix inversion. Therefore, the extra operations in BWAdaGVI do not increase the overall asymptotic computational complexity relative to the standard Bures-Wasserstein GVI method.
>
> We will include this computational complexity analysis of BWAdaGVI in the revision.

---

> ### Author Response · Authors · 2025-11-21
> **Response (continued)**
>
> > 1. Can the authors clarify how restrictive the local geodesic smoothness assumption is in practice? For example, are there realistic manifolds or objective functions where it might fail to hold, and how would RAdaGD behave in such cases?
>
> As demonstrated in Theorem 3.3 and the entailed discussions, the local geodesic smoothness is not restrictive and is very general in fact. The most of the instances we encounter in practice are $C^2$ functions (e.g., all instances in [7] are $C^2$).
>
> While local geodesic smoothness is not restrictive, generalized geodesic convexity could limit the range of the function class.
> However, even without this generalized geodesic convexity, our algorithm displays empirical strength at least for some standard benchmark experiments, which we have already included in Appendix G.
>
> > 2. As I've understood it, the experiments are limited to Poisson regression on the Bures–Wasserstein manifold. Could the authors provide results or discussion on how RAdaGD performs on other manifolds (e.g., SPD or hyperbolic spaces) or compare it directly to Riemannian Adam or stochastic methods?
>
> We have included the experiments on Gamma regression GVI, SPD space, and hyperbolic spaces in Appendix G. For the Riemannian Adam and stochastic method, we refer to the answer we made in the above.
>
> > 3. How sensitive is RAdaGD to the choice of initial step size or curvature estimates in practice, and what is the computational overhead of computing the adaptive step-size parameters compared to standard Riemannian gradient descent?
>
> RAdaGD is very robust to the choice of the initial stepsize as well as the curvature estimate (more precisely, the choice of $\zeta$). For the curvature estimate, we have included Figure 5: Here, theoretically $\zeta =$None instance is the proper curvature estimate (the instance of Corollary 4.8), but one can see its behavior is similar to $\zeta = 1$ case (wrong curvature estimate).
>
> For the initial stepsize, while we did not include the figure, even if we set the initial stepsize very small (to make the conservative choice), for every experience within 10-20 iterates the stepsize pattern becomes similar to the usual behavior. If needed, we can include the comparison on the robustness to the initial step size choice.

---

> ### Author Response · Authors · 2025-11-21
> **References**
>
> **References**
>
> [1] Malitsky and Mishchenko. Adaptive gradient descent without descent. ICML 2020.
>
> [2] Oikonomidis et al. Adaptive proximal gradient methods are universal without approximation. ICML 2024.
>
> [3] Malitsky and Mishchenko. Adaptive proximal gradient method for convex optimization. NeurIPS 2024.
>
> [4] Zhou et al. AdaBB: Adaptive Barzilai-Borwein method for convex optimization. Mathematics of Operations Research, 2025.
>
> [5] Zhang and Sra. First-order Methods for Geodesically Convex Optimization. COLT 2016.
>
> [6] Kim and Yang. Accelerated Gradient Methods for Geodesically Convex Optimization: Tractable Algorithms and Convergence Analysis. ICML 2022.
>
> [7] Han et al. On Riemannian optimization over positive definite matrices with the Bures-Wasserstein geometry. NeurIPS 2021.

---

### Official Review · Reviewer_Jjvv · 2025-10-30

**Soundness:** 3
**Presentation:** 3
**Contribution:** 2
**Rating:** 4
**Confidence:** 3

**Summary:**

This paper proposes a adaptive optimization method, RAdaGD, for smooth optimization problems defined on Riemannian manifolds. The authors extend adaptive gradient methods to the non-Euclidean setting under relaxed smoothness and convexity assumptions—specifically, local geodesic smoothness and generalized geodesic convexity—and provide a non-ergodic $\mathcal{O}(1/k)$ convergence rate. The approach is further applied to Gaussian Variational Inference (GVI) problems formulated on the Bures–Wasserstein manifold of Gaussian distributions. However, this paper heavily relies on the algorithms and theories presented in Suh & Ma (2025). Furthermore, several theoretical assumptions and proof steps require clarification or stronger justification.

**Strengths:**

1.	The authors propose a family of Riemannian adaptive gradient descent algorithms, RAdaGD, which automatically adjusts the step size by approximating the local smoothness parameters.
2.	The introduction of local geodesic smoothness and generalized geodesic convexity provides a new theoretical framework that generalizes existing smoothness/convexity assumptions.
3.	The application to Gaussian Variational Inference under the Bures–Wasserstein geometry is well motivated and could be useful in practice.

**Weaknesses:**

1.	The main theoretical results (e.g., Theorem 4.1) rely on an upper bound $L \geq L_k$ for $k \geq 0$. While the authors address the existence of such a bound, this assumption obscures the distinction between global and local smoothness, thereby reducing the theoretical appeal of the proposed framework.
2.	The constants in the $\mathcal O{(1/k)}$ rate (e.g., dependence on $r, L, R, \bar \zeta$) should be discussed. If these constants can be large in curved manifolds, the theoretical rate may not be meaningful in practice.
3.	Theorem 3.3 claims that all $C^2$ functions are locally geodesically smooth. While intuitively true, the proof relies on the boundedness of the Riemannian Hessian. The manuscript should more explicitly state the curvature-dependence of this claim.
4.	The authors mentioned another adaptive gradient descent algorithm (Ansari-O¨ nnestam & Malitsky (2025)) on a Riemannian manifold in the previous text , but did not conduct a comparison in the experiments. The authors claim that the paper was withdrawn from arXiv on September 15, 2025, yet its V1 remains accessible for viewing.
5.	In Corollary 5.4, the author claims that $\bar \zeta =1$, based on the assumption in Corollary 4.6 that $\bar \zeta =sup_{x \in N} \zeta(d(x,x_*))$, but I have not found a proof for $\zeta = 1$.

**Questions:**

1.	The main theoretical results (e.g., Theorem 4.1) rely on an upper bound $L \geq L_k$ for $k \geq 0$. While the authors address the existence of such a bound, this assumption obscures the distinction between global and local smoothness, thereby reducing the theoretical appeal of the proposed framework.
2.	The constants in the $\mathcal O{(1/k)}$ rate (e.g., dependence on $r, L, R, \bar \zeta$) should be discussed. If these constants can be large in curved manifolds, the theoretical rate may not be meaningful in practice.
3.	Theorem 3.3 claims that all $C^2$ functions are locally geodesically smooth. While intuitively true, the proof relies on the boundedness of the Riemannian Hessian. The manuscript should more explicitly state the curvature-dependence of this claim.
4.	The authors mentioned another adaptive gradient descent algorithm (Ansari-O¨ nnestam & Malitsky (2025)) on a Riemannian manifold in the previous text , but did not conduct a comparison in the experiments. The authors claim that the paper was withdrawn from arXiv on September 15, 2025, yet its V1 remains accessible for viewing.
5.	In Corollary 5.4, the author claims that $\bar \zeta =1$, based on the assumption in Corollary 4.6 that $\bar \zeta =sup_{x \in N} \zeta(d(x,x_*))$, but I have not found a proof for $\zeta = 1$.

---

> ### Author Response · Authors · 2025-11-21
>
> We are glad that the reviewer found our algorithm RAdaGD meaningful and its application to GVI under the BW geometry “is well motivated and could be useful in practice.” We are grateful to the reviewer for their detailed feedback, which we will take into account when preparing the revised version.
> We address the questions below.
>
> > 1. The main theoretical results (e.g., Theorem 4.1) rely on an upper bound $L \geq L_k$ for $k \geq 0$. While the authors address the existence of such a bound, this assumption obscures the distinction between global and local smoothness, thereby reducing the theoretical appeal of the proposed framework.
>
> We thank the reviewer for the detailed question, which gives us the opportunity to clarify this point.
> Our algorithm does have a theoretical guarantee that makes the local smoothness assumption meaningful.
> There are two aspects we would like to address in response to this question.
>
> The first point is that $L$ is not a global quantity determined by the function $f$. Rather, $L$ is a local smoothness parameter defined on a ball determined by the initial point $x_0$. This is because we only use local information within the region that contains all the iterates.
>
> The second point is that $L$ is actually a very conservative bound used for simplicity. For example, the convergence rate of Theorem 4.5 (i) is based on the inequalities $s_{k+1} A_k (f(x_k) - f_\star) \leq \frac{1}{2} R$ in Line 1652 and $s_k \geq r\min\left\\{ \frac{1}{L_0}, \frac{1}{L_1}, \dots, \frac{1}{L_k} \right\\}$ in Line 1614.
> Thus, we could in fact obtain a tighter bound
> $$f(x_k) - f_\star \le \max\\{ L_0, L_1, \dots, L_k \\}  \frac{R}{2rA_k}.$$
> In short, the results can be rewritten in terms of the finite number of local parameters $L_i$.
>
> > 2. The constants in the $O(1/k)$ rate (e.g., dependence on $r, L, R, \bar \zeta$) should be discussed. If these constants can be large in curved manifolds, the theoretical rate may not be meaningful in practice.
>
> We would like to address this point based on the parameter selection that yields the tightest convergence rate. Consider the parameters of Theorem 4.1 with $\alpha = 1$. From the definition of $L$ in the statement, we have $||\mathrm{Grad} f(x_0)|| \le L^2 d(x_0, x_\star)^2$.
> Then
> $$
> r = \frac{1}{3+\bar{\zeta}}, \quad R^2 \le d(x_0,x_\star)^2\left( 1 + \frac{(2+\bar{\zeta})^2 (1+\bar{\zeta})}{(3+\bar{\zeta})^2} \frac{L^2}{L_0^2} \right).
> $$
> For simplicity, consider the case that $f$ is $\bar{L}$-smooth.
> If we had access to $\bar{L}$, we could simply set $L_0 = \bar{L}$.
> Note that $L \le \bar{L}$, since $L$ is the local smoothness parameter on the closed ball defined in Theorem 4.1.
> Applying these to inequality (9), we obtain (there was a typo in the current draft--the $R$ in the bound should be changed to $R^2$):
> $$
> f(x_k) - f_\star \le \frac{3+\bar{\zeta}}{2(k+2+\bar{\zeta})} \left(1 +  \frac{(2+\bar{\zeta})^2 (1+\bar{\zeta})}{(3+\bar{\zeta})^2} \right) L d(x_0,x_\star)^2.
> $$
>
> To check the optimality of the given constant factor, and in particular the dependency on the curvature, we can compare our result to the non-adaptive RGD [1], where they give the analysis on Hadamard manifold (manifold with non-positive curvature). [1, Theorem 13] shows that if $f$ is g-convex and $L$-smooth on a bounded set $N \subseteq M$ with diameter $D$, then (projected) RGD with a fixed step size $s_k = 1/L$ achieves the convergence rate
> $$
> f(x_k) - f_\star \leq \frac{\bar \zeta L }{2 (k - 1 + \bar \zeta)} d(x_0, x_\star)^2, \quad \text{for all } k \geq 1.
> $$
>
> When comparing this non-adaptive RGD result on Hadamard manifolds with our method, our convergence rate includes an additional constant factor of order $O(\bar \zeta)$ constant factor. This factor is harmless when the manifold has nonnegative curvature (in which case $\bar \zeta = 1$, and only the 17/2 constant appears), but as the reviewer correctly noted, it may become large on manifolds with strong negative curvature. This raises the question of whether the extra factor is an unavoidable price for achieving adaptivity, or simply an artifact of a loose analysis. At the moment, unfortunately we do not know the answer. Determining whether this constant can be tightened or whether it represents an inherent cost of adaptivity would be an interesting direction for future work.
>
> We thank the reviewer for highlighting this issue and will add a remark discussing the potential suboptimality of this constant factor in the revised version.
>
> > 3. Theorem 3.3 claims that all $C^2$ functions are locally geodesically smooth. While intuitively true, the proof relies on the boundedness of the Riemannian Hessian. The manuscript should more explicitly state the curvature-dependence of this claim.
>
> The boundedness of the Riemannian Hessian comes from the fact that $f \in C^2(M)$, and it does not involve any curvature information of the manifold.

---

> ### Author Response · Authors · 2025-11-21
> **Response (continued)**
>
> > 4. The authors mentioned another adaptive gradient descent algorithm (Ansari-O¨nnestam & Malitsky (2025)) on a Riemannian manifold in the previous text, but did not conduct a comparison in the experiments. The authors claim that the paper was withdrawn from arXiv on September 15, 2025, yet its V1 remains accessible for viewing.
>
> We had in fact conducted experiments comparing our method with [2] before it was withdrawn. In fact, our current codes in supplementary materials already include the code for the comparison with [2], and its performance was comparable to ours. However, our algorithm offers stronger theoretical guarantees, as [2] does not address geodesically incomplete manifolds and considers only nonnegatively curved settings. While we are open to including these comparative results in the paper, Reviewer jiEp in fact suggests removing the references to [2]. To reconcile both reviewers' comments, we propose leaving the experiments unchanged, while providing additional implementation codes with instructions in the supplementary material that enables readers to reproduce comparisons with [2] if desired.
>
> > 5. In Corollary 5.4, the author claims that $\bar \zeta = 1$, based on the assumption in Corollary 4.6 that $\bar \zeta = \sup_{x \in N}\zeta(d(x, x_*))$, but I have not found a proof for $\zeta = 1$.
>
> We thank the reviewer for raising this question. In Corollary 5.4, we work in the Bures–Wasserstein (BW) space, which is known to have nonnegative curvature; thus $\bar \zeta = 1$. This also follows directly from Lemma F.5. Our current submission addresses this in Lines 1241 and 2289. We originally thought this point is clear as in the proof of Corollary 5.4, we explicitly mentioned the use of Lemma F.5 in Line 1968. However, we recognize that this detail may be easy to miss and is worth emphasizing. We will revise the manuscript to make this point more transparent.

---

> ### Author Response · Authors · 2025-11-21
> **References**
>
> **References**
>
> [1] Zhang and Sra. First-order Methods for Geodesically Convex Optimization. COLT 2016.
>
> [2] Ansari-Onnestam and Malitsky. Adaptive gradient descent on Riemannian manifolds with nonnegative curvature (v1). Arxiv 2025.

---

### Official Review · Reviewer_j1ST · 2025-11-05

**Soundness:** 2
**Presentation:** 2
**Contribution:** 2
**Rating:** 6
**Confidence:** 4

**Summary:**

This paper introduces RAdaGD, a novel family of adaptive gradient descent methods on Riemannian manifolds. The method adapts the step size automatically—without explicit line search—by estimating a local smoothness parameter, achieving the first known non-ergodic convergence rate \mathcal{O}(1/k) under local geodesic smoothness and generalized geodesic convexity. The authors extend adaptive optimization concepts from the Euclidean setting (e.g., Suh & Ma, 2025) to Riemannian manifolds via nontrivial generalizations of Lyapunov analyses, parallel transport arguments, and curvature-dependent scaling factors.
A notable application is given to Gaussian Variational Inference (GVI), where the method provides the first convergence guarantee without assuming global L-smoothness of the target log-density. Empirical experiments confirm the theoretical results and show competitive or superior performance compared to existing algorithms.

**Strengths:**

The paper is rigorous, technically elegant, and addresses a clear theoretical gap in Riemannian optimization—namely, the lack of adaptive step-size methods with formal convergence guarantees. The introduction of local geodesic smoothness as a relaxation of traditional L-smoothness is both natural and powerful, broadening the class of admissible objective functions while preserving tractability.
The convergence proofs are carefully executed and extend nontrivial elements from Euclidean adaptive methods to manifold geometry (e.g., dealing with exponential/log maps, curvature terms, and parallel transport). The theoretical guarantees (non-ergodic \mathcal{O}(1/k) rate) are significant, and the paper connects the framework elegantly to practical applications in Bayesian inference. The exposition is also exemplary—clear definitions, consistent notation, and intuitive motivation for the introduced assumptions.

**Weaknesses:**

While the paper fills a critical theoretical gap, I have a few questions regarding its positioning and broader implications:
	1.	Relation to recent Riemannian adaptive methods.
The paper positions itself as the first of its kind for adaptive algorithms on manifolds. However, a recent arXiv preprint (link https://arxiv.org/abs/2306.16617) also studies adaptive Riemannian optimization with curvature-aware preconditioning and adaptive metric scaling. Could the authors clarify whether RAdaGD fundamentally differs from these approaches (e.g., in the use of local smoothness estimation versus curvature-adapted preconditioning)? Are the convergence guarantees directly comparable, or do they apply to distinct settings (e.g., deterministic vs. stochastic, local vs. global smoothness)?
	2.	Tightness of assumptions.
The local geodesic smoothness assumption, while weaker than L-smoothness, still presupposes control of curvature-dependent Lipschitz constants L_K. Is it possible to relax this further, for example to Hölder-smooth functions, or to obtain adaptive control of curvature-dependent constants during optimization?
	3.	Empirical scope.
The experiments are clean but limited to Gaussian Variational Inference tasks. Given the generality of the method, could the framework extend naturally to Riemannian neural optimization or hyperbolic embeddings (where curvature varies strongly)? A short experiment in such a context could enhance the impact.
	4.	Adaptivity mechanism.
The paper introduces adaptive sequences A_k, B_k, \tilde{B}_k, but it would be helpful to clarify whether the adaptivity primarily tracks local smoothness, curvature, or both. In other words, is the method “geometry-aware” in practice, or purely driven by gradient variance?

Overall, these are minor concerns—the theoretical contribution is strong and original.

**Questions:**

Please comment about the weaknesses

---

> ### Author Response · Authors · 2025-11-21
>
> We are glad that the reviewer considers our paper "rigorous" and "technically elegant," and we thank them for their positive evaluation of the theoretical contribution of our adaptive algorithm. We are also grateful to the reviewer for their detailed feedback, which we will carefully consider when revising the paper.
>
> > 1. **Relation to recent Riemannian adaptive methods.** The paper positions itself as the first of its kind for adaptive algorithms on manifolds. However, a recent arXiv preprint (link: \url{https://arxiv.org/abs/2306.16617}) also studies adaptive Riemannian optimization with curvature-aware preconditioning and adaptive metric scaling. The authors should clarify how RAdAGD fundamentally differs from these approaches (e.g., in the use of local smoothness estimation versus curvature-adapted preconditioning). Are the convergence guarantees directly comparable, or do they apply to distinct settings (e.g., deterministic vs. stochastic, local vs. global smoothness)?
>
> We thank the reviewer for pointing out a valuable reference. This paper seems worth mentioning in our paper.
>
> However, we would like to clarify that our claim of being the first of its kind regarding the non-ergodic convergence rate of the function value, $f(x_k) - f(x_\star) \le \mathcal{O}(1/k)$, remains valid.
>
> In short, the mentioned paper considers a (geodesic) strongly monotone setup (Assumption 4.8, [1]) and uses the gradient norm as a performance metric (Theorem 4.9, [1]). Therefore, this paper differs from ours in both the function (operator) class and the performance metric. While monotone operators are often considered a generalized class, since they cover gradients of convex functions, [1] considers a $\mu$-strongly monotone and (globally) $\ell$-smooth operator. Since our paper does not consider $\mu$-strongly convex functions, the setup in [1] does not apply to our work.
>
> In addition, the adaptive estimation of $\hat\mu_t \approx \mu$ and $\hat\ell_t \approx \ell$ in [1, Algorithm 2] relies on a line-search procedure that can incur restart costs per iteration, whereas our approach avoids this, leading to a lower per-iteration cost.
>
> We will add this to the discussion in the revised version.
>
> > 2. **Tightness of assumptions.** The local geodesic smoothness assumption, while weaker than \(L\)-smoothness, still presupposes controlled curvature-dependent Lipschitz constants \(L_k\). Is it possible to relax this further, for example to Hölder-smooth functions, or to obtain adaptive control of curvature-dependent constants during optimization?
>
> We believe that extending the method to Hölder-smooth functions would be a possible and interesting direction of extension. In [2], they consider an accelerated adaptive method called AC-FGM, which uses the same approximation $L_k$ as ours but in Euclidean space. While their primary focus is on the locally smooth convex setting, they also extend their analysis to the H\"older-smooth case by introducing a variant of $L_k$ (Equation 3.8 in [2]). We expect that a similar variant could also be developed for our RAdaGD.
>
> > 3. **Empirical scope.** The experiments are clean but limited to Gaussian Variational Inference tasks. Given the generality of the method, could the framework extend naturally to Riemannian neural optimization or hyperbolic embeddings (where curvature varies strongly)? A short experiment in such a context could enhance the paper’s impact.
>
> We provided additional experiments on Appendix G. The experiments include both theoretically guaranteed tasks (Appendix G.3.1), as well as standard benchmark experiments (Appendix G.2, G.3.2). These experiments demonstrate the empirical effectiveness of our method.

---

> ### Author Response · Authors · 2025-11-21
> **Response (continued)**
>
> > 4. **Adaptivity mechanism.** The paper introduces adaptive sequences \(A_k, B_k, \tilde{B}_k\), but it would be helpful to clarify whether the adaptivity primarily tracks local smoothness, curvature, or both. In other words, is the method truly “geometry-aware” in practice, or purely driven by gradient variance?
>
> Basically, all these $A_k$, $B_k$, and $\tilde{B}_k$ are designed carefully to satisfy the inequalities in (23), so that the Lyapunov analysis is valid, which is the heart of the convergence analysis.
>
> For more high-level idea, the local smooth parameter is estimated in Equation (5) as $L_{k+1}$. Then, the coefficient $r_k^L$, which is constructed using $A_k, B_k$, and $\tilde B_k$, adjusts the magnitude of $1/L_{k+1}$, corresponding to Equation (6). This adjustment is something we can (at least partially) control, via the choice of $A_k, B_k$, and $\tilde B_k$ in the algorithm. The geometric information (curvature, distance bound) is used when constructing $r_k^L$, via $\tilde B_k$. $\tilde{B}_k$ reflects the geometric information of the manifold by involving $\bar \zeta$ in Equation (12). Larger $\tilde{B}_k$ means the algorithm will be able to cope with more manifolds. This coincides with our intuition. If we choose very large $\tilde{B}_k$ so that Equation (12) becomes trivial, this implies we have chosen very small stepsizes. Then, under this very small stepsize, the manifold information is insignificant, as locally the manifold will just be almost an Euclidean. On the other hand, the larger the $\tilde{B}_k$, the smaller the $r$, and as a result we get the worse convergence rate $O(L/(rA_k))$. As a result, it is preferrable to choose the tightest $\tilde{B}_k$, which correspond to set $\tilde{B}_k$ to reflect the geometry properly (relavant to $\bar \zeta)$.

---

> ### Author Response · Authors · 2025-11-21
> **References**
>
> **References**
>
> [1] Cai et al. Last-Iterate Convergence of Adaptive Riemannian Gradient Descent for Equilibrium Computation. Arxiv 2023.
>
> [2] Li and Lan. A simple uniformly optimal method without line search for convex optimization. Mathematical Programming, 2025.

---

### Official Review · Reviewer_LVsh · 2025-11-05

**Soundness:** 3
**Presentation:** 3
**Contribution:** 3
**Rating:** 6
**Confidence:** 3

**Summary:**

The paper proposes RAdaGD, a line-search-free family of adaptive gradient descent methods on general Riemannian manifolds. The method adapts the stepsize by estimating a local smoothness quantity from two consecutive iterates and gradients, and it is analyzed under local geodesic smoothness and generalized geodesic convexity, which are weaker than the usual global geodesic $L$-smoothness. The authors prove a non-ergodic convergence rate
$$
f(x_k)-f(x^\star) \le O(1/k)
$$
matching classical Riemannian gradient descent with known $L$, and also give bounds on $|\operatorname{Grad} f(x_k)|^2$. A technically involved part of the analysis is the treatment of curvature via the parameter $\zeta(\cdot)$ and the triad $(A_k,B_k,\tilde B_k)$ in the Lyapunov function. As a main application, the paper shows how to instantiate RAdaGD for Gaussian variational inference on the Bures–Wasserstein manifold, obtaining (to the authors’ knowledge) the first convergence guarantee for GVI without assuming global $L$-smoothness of the log-density, under an additional eigenvalue-boundedness condition. Experiments on Poisson regression GVI and negatively curved spaces demonstrate competitiveness with prior Riemannian and BW-based optimizers.

**Strengths:**

* A concrete, implementable Riemannian adaptive gradient method (RAdaGD) that removes line search and still attains $O(1/k)$ in function value.
* Relaxation from global geodesic $L$-smoothness to local geodesic smoothness, supported by the statement that every $C^2$ function on a complete Riemannian manifold is locally geodesically smooth; this enlarges the class of admissible objectives.
* Careful curvature-aware analysis via the $\zeta(\cdot)$ parameter and the triplet $(A_k,B_k,\tilde B_k)$, including extensions to unbounded or unknown curvature.
* Application to Gaussian variational inference on the Bures–Wasserstein manifold, yielding (under an eigenvalue assumption) what appears to be the first convergence guarantee for non-$L$-smooth GVI.
* Empirical results on Poisson regression GVI that illustrate the claimed setting where prior methods lack guarantees but RAdaGD still converges.

**Weaknesses:**

* The core convergence theorems require generalized geodesic convexity, which is stronger than plain geodesic convexity and may limit applicability on general manifolds or nonconvex statistical objectives.
* The method still assumes access to exponential maps and parallel transport on the manifold; this is reasonable for BW$(\mathbb R^n)$ but can be a practical limitation on more general manifolds where only retractions and vector transports are available.
* The step-size rule in Algorithm 1 depends on quantities that must be computed from two consecutive gradients and parallel transports; while explicit, it is more involved than standard RGD and the paper could more clearly quantify this overhead.
* Experiments are focused on a few GVI scenarios and do not yet explore larger-scale or stochastic-gradient settings, which are natural for VI in practice.
* The GVI convergence result needs an additional uniform lower bound on the covariance eigenvalues, which is plausible but not guaranteed by the algorithm itself.

**Questions:**

1. Can the authors clarify whether the adaptive rule for $s_{k+1}$ can be made compatible with retractions/vector transports (as opposed to exact $\exp$ and parallel transport) while preserving the rates, possibly with higher-order error terms?
2. In the BW-GVI application, the extra step-size clipping in equation (17) depends on the largest eigenvalue of an expectation involving $\nabla^2 V(X)$. How is this implemented in practice when only Monte Carlo estimates of expectations are available, and how sensitive is convergence to this clipping?
3. The convergence proof in Section 4 relies on the compactness/boundedness of the iterates (via the Lyapunov argument). Is there an example of a nonnegatively curved manifold and locally geodesically smooth, generalized geodesically convex function where RAdaGD would fail without this boundedness step?
4. For the Poisson regression GVI example, can the authors provide a small ablation showing RAdaGD vs. a Riemannian method with backtracking line search, to separate the benefit of adaptivity from the benefit of avoiding line search?
5. The paper mentions possible stochastic variants. Are there obstacles in controlling the estimate of $L_{k+1}$ in (5) under stochastic gradients, or is it mainly a matter of concentration bounds?

---

> ### Author Response · Authors · 2025-11-21
>
> We thank the reviewer for evaluating our paper positively and for carefully noting that our contribution includes extensions to unbounded or unknown curvature. We are grateful to the reviewer for their detailed and meaningful feedback, which we will take into account when preparing the revised version.
>
> > The core convergence theorems require generalized geodesic convexity, which is stronger than plain geodesic convexity and may limit applicability on general manifolds or nonconvex statistical objectives.
>
> We agree with the reviewer that generalized geodesic convexity is stronger than usual geodesic convexity. That said, to advocate ourselves, we would like to clarify a few things.
>
> First, while the generalized geodesic convexity seems restrictive, we identified non-trivial and practically motivated instances that satisfies the assumption (Section 5, Proposition G.2). In fact, the Gaussian variational inference problem was our main motivation to come up with this approach. We think these concrete applications showcase our method to be practical and applicable.
>
> Second, the need of the generalized geodesic convexity comes from the derivation of Riemannian version of co-coercivity type inequality, as in Proposition 3.4. In fact, this co-coercivity type is what we really need, and the generalized geodesic convexity is only the sufficient condition for this. If one can weaken the condition of co-coercivity inequality (which we are unaware of unfortunately), then it is possible to extend our method to more general setup as well. Such necessity was also discussed in  [1, Section A.2].
>
> Lastly, although certain problem instances do not have full theoretical guarantees, we observed the empirical strength of our method compared to plain RGD on standard benchmarks (Appendix G.2, G.3). Note even plain RGD does not have theoretical guarantee in these setups, due to either lack of geodesic convexity or smoothness. However, they are still widely used due to their empirical effectiveness. In this regard, we believe the empirical strength of our method additionally validates the effectiveness of our method.
>
> > The method still assumes access to exponential maps and parallel transport on the manifold; this is reasonable for $BW(R^n)$ but can be a practical limitation on more general manifolds where only retractions and vector transports are available.
>
> We thank the reviewer for sharing these valuable thoughts. Extending our method to retraction map and more general vector transport is an interesting direction we are also thinking of, and we have some heuristic thoughts about it. Please see our response to the related question below for more details.
>
> > The step-size rule in Algorithm 1 depends on quantities that must be computed from two consecutive gradients and parallel transports; while explicit, it is more involved than standard RGD and the paper could more clearly quantify this overhead.
>
> We would like to kindly ask the reviewer what the reviewer meant by 'quantify this overhead'. Does it mean the computational complexity? If that is the case, it would depend on the specific choice of the geometry (due to the dependency on Riemannian metric and parallel transport), so we do not think we can explicitly provide the general analysis on it.
>
> That said, at least the Bures-Wasserstein GVI application in Section 5 (BWAdaGVI) has the same computational complexity as the plain Bures-Wasserstein GVI. The only extra steps of BWAdaGVI, compared with plain Bures-Wasserstein GVI, are the computations of $\hat L_{k+1}$ and $s_{k+1}$ in Lines 1933 and 1937. We clarify that we reuse the gradients computed for the algorithm’s iterations, and that extra computation is only needed for the parallel transports and Riemannian metric. These added procedures require only matrix multiplications, square-root, inversion, trace evaluations, and computing the maximum singular value. All of these operations have computational complexity no worse than that of the Bures-Wasserstein gradient computation itself (Line 1927), which already involves a matrix inversion. Therefore, the extra operations in BWAdaGVI do not increase the overall asymptotic computational complexity relative to the standard Bures-Wasserstein GVI method.
>
> We will include this computational complexity analysis in the revised version.

---

> ### Author Response · Authors · 2025-11-21
> **Response (continued)**
>
> > Experiments are focused on a few GVI scenarios and do not yet explore larger-scale or stochastic-gradient settings, which are natural for VI in practice.
>
> We agree with the reviewer, and it is true we did not explored the stochastic setup in this paper. Extending our analysis to stochastic setup is one of our key future directions we would like to delve into deeper. For this paper, we thought Lemma E.2 already provides a sufficient contribution, as these instances were not previously solvable using existing methods (e.g., FBGVI in [2]).
>
> > The GVI convergence result needs an additional uniform lower bound on the covariance eigenvalues, which is plausible but not guaranteed by the algorithm itself.
>
> It is true that the algorithm itself does not guarantee this property. That said, we have identified some ideal cases when such lower bound is theoretically guaranteed (Appendix E.1.4, from Line 2041).
>
> We also conjecture that the issue can be resolved, either by the current algorithm with extra steps such as eigenvalue clipping (which can be viewed as a variant of projected gradient in a broad sense; we partially discussed this direction in Appendix E.1.4, from Line 2091), or by modifying the algorithm through a forward–backward approach as in [2]. We leave this as an interesting direction for future work.
>
> At least in practice, one can consider the smallest eigenvalue clipping by a very small threshold, which is a common practice when dealing with SPDs for numerical stability.

---

> ### Author Response · Authors · 2025-11-21
> **References**
>
> **References**
>
> [1] Altschuler et al. Averaging on the Bures-Wasserstein manifold: Dimension-free convergence of gradient descent. NeurIPS 2021.
>
> [2] Diao et al. Forward-backward Gaussian variational inference via JKO in the Bures–Wasserstein space. ICML 2023.
>
> [3] Huang and Wei. Riemannian proximal gradient methods. Mathematical Programming. 2022.

---

> ### Author Response · Authors · 2025-11-21
> **Answering the Question part**
>
> > 1. Can the authors clarify whether the adaptive rule for \(s_{k+1}\) can be made compatible with retractions/vector transports (as opposed to exact and parallel transport) while preserving the rates, possibly with higher-order error terms?
>
> We believe the extension to the retraction map is feasible, under some technical modifications. When we use the retraction update $x_{k+1} = Ret_{x_k}(-s_k \nabla f(x_k))$, the changes in our analysis happens in Lemma D.1, specifically Line 1434–1445. One needs to substitute $\log_{x_k}x_{k+1} = -s_k \nabla f(x_k)$ by $\log_{x_k}x_{k+1} = -s_k \nabla f(x_k) + e_k$, where $e_k = O(||s_k \nabla f(x_k)||^2)$. Then, the only change is in Line 1444, where we have the additional terms $$-\zeta_{k+1}\langle \log_{x_k}x_{k+1}, e_k \rangle - \zeta_{k+1}||e_k||^2 + 2 \langle e_k, \log_{x_{k+1}}x_* \rangle.$$ Using Cauchy-Schwarz, these additional terms are bounded by $$\zeta_{k+1}||\log_{x_k}x_{k+1}||\\:||e_k|| - \zeta_{k+1}||e_k||^2 + 2 d(x_{k+1}, x_*)||e_k||.$$ Hence, under a priori boundedness assumption on $d(x_{k+1}, x_*)$ and using $\log_{x_k}x_{k+1} = -s_k \nabla f(x_k) + e_k$ again, the above additional terms are at most $O(||s_k \nabla f(x_k)||^2)$.
>
> Plugging-in this result into Lyapunov analysis, we get $V_{k+1} - V_k \leq O(||s_k \nabla f(x_k)||^2)$, and therefore
> $$V_{k} \leq V_{k_0 - 1} + O\left(\sum_{i=k_0-1}^{k-1}||s_i \nabla f(x_i)||^2\right).$$
> Comparing this with the exact exponential map formulation $V_{k} \leq V_{k_0 - 1}$, there is an additional second-order term. A more refined way of stepsize rule seems needed, but at least in the high level, the only difference is a controllable error term by the stepsize rules. Thus, we believe our analysis can be carried over to retraction map up to higher order terms, but with additional technical adjustments to properly control the error by modifying stepsize rules.
>
> If one wants to avoid the use of higher order term argument, one can consider replacing the required inequalities, precisely co-coercivity type inequality (Lemma C.1, Proposition 3.4) and cosine law type inequality (Lemma F.1), in terms of retraction map, the inverse of retraction map, and the retraction-based vector transport instead of exponential map, logarithmic map, and parallel transport. Such practice was done in for example [3, Assumption 2,4]. We believe under this modified assumption, the machinery directly carries over. However, in this case, identifying whether such assumptions are satisfied for certain instances will be another bottleneck.
>
> In sum, under some technical modification (either algorithmic or assumption modification), our analysis can be carried over to retraction map. We stayed with the exponential map for (1) the brevity of the paper, and (2) the exponential map setup was sufficient for our core applications (Section 5).
>
> > 2. In the BW-GVI application, the extra step-size clipping in equation (17) depends on the largest eigenvalue of an expectation involving $\nabla^2 V(X)$. How is this implemented in practice when only Monte Carlo estimates of expectations are available, and how sensitive is convergence to this clipping?
>
> Perhaps the most naive approach would be to directly substitute $E[\nabla^2 V(X)]$ to $\hat{\nabla^2 V(X)}$, the Monte Carlo estimate, and implement the same procedure accordingly. This is certainly feasible as $X$ will be just Gaussian random vector. The Monte Carlo estimate is not problematic for the stepsize clipping (Equation (17)), as the empirical maximum eigenvalue will concentrate to the population maximum eigenvalue.
>
> More specifically, the key is to use Weyl's inequality, which states $\max_{i}|\lambda_i(\hat S_k) -\lambda_i(S_k)| \leq ||\hat S_k - S_k||$, where the norm is the operator norm. Then this problem can be translated to the concentration of $\hat \nabla^2 V(x)$ to $E_{\mu}[\nabla^2 V(x)]$ with respect to the operator norm. Since $\hat{\nabla^2 V(x)}$ is a Monte Carlo estimate, it will concentrate to $E[\nabla^2 V(x)]$ as the number of Monte Carlo samples increases, possibly exponentially fast depending on the form of $\nabla^2 V(x)$.
>
> On the other hand in practice, at least for all of our experiments (Poisson in Section 5, Gaussian and Gamma in Appendix G), with or without this clipping did not yield the significant difference.

---

> ### Author Response · Authors · 2025-11-21
> **Answering the Question part (continued)**
>
> > 3. The convergence proof in Section 4 relies on the compactness/boundedness of the iterates (via the Lyapunov argument). Is there an example of a nonnegatively curved manifold and locally geodesically smooth, generalized geodesically convex function where RAdaGD would fail without this boundedness step?
>
> We would like to clarify that the boundedness of the iterates is the consequence of the algorithmic construction, not the assumption or requirement. When the manifold is possibly negatively curved, as stated in Corollary 4.7, we require information about a bound on a **single** iterate. For non-negatively curved manifolds, $\zeta = 1$, so boundedness is a direct consequence without any further assumptions. Therefore, for non-negatively curved spaces as long as $A_k, B_k$ are properly constructed and assumptions are satisfied, the boundedness is the consequence of the algorithm, and the theoretical guarantees of RAdaGD are valid as well.
>
> > 4. For the Poisson regression GVI example, can the authors provide a small ablation showing RAdaGD vs. a Riemannian method with backtracking line search, to separate the benefit of adaptivity from the benefit of avoiding line search?
>
> We thank the reviewer for their meaningful suggestion.
> We implemented a backtracking line search based on the Armijo condition. The search checks whether the decrease in the objective satisfies the Armijo criterion:
> $$
> f(\exp_{x_k}(\alpha_k d_k)) - f(x_k) > a \langle \nabla f(x_k), d_k \rangle,
> $$
> where $a = 10^{-4}$ and $d_k = -\nabla f(x_k)$. Starting with an initial step size of 1, we repeatedly updated
> $\alpha_k \leftarrow c \alpha_k$ with a contraction factor $c = 0.5$. To avoid an infinite loop, we set the maximum number of backtracks to 40.
>
> For the Poisson regression GVI example, we observed that our algorithm outperforms this line search implementation, which required more than 6,000 additional function evaluations. We obtained similar results for other parameter choices.
> We added the code and the figure for Poisson Regression GVI experiment in our supplementary file.
>
> > 5. The paper mentions possible stochastic variants. Are there obstacles in controlling the estimate of $L_{k+1}$ in (5) under stochastic gradients, or is it mainly a matter of concentration bounds?
>
> Note $L_{k+1}$ appears as a fractional quantity. Controlling the stochasticity of such fractions (ratio-type parameters) is typically more challenging, and we therefore expect that deriving concentration bounds for $L_{k+1}$ will require additional technical care. The main difficulty is that even when the true value is bounded, the stochastic fraction can diverge if its denominator becomes small. To address this, if we write the stochastic gradient as $g(x) = \nabla f(x) + \epsilon(x)$ for some mean-zero noise vector $\epsilon(x)$, then we must impose assumptions on $\epsilon(x)$ that go beyond standard tail conditions. Identifying the precise and minimal assumptions required on $\epsilon(x)$ to ensure concentration of the fractional term is the challenge we expect when extending to the stochastic gradient.

---

> > ### Comment · Reviewer_LVsh · 2025-11-26
> >
> > Dear authors,
> >
> > Thank you for your detailed response addressing all of my questions and concerns. I happily increased my score :)

---

### Official Review · Reviewer_jiEp · 2025-11-07

**Soundness:** 4
**Presentation:** 3
**Contribution:** 3
**Rating:** 8
**Confidence:** 4

**Summary:**

The authors propose adaptive gradient descent methods on Riemannian manifolds. Following the approach introduced in Suh and Ma (2025), they show that the iterates of their algorithm converge in objective value at a rate of $\mathcal{O}(1/k)$ when the potential function $V$ is locally geodesically smooth and generalized geodesically convex. They apply this framework to Gaussian Variational Inference (GVI), establishing provable computational guarantees while relaxing the standard $L$-smoothness assumption.

**Strengths:**

Overall, the paper is carefully written and mathematically rigorous. It extends the work of Suh and Ma (2025) to the setting of optimization on Riemannian manifolds and establishes convergence guarantees for the proposed algorithm. In particular, this work provides a algorithm with convergence results for Gaussian Variational Inference (GVI) even when the target distribution is not log-smooth.

**Weaknesses:**

The main idea appears to be a rather direct extension of Suh and Ma (2025) to the Riemannian setting, with additional technical complications arising from the non-Euclidean geometry. In particular, Algorithm 1 in the paper is largely a manifold-based translation of Algorithm 2 in Suh and Ma (2025). The comparison with the reference paper is somewhat limited and does not sufficiently emphasize the novel challenges or technical difficulties encountered in extending the analysis to the manifold case.
In Section 4.1, the authors mention that the constants $\tilde B_k$ are introduced to handle the terms $\zeta_k$, which are specific to the manifold setting (since $\zeta_k \equiv 1$ in Euclidean geometry). A more detailed explanation and comparison in this direction would be appreciated—especially to clarify how these constants affect the algorithm’s behavior and the convergence analysis.

**Questions:**

1. The citation to \emph{Ansari-Onnestam \& Malitsky (2025)} should be removed, as that paper has been withdrawn.

2. Some discussion should be included for non-convex objectives, clarifying whether the proposed methods or convergence results extend in any way beyond the geodesically convex case.

3. Since only geodesic convexity is assumed, there may exist multiple minimizers. It would be helpful to clarify whether the choice of $x^*$ in the statements of results (e.g., Theorem~4.1) affects the conclusions or constants.

4. In Section 4.2, the authors address the case of manifolds with negative sectional curvature and unbounded $\bar{\zeta}$ by assuming that $d(x_0, x_*)$ is bounded above (in Corollary 4.7). This condition appears vacuously true, since otherwise the constant $R$ defined in Theorem 4.1 would be infinite, implying that the smoothness parameter $L$ should be infinite as well. It is unclear what substantive difference is between Corollary 4.7 here and Corollary 4.6 in the previous section. In particular, as suggested by Corollary 4.7, what happens if we set $\tilde B_k = B_k + \bar{\zeta}_0$?

5. Although the proposed method is a direct extension of Suh and Ma (2025)  to the manifold case, more intuition and explanation should be provided regarding Algorithm~1. For instance, what are the respective roles of $A_k$, $B_k$, and $\tilde B_k$ in the algorithm? Is there an intuitive interpretation of these quantities or of the update mechanism?

6. In line 326, the sentence
    ``The following corollary states that, as a trade-off, if $\zeta$ is known to be bounded, we can still achieve the same asymptotic convergence rate without any prior knowledge of $\zeta$.''
is unclear. What does $\zeta$ refer to here? Is it the function defined in equation~(2), or is this a typo?

7. In Section E.1.3, the proofs of parts (iii) and (iv) are provided, although these statements do not appear in Proposition 5.3. It would be clearer to reorganize the proof into Steps 1--4 instead. In Lemma E.1 (line 1832), the phrase should read:
    ``is a compact set in ${\rm BW}(\mathbb{R}^n)$.''

**Details Of Ethics Concerns:**

N.A.

---

> ### Author Response · Authors · 2025-11-21
>
> We are happy to hear that the reviewer found our paper “carefully written and mathematically rigorous.”
> We also thank the reviewer for their thorough feedback and will consider it carefully when preparing the final version, if our paper is accepted. We address the questions below.
>
> > The comparison with the reference paper is somewhat limited and does not sufficiently emphasize the novel challenges or technical difficulties encountered in extending the analysis to the manifold case.
>
> We thank the reviewer for the suggestion and for giving us the opportunity to address this point in detail. Indeed, some technical challenges were encountered in extending the analysis to the manifold case, which we describe below.
>
> In Section 3, we established important facts related to the function class, which we believe constitute an independent contribution beyond the context of extending the algorithm analysis to manifolds. In particular, the main challenges arose in proving Theorem 3.3 and Proposition 3.4. While looks direct, it actually requires the careful argument on the compactness, as the notion of compactness becomes more strict beyond the Euclidean space.
>
> On the algorithm design side, $\tilde B_k$ is a novel parameter required to handle the curvature parameter $\zeta_k$. By introducing $\tilde B_k$, we identified precisely where the curvature-related constant should enter when generalizing the Euclidean analysis to manifolds. Furthermore, in Section 4.2, dealing with unbounded or unknown $\bar \zeta$ requires additional technical arguments in the proof that would not appear in the Euclidean setup. For Corollary 4.7, we needed a new inductive argument to avoid a circular argument in order to properly show the boundedness of $\zeta_k$.
> For Corollary 4.8, the condition $\lim_{k\to\infty} (\tilde{B}_k - B_k) = \infty$ was not previously considered, and it was required to handle the unknown $\bar{\zeta}$.
>
> For the proof arguments in Section 5, the fact that the Bures-Wasserstein space need not be geodesically complete was perhaps the most technically challenging part of the paper. In the absence of geodesic completeness, the compactness arguments must be addressed through an intricate analysis of the underlying geometry. As a result, the algorithm requires additional adjustments, in particular the modification in Equation (17). The proofs in Appendices E and F show that extending compactness arguments is far from straightforward on geodesically incomplete manifolds, in contrast to the complete manifold setting.
>
> > [...] the authors mention that the constants $\tilde B_k$ are introduced to handle the terms $\zeta_k$, which are specific to the manifold setting [...] clarify how these constants affect the algorithm’s behavior and the convergence analysis.
>
> Basically, the motivation of $\tilde B_k$ is to handle the equation in Line 1408 that originates from Lemma F.1 and to maintain the validity of the Lyapunov analysis. This motivates the condition in inequality (12), where $\tilde B_k$ upper bounds $B_k$ with the additional curvature term $\zeta_k$.
>
> A larger value of $\tilde B_k$ makes condition (12) easier to satisfy. On the other hand, increasing $\tilde B_k$ also leads to smaller step sizes because of the update rules in (6) and (7), and it reduces the value of $r$ appearing in the third inequality of (10). As a consequence, the convergence rate $O(L/(rA_k))$ becomes worse.
>
> Therefore, it is preferable to choose the tightest $\tilde B_k$ while satisfying (12), which corresponds to setting $\tilde B_k$ to reflect the geometry (relevant to $\bar \zeta$). We will elaborate on this in our revision.
>
> > 1. The citation to Ansari-Onnestam & Malitsky (2025) should be removed.
>
> We appreciate the reviewers suggestion, and will remove this reference.
>
> > 2. Some discussion should be included for non-convex objectives, clarifying whether the proposed methods or convergence results extend in any way beyond the geodesically convex case.
>
> Our theoretical guarantee relies on co-coercivity type inequality (4), which requires the generalized geodesic convexity as the sufficient condition to the best of our knowledge. That said, we observed empirically our method is effective under non generalized geodesically convex, or possibly even non geodesically convex setups, as shown in Appendix G.2 and G.3.2.
>
> > 3. Since only geodesic convexity is assumed, there may exist multiple minimizers. It would be helpful to clarify whether the choice of in the statements of results affects the conclusions or constants.
>
> The convergence rate to the minimum function value does not depend on the specific choice of the minimizer. Under this setup, we can only say the qualitative convergence toward a minimizer, as stated in Theorem 4.5. We expect we can extend our analysis to strongly convex setting possibly using the restarting method as in [1], while we did not pursue this generality.

---

> ### Author Response · Authors · 2025-11-21
> **Response (continued)**
>
> > 4. In Section 4.2, the authors address the case of manifolds with negative sectional curvature and unbounded $\bar{\zeta}$ by assuming that $d(x_0, x_\star)$ is bounded above (in Corollary 4.7). This condition appears vacuously true, since otherwise the constant $R$ defined in Theorem 4.1 would be infinite, implying that the smoothness parameter $L$ should be infinite as well. It is unclear what substantive difference is between Corollary 4.7 here and Corollary 4.6 in the previous section. In particular, as suggested by Corollary 4.7, what happens if we set $\tilde{B}_k = B_k + \bar{\zeta}_0$?
>
> The point of the assumption in Corollary 4.7 is not the boundedness of $d(x_0, x_\star)$, but rather the fact that it is bounded by a **known** constant. Recalling the definition in line 203, Corollary 4.6 is a special case of Corollary 4.7, since the known global bound $\bar{\zeta}$ is indeed a bound for $d(x_0, x_\star)$. The point of Corollary 4.7 is that the bound does not need to be a global bound, so the manifold does not need to be bounded. While knowing the bound on $d(x_0, x_\star)$ can still be considered restrictive, Corollary 4.7 at least has a weaker assumption compared to Corollary 4.6, which imposes a common type of assumption in the literature [2, 3].
>
> > 5. What are the respective roles of $A_k$, $B_k$, and $\tilde{B}_k$ in the algorithm? Is there an intuitive interpretation of these quantities or of the update mechanism?
>
> Basically, all these $A_k$, $B_k$, and $\tilde{B}_k$ are designed carefully to satisfy the inequalities in (23), so that the Lyapunov analysis is valid, which is the heart of the convergence analysis.
>
> As we can see, the convergence rates in Theorem 4.5 are written in terms of $A_k$, and a larger $A_k$ guarantees faster convergence. However, the first inequality of (10) implies $A_{k+1} \le A_k + 1 \le A_0 + k + 1$, thus $A_k$ cannot grow arbitrarily large, and its largest possible order is $\mathcal{O}(k)$.
>
> $\tilde B_k$ is a term used to handle the curvature of the manifold. $\tilde B_k$ can be thought of as a variant of $B_k$ that upper-bounds $B_k$ while incorporating an additional curvature term $\zeta_k$, as we can see in (12). $\tilde B_k$ appears in (6) and ensures that (23b) and (23c) hold, which are the core inequalities for $s_{k+1}$ that make the Lyapunov analysis valid. A larger $\tilde B_k$ makes the step size smaller, making (23b) and (23c) easier to satisfy, but a smaller step size often results in slower convergence in practice.
>
> Then one question could be: wouldn’t a smaller $B_k$ allow a smaller $\tilde B_k$, and thus a larger step size?
> However, if $\liminf_{k\to\infty}\frac{A_k}{B_k} = \infty$, then $r_k^L$ may go to zero because of the second option $\left( \frac{A_k}{B_k} + \frac{\tilde{B}_{k+1}}{A_k} \right)^{-1}$, which means the step size vanishes.
> Therefore, $B_k$ should also have the same asymptotic order as $A_k$, and $\tilde B_k$ cannot be made arbitrarily small.
>
> > 6. What does $\bar \zeta$ refer to in Line 326?
>
> It is the same $\bar \zeta$ as in Line 203.
>
> > 7. Section E.1.3, the proofs of parts (iii) and (iv) are provided, although these statements do not appear.
>
> Thank you very much for pointing out this typo. The current result of (ii) in Proposition 5.3--the local geodesic smoothness of KL--is the aggregation of proofs of (ii), (iii), and (iv). We originally provided Proposition 5.3 in four components: (ii)--smoothness of the potential, (iii)--smoothness of the entropy, and (iv)--smoothness of KL, but later combined these into one. During this modification we missed modifying the structure of the proof. We will fix this, by steps as the reviewer suggested, to make the proof clear.
>
> > 7. In Lemma E.1 (line 1832), the phrase should read: "is a compact set in $BW(R^n)$."
>
> We appreciate the reviewer for finding the typo. We will fix the typo.

---

> ### Author Response · Authors · 2025-11-21
> **References**
>
> **References**
>
> [1] Park et al. Acceleration via silver stepsize on Riemannian manifolds with applications to Wasserstein space. NeurIPS 2025.
>
> [2] Zhang and Sra. First-order Methods for Geodesically Convex Optimization. COLT 2016.
>
> [3] Kim and Yang. Accelerated Gradient Methods for Geodesically Convex Optimization: Tractable Algorithms and Convergence Analysis. ICML 2022.

---

> > ### Comment · Reviewer_jiEp · 2025-11-26
> >
> > Dear authors,
> >
> > Thank you for your detailed response addressing all of my questions and concerns.

---

### Meta-Review · Area_Chair_RaSJ · 2026-01-08

**Summary:**

The paper proposes an adaptive gradient descent method, which automatically adapts the step size parameter without using a line search and proves O(1/k) convergence. The algorithm is applied to the Gaussian variational inference problem with promising empirical results.

Reviewers unanimously liked the submission, and concerns about relation to very recent related works, and questions about constants in the convergence bound are adequately addressed by the rebuttal. However, I share the reviewer's overall concerns about the limited and slightly sparse experimental section. Comparisons and also citations to other Gaussian variational inference methods (such as, natural-gradient methods which use Fisher information metric rather than Wasserstein metric, recent proximal BBVI methods, etc.) are missing and will make the paper more complete.

However, as the work's primary aim is to establish analysis of adaptive gradient descent on Riemannian manifolds, and all reviewers appreciated that contribution, the paper is recommended for acceptance.

For the final version, I encourage the authors to conduct more thorough extensive numerical results and also significantly improve the presentation of the experimental results. Text and lines in Figure 1 are very small and hard to read, and there is plenty of space left and right of the figure.

**Reviewer Concerns:**

Reviewer concerns about connections to recent related works, and detailed questions about the convergence result were well-addressed in my opinion. Reviewer's concerns about experimental evaluation remain largely outstanding.

**Reviewer Scores:**

Reviewers gave high scores and would have likely maintained or increased them -- some reviewers indicated that all concerns have been addressed and to increase their score in response to the authors.

---

### Decision · Program_Chairs · 2026-01-26

Accept (Poster)